# The Rashomon Set Has It All: Analyzing Trustworthiness of Trees under Multiplicity

Ethan Hsu[1]    Tony Cao[1]    Lesia Semenova[2*]    Chudi Zhong[3]

[1] Duke University [2] Rutgers University [3] UNC Chapel Hill
{ethan.hsu, tony.cao}@duke.edu
lesia.semenova@rutgers.edu, chudi@unc.edu

## Abstract

In practice, many models from a function class can fit a dataset almost equally well. This collection of near-optimal models is known as the Rashomon set. Prior work has shown that the Rashomon set offers flexibility in choosing models aligned with secondary objectives like interpretability or fairness. However, it is unclear how far this flexibility extends to different trustworthy criteria, especially given that most trustworthy machine learning systems today still rely on complex specialized optimization procedures. *Is the Rashomon set all you need for trustworthy model selection? Can simply searching the Rashomon set suffice to find models that are not only accurate but also fair, stable, robust, or private, without explicitly optimizing for these criteria?* In this paper, we introduce a framework [2] for systematically analyzing trustworthiness within the Rashomon set and conduct extensive experiments on high-stakes tabular datasets. We focus on sparse decision trees, where the Rashomon set can be fully enumerated. Across seven distinct metrics, we find that the Rashomon set almost always contains models that match or exceed the performance of state-of-the-art methods specifically designed to optimize individual trustworthiness criteria. These results suggest that for many practical applications, computing the Rashomon set once can serve as an efficient and effective method for identifying highly accurate and trustworthy models. Our framework can be a valuable tool for both benchmarking Rashomon sets of decision trees and studying the trustworthiness properties of interpretable models.

## 1 Introduction

With the increasing use of machine learning (ML) in high-stakes domains such as healthcare, lending, and criminal justice, the demand for models that satisfy multiple trustworthiness criteria, such as interpretability, robustness, fairness, privacy, and regulatory compliance, has grown substantially [53, 65, 82, 87]. However, achieving these properties in practice remains difficult, especially when they must be satisfied simultaneously. Most algorithms address only one aspect of trustworthiness at a time, typically by adding a new objective or constraint to the loss function. This often requires solving a specialized (usually non-convex) optimization problem which is tailored to that specific criterion, and is rarely transferable across objectives. As a result, developing trustworthy ML systems today often means solving a different optimization problem for each property, which is computationally expensive, resource-intensive, and can be infeasible in legally constrained environments.

Recent work challenges the assumption that separate optimization is always necessary. Building on the Rashomon Effect [10], which describes the existence of many models that can perform nearly

---

*The work was done while at Microsoft Research NYC.

[2] https://github.com/EtHsu0/rashomon-framework

as well as the best one for a given dataset, researchers have proposed methods to construct and analyze the Rashomon set, the collection of near-optimal models [6, 28, 43, 44, 62, 74, 95, 98]. These methods make it possible to enumerate or approximate the Rashomon set across different hypothesis spaces, enabling new approaches to model selection that do not require additional retraining [71]. This raises a fundamental question: *Can the Rashomon set already contain models that satisfy trustworthiness goals, without the need for separate, objective-specific optimization?*

To answer the question, we introduce a framework for systematically evaluating trustworthiness within the Rashomon set. We focus on decision trees and their Rashomon sets because they are interpretable, simple, and well-suited for high-stakes decision-making problems [70]. We use TreeFARMS [95], which can enumerate all sparse trees within the epsilon loss of the optimal tree, to study whether this full set of near-optimal models inherently contains ones that naturally satisfy a wide range of trustworthiness properties. Our framework supports seven different trustworthy measures, including (1) adversarial robustness [48], (2) stability to data perturbations [47], (3) protection of privacy against membership attacks [75], (4) unlearning a small portion of data [8], and fairness metrics, such as (5) statistical parity [25], (6) equalized odds and (7) equal opportunity [39]). We systematically compare the sparse trees in the Rashomon set with trees optimized for specific criteria.

Our contributions include (1) introducing an open and extensible evaluation framework with standard datasets, baselines, trustworthy metrics and attacks, and evaluation protocols, enabling reproducible research on trustworthy model selection and interpretable model evaluation; (2) showing that the Rashomon set of sparse decision trees often contains models that match or exceed the performance of specialized models across multiple trustworthiness criteria, easing the need for specialized optimization per criterion; (3) showing that models optimized for one property (e.g., fairness) do not always generalize to others (e.g., robustness), motivating the explicit model selection within the Rashomon set instead of separate optimization.

Our findings suggest that enumerating near-optimal models, rather than retraining for each new objective, offers a practical and principled strategy for building responsible and trustworthy ML systems. By leveraging the natural diversity within the Rashomon set, practitioners can select models that align with application-specific constraints and learn tradeoffs between trustworthiness criteria, thereby bridging the gap between theoretical insight and real-world deployment.

## 2   Related Works

We discuss related work on the Rashomon Effect, decision trees, and trustworthy benchmarks and frameworks.

**Rashomon Effect.** The Rashomon set, a formal quantification of the Rashomon Effect, contains multiple different models that achieve approximately equal performance [28, 43, 62, 71, 74, 95]. Recent work in this area can be broadly categorized into those that focus on computing and characterizing the Rashomon Effect and the Rashomon set [43, 62, 88, 95, 98] and those that study the implications of large Rashomon set for different applications and trustworthy machine learning as a whole [6, 32, 89]. In this paper, we focus on TreeFARMS [95], which finds the Rashomon set of sparse decision trees. Many works in this domain that focus on understanding fairness and less discriminative hypothesis in the presence of a large Rashomon set are the closest to our work [7, 18, 20, 33, 52, 64]. However, none of the prior works consider multiple trustworthy criteria within one Rashomon set of interpretable models.

**Decision trees.** Decision trees are among the most popular methods in interpretable machine learning. Recent advances can find sparse optimal trees using either mathematical programming solvers [1, 4, 5, 26, 36, 80, 81] or dynamic programming with branch-and-bound [2, 21, 58, 63]. Recent research also incorporates other metrics, such as fairness [46, 79], robustness [12, 13, 14, 37, 47, 83, 84], and privacy [85], into the optimization problem, aiming to make sparse decision trees align with more trustworthy principles. Despite these advancements, there has been no systematic evaluation of these algorithms, nor has any study specifically examined if sparse decision trees that achieve high accuracy naturally exhibit trustworthy properties without being explicitly optimized for them.

**Trustworthy Benchmarks and Frameworks.** Trustworthiness in ML has emerged as a critical concern, especially as AI systems are increasingly deployed in high-stakes environments. While trustworthiness encompasses a broad spectrum of principles, metrics such as interpretability, robust-

ness, privacy, and fairness consistently emerge as essential components [49, 54]. Benchmarks have been developed for robustness [19, 22, 40, 76], privacy [67, 78], and fairness [3, 38, 90]. Beyond individual trustworthiness benchmarks, some comprehensive trustworthiness benchmarks have been proposed [45, 72]. However, existing benchmarks primarily focus on deep learning models, leaving interpretable models, such as sparse decision trees, largely unexamined. Given their extensive use in healthcare, finance, and criminal justice, evaluating sparse decision trees under a rigorous trustworthiness framework is crucial.

In this work, we develop a framework for interpretable models, assessing robustness, privacy, and fairness while leveraging the Rashomon set as a unifying concept (see Section 3). Our framework enables researchers to explore whether models within the Rashomon set can naturally satisfy multiple trustworthiness criteria without sacrificing accuracy, providing a new perspective on the design and evaluation of trustworthy interpretable models. In Section 4, we provide empirical evidence that the Rashomon set often contains trustworthy models and analyze sparsity, timing, and cross-property behavior using our framework.

## 3 Background and Evaluation Framework

Our framework provides a systematic approach for evaluating the trustworthiness of models within the Rashomon set of sparse decision trees. It integrates five evaluation components, including robustness, stability to noise, membership inference, machine unlearning, and fairness. For each criterion, we define quantitative metrics, select state-of-the-art baseline algorithms that explicitly optimize for that property, and apply standardized datasets and evaluation protocols for fair comparison. We formally define the Rashomon set of sparse decision trees next and then focus on each trustworthy property.

### 3.1 The Rashomon set of sparse decision trees

Let $\{(\boldsymbol{x}_i, y_i)\}_{i=1}^{n}$ be the training dataset, where $\boldsymbol{x}_i \in \{0,1\}^p$ are binary features and $y_i \in \{0,1\}$ are labels. Let $\ell(t, \boldsymbol{x}, \boldsymbol{y}) = \frac{1}{n}\sum_{i=1}^{n} 1[\hat{y}_i \neq y_i] + \lambda H_t$ be the loss of tree $t$ on the training set, where $\hat{y}_i = t(\boldsymbol{x}_i)$, $H_t$ is the number of leaves in tree $t$ and $\lambda$ is a regularization parameter. The loss function controls both the misclassification loss and the sparsity of the tree. Following the definition in Semenova et al. [74] and Xin et al. [95], we define the Rashomon set of sparse decision trees as follows: Let $t_{\text{ref}}$ be a reference model from $\mathcal{T}$, where $\mathcal{T}$ is a set of binary decision trees. The $\epsilon$-Rashomon set is a set of all trees $t \in \mathcal{T}$ with $\ell(t, \boldsymbol{x}, \boldsymbol{y})$ at most $\ell(t_{\text{ref}}, \boldsymbol{x}, \boldsymbol{y}) + \epsilon$: $R_{\text{set}}(\epsilon, t_{\text{ref}}, \mathcal{T}) := \{t \in \mathcal{T} : \ell(t, \boldsymbol{x}, \boldsymbol{y}) \leq \ell(t_{\text{ref}}, \boldsymbol{x}, \boldsymbol{y}) + \epsilon\}$.

Typically, the reference model is an empirical risk minimizer $t_{\text{ref}} \in \arg\min_{t \in \text{trees}} \ell(t, \boldsymbol{x}, \boldsymbol{y})$. Xin et al. [95] propose the TreeFARMS algorithm, the first method to construct the Rashomon set to find *all* good sparse decision trees. It uses mathematical bounds to prune infeasible spaces, dynamic programming for computation reuse, and the model set representation to extract and store the entire Rashomon set. TreeFARMS can find millions of good sparse trees within a short amount of time (within seconds or minutes, depending on the dataset size, see Section 4.4).

While TreeFARMS can enumerate all good sparse trees, it remains unclear whether models within the Rashomon set inherently satisfy trustworthiness principles. Also, in the presence of thousands or millions of near-optimal trees, it might not be clear which model to choose for deployment. Next, we investigate these questions by benchmarking sparse trees from the Rashomon set and other tree methods across multiple trustworthiness criteria.

### 3.2 Robustness

Robustness ensures models maintain performance under various conditions such as adversarial perturbations and data noise [30, 59]. Here, we focus on *adversarial robustness* and *stability* to random perturbations and investigate whether robust models are naturally contained in diverse Rashomon sets.

***Adversarial robustness*** measures the ability of a machine learning model to correctly classify inputs that have been intentionally perturbed through white-box or black-box attacks [34]. Given that decision trees are inherently interpretable, meaning their structure is humanly understandable, we primarily consider white-box attacks (attacks with information about the model). Specifically, we

consider evasion-style attacks, which aim to minimally perturb an input to cause misclassification. Given a dataset $\mathcal{D} = \{(\boldsymbol{x}_i, y_i)\}_{i=1}^n$, and a tree $t \in \mathcal{T}$, Kantchelian et al. [48] propose an algorithm that generates adversarial examples $\boldsymbol{x}_i'$, such that the misclassification error of $t$ is maximized on the dataset $\mathcal{D}' = \{(\boldsymbol{x}_i', y_i)\}_{i=1}^n$. In other words, if $t(\boldsymbol{x}) = y$, the algorithm outputs a perturbed point $\boldsymbol{x}'$ that results in $t(\boldsymbol{x}') \neq y$. The perturbations are constrained such that $\|\boldsymbol{x}_i' - \boldsymbol{x}_i\|_\infty \leq \theta$, where $\theta \in \mathbb{R}^+$ specifies the strength of the attack. If no such $\boldsymbol{x}_i'$ exists under the constraint, the original input $\boldsymbol{x}_i$ remains unchanged. We create an evaluation set $\mathcal{D}^{\text{adv}} = \{(\boldsymbol{x}_i^{\text{adv}}, y_i)\}_{i=1}^n$, where for each $\boldsymbol{x}_i$, we take the nearest adversarial example $\boldsymbol{x}_i'$, and apply the distance based on $\theta$: $\boldsymbol{x}_i^{\text{adv}} = \boldsymbol{x}_i + \theta \frac{\boldsymbol{x}_i - \boldsymbol{x}_i'}{|\boldsymbol{x}_i - \boldsymbol{x}_i'|}$.

Many prior works focus on improving the adversarial robustness of decision trees against this attack. Common approaches include those that globally optimize over the space of decision trees, such as ROCT-V [84]; those that greedily focus on local optimizations using adversarially modified impurity measures, such as GROOT [83]; and those that construct decision trees with theoretically provable robustness guarantees, such as FPRDT [37]. We select the most recent methods from each category as baselines: ROCT-V, GROOT, and FPRDT. We also include the greedy method CART [9].

**ROCT-V** [84] finds optimal robust decision trees. It frames robust tree learning as a min-max problem over the 0-1 loss and solves it using mixed-integer programming (MIP).

**FPRDT** [37] is a greedy recursive approach for constructing robust decision trees. It directly minimizes the adversarial 0-1 loss by making a tradeoff between global and local optimizations over the potential splitting features and thresholds. FPRDT has a computational complexity of $O(n \log n)$, which is the smallest among all provably robust decision trees.

**GROOT** [83] makes greedy splits according to the adversarial Gini impurity – a splitting criterion that measures the worst-case Gini impurity after an attacker has maximally worsened the split by moving points within a specified perturbation range. Since impurity is concave to the number of modified data points, GROOT uses its analytical solution to compute the function in constant time.

*Stability* in our context refers to a model's ability to maintain accurate predictions under natural perturbations of the input data. We follow the approach from Justin et al. [47] to evaluate this property. First, for every feature with index $j \in \{0, \ldots p\}$, a "confidence level" $q_t^j$ is sampled from a normal distribution: $q_t^j \sim \mathcal{N}(\rho, \sigma)$, where $\rho$ and $\sigma$ are the normal distribution parameters. The value $q^j$ represents the likelihood that the feature $j$ remains unperturbed. If $q_t^j = 1$, then no perturbation occurs. Next, the noise is sampled as $\xi_i^j \sim S_i^j \cdot (G_i^j - 1)$, where $G_i^j \sim \text{Geom}(q_i^j), S_i^j = 2 \cdot B_i^j - 1$ with $B_i^j \sim \text{Bernoulli}(0.5)$.

Intuitively, $G_i^j$ represents the strength of the noise for feature $j$ of sample $i$, and $S_i^j$ determines the sign, ensuring equal probability of positive and negative noise. Because features and splits are integer-valued, this symmetric geometric step targets flips/threshold crossing: any nonzero step flips a binary feature, and a split at threshold $\theta$ is crossed iff $|\xi_i^j| \geq d(x_i^j, \theta)$ (the integer distance to $\theta$). The new dataset is then $\mathcal{D}^{\text{stab}} = \{(\boldsymbol{x}_i^{\text{stab}}, y_i)\}_{i=1}^n$, where $x_i^{\text{stab},j} = x_i^j + \xi_i^j$.

We use **ROCT-N** [47] as the stability baseline. It finds a globally optimal robust tree using a two-stage robust optimization approach. The first stage determines the tree structure to maximize correctly classified training samples under worst-case perturbation, where confidence levels $q_t^j$ define the uncertainty set for each feature $j$. The second stage optimizes the classification of training samples after observing the worst-case perturbation. This problem is formulated as a mixed-integer program and solved using a customized Benders decomposition algorithm.

For both robustness and stability, we evaluate CART, FPRDT, GROOT, ROCT-N, ROCT-V, and Tree-FARMS across 8 datasets (Table 5) with standard five-fold nested cross-validation. The configuration for each method is in Appendix A.

### 3.3 Privacy

Protecting privacy is essential for machine learning systems that handle sensitive or personally identifiable data. Among many privacy threats studied in the literature, we focus on membership inference attacks and machine unlearning.

### 3.3.1 Membership Inference Attack

Membership inference attacks (MIAs) aim to determine whether a particular data point was used to train a given model [75]. We evaluate whether sparse decision trees within the Rashomon set are inherently resistant to MIAs compared to explicitly private trees. To comprehensively evaluate privacy, our setup includes both defense mechanisms and attack mechanisms.

Existing defenses against MIAs fall into two broad categories: theoretical guarantees such as differential privacy [24], and empirical defenses that aim to reduce overfitting or confidence leakage. In this work, we focus on the former and compare TreeFARMS to representative differentially private decision-tree algorithms. Differential privacy offers formal protection against MIAs by injecting randomness during training. Formally, an algorithm $\mathcal{M}$ is $(\eta_1, \eta_2)$-differentially private if for all datasets $\mathcal{D}, \mathcal{D}'$ that differ on a single element, and for any $S \subseteq \text{range}(\mathcal{M})$, $P[\mathcal{M}(\mathcal{D}) \in S] \leq e^{\eta_1} P[\mathcal{M}(\mathcal{D}') \in S] + \eta_2$, where $\eta_1, \eta_2 \geq 0$. The basic idea of differential privacy is to ensure that individual data points cannot be identified while still preserving overall data utility. We include three representative DP tree algorithms as baselines: PRIVA [85], BDPT [35], and DPLDT from DiffPrivLib [41]. These greedy algorithms introduce randomness into split selection and leaf labeling to ensure privacy, though often at the cost of predictive performance.

**PRIVA** [85] first determines quantile-based bins for numerical features and then uses private histograms to select good splits with minimal privacy budget. It partitions the data recursively until it reaches leaf nodes, which are then labeled using a noise-based majority vote mechanism.

**BDPT** [35] builds the tree top-down. The best splitting attribute is chosen based on the Gini index via the exponential mechanism, and continuous attributes are discretized using down-sampling. At each leaf node, noisy class counts are calculated (using Laplace noise) to determine the final label.

**DPLDT** [41] implements the randomized split tree from [29]. Each tree is built by randomly selecting a feature at each node and partitioning the data accordingly. The leaf nodes use the Exponential Mechanism (with smooth sensitivity) to output only the majority label to maintain differential privacy.

To evaluate the privacy of both TreeFARMS and DP tree models described above, we adopt four representative MIA algorithms from the literature, varying in the level of adversarial knowledge and access to the model, ranging from simple prediction-based heuristics to shadow-model training.

- The **baseline attack** [97] infers membership based on the correctness of the model's predictions: a sample is considered a member if the model predicts it correctly. This attack serves as a reference point for other methods, relying purely on the extent of model overfitting.
- The **label-only attack** [55] requires only hard class predictions. It infers the membership by estimating the minimum perturbation needed to flip the predicted label. If this distance exceeds a threshold, the sample is inferred to be a member. We calibrate a membership threshold at the 50th percentile of distance scores computed on a pool of unlabeled points.
- The **label-only supervised attack** [15] is a stronger variant of label-only attack, which leverages partial knowledge of the data distribution to calibrate the threshold. In experiments, the attacker is given 500 reference samples with known membership status to determine the threshold.
- The **shadow model attack** [75] trains auxiliary models to mimic the target model and uses a separate classifier to predict the membership.

We evaluate on a balanced dataset of 1000 samples, where 500 samples are randomly sampled from the training set (members) and 500 from the test set (non-members). For two label-only attacks, we allow up to five perturbation queries per sample. Detailed configurations are in Appendix B.1.

### 3.3.2 Machine Unlearning

Machine unlearning refers to the process of removing the influence of specific training data from a trained model, ensuring that the model behaves as if those data points had never been used [8, 68]. It has become an important topic, with extensive work in deep learning and ensemble models [11, 73, 92], but little analysis has been conducted on sparse trees. To address this gap, besides TreeFARMS, we consider two unlearning algorithms in our framework: data removal-enabled (DaRE) forests [11] and GBDT unlearning [57].

**DaRE** [11] is an unlearning algorithm for random forests that leverages randomness and caching to enable efficient unlearning. The trees in the forest can follow a greedy top-down approach, referred

to as G-DaRE, or incorporate random layers in the top, where splits are chosen uniformly at random, referred to as R-DaRE.

**GBDT unlearning** [57] provides an algorithm for gradient boosted trees that uses intermediate data statistics to decide if subtrees need retraining. It uses random split point selection to limit split values and optionally adds random layers to restrict retraining to a subset of subtrees. The standard version without random layers is called G-Boosting, while the version with random layers is R-Boosting.

Unlearning in TreeFARMS requires *no* retraining. Instead, we can directly search within the Rashomon set to find an optimal tree after removing data points. Theorem 5.3 in [95] states that if $\epsilon \geq \frac{2K}{n}$, where $K$ is the number of removed points and $n$ is the original data size, the optimal tree after removal remains in the Rashomon set trained on the full dataset. Thus, with a properly chosen $\epsilon$, the optimal tree can always be found in the Rashomon set, regardless of which or how many (up to $K$) samples are removed.

We evaluate these methods on 5 datasets. Configurations for each method are in Appendix B.2.

## 3.4 Fairness

We consider three group fairness metrics: statistical parity, equal opportunity, and equalized odds. Throughout the paper, we report each metric as a *parity score* in $[0, 1]$, where higher is better (1 indicates perfect parity). Let $Y, \hat{Y}, A$, and $\mathcal{A}$ be random variables for labels, predicted labels, sensitive features, and a set of possible values for $A$, respectively.

**Statistical Parity** (sp) [25] requires $P(\hat{Y} = 1 \mid A = a) = P(\hat{Y} = 1 \mid A = b)$. We report $\mathrm{sp} := 1 - |P(\hat{Y} = 1 \mid A = 0) - P(\hat{Y} = 1 \mid A = 1)|$.

**Equal Opportunity** (eopp) [39] requires $P(\hat{Y} = 1 \mid Y = 1, A = a) = P(\hat{Y} = 1 \mid Y = 1, A = b)$. We report $\mathrm{eopp} := 1 - |P(\hat{Y} = 1 \mid Y = 1, A = 0) - P(\hat{Y} = 1 \mid Y = 1, A = 1)|$.

**Equalized Odds** (eodds/eo) [39] requires $P(\hat{Y} = 1 \mid A = a, Y = y) = P(\hat{Y} = 1 \mid A = b, Y = y)$ for $y \in \{0, 1\}$. We report $\mathrm{eodds} := 1 - \max_{y \in \{0,1\}} |P(\hat{Y} = 1 \mid A = 0, Y = y) - P(\hat{Y} = 1 \mid A = 1, Y = y)|$.

We evaluate the fairness performance of trees within the Rashomon set and trees from optimal fair tree algorithms, DPF [79] and FOCT [46]. Both are in-processing methods that incorporate fairness constraints directly into the optimization process. We also include the greedy method CART [9] and a post-processing method LinearPost [93, 94] as baselines.

**DPF** [79] finds an optimal tree with a given depth that minimizes misclassification loss with a statistical parity constraint. This constraint ensures that the difference in positive rates does not exceed a predefined threshold $\delta$. The optimization problem is solved using dynamic programming, and a custom bound is used to prune partial solutions that cannot lead to the optimal fair tree.

**FOCT** [46] formulates the optimization problem as a MIP problem, which a mathematical solver then solves. This algorithm can incorporate fairness constraints using the above-mentioned metrics, ensuring that the absolute difference remains within a predefined threshold $\delta$.

**LinearPost** [93] is a post-processing method to achieve fairness by linearly transforming the predictions of the base classifier. A tolerance $\alpha$ controls the tradeoff between accuracy and fairness. LinearPost is model-agnostic, and we use CART and Gradient-Boosted Tree as base predictors, denoted as PostCART and PostGBT.

We consider binary classification with binary sensitive features. The sensitive feature is not used in training but is used for evaluation. Detailed setup is available in Appendix C. Note that DPF is only optimized for statistical parity, whereas other methods can be applied to all three fairness metrics.

## 4 Experimental Results

Our evaluation aims to answer the following questions: **Q1**. How do models in the Rashomon set compare to baselines optimized for specific trustworthiness criteria, such as robustness, privacy, and fairness (Section 4.1)? **Q2**. What proportion of models in the Rashomon set outperforms baseline models on key metrics, and how consistently does this occur across datasets (Section 4.2)? **Q3**.

What connections, if any, exist between model sparsity and trustworthy properties, and how do these insights inform model selection (Section 4.3)? **Q4**. How does the computational cost of computing the Rashomon set compare to training separate models optimized for individual criteria (Section 4.4)? **Q5**. Are there observable interactions between different trustworthiness criteria (e.g., do fair models also tend to be robust) (Section 4.5)?

To address these questions, we conducted a comprehensive empirical analysis of sparse decision trees in the Rashomon set and compared them to baselines using our evaluation frameworks described in Section 3. We use up to eight different datasets for each metric (see Table 5,11,17, and 20). We also provide results for other hypothesis spaces in Appendix D.

## 4.1 Q1. The Rashomon Set of Sparse Decision Trees Contains Trustworthy Models

**Adversarial Robustness**. Figure 1 compares the adversarial accuracy of trees within the Rashomon set to various baseline models on the test set. The light blue density plot represents the distribution of adversarial accuracy for all trees in the Rashomon set. Vertical lines indicate the accuracy of baseline models, including CART, FPRDT, GROOT, and ROCT, as well as some trees from the Rashomon set, such as the optimal tree, the tree with the fewest leaves, the tree with the most leaves, and RSET_kan, a tree chosen from the held-out selection set. Many trees within the Rashomon set achieve higher accuracy than the baselines, indicating their robustness against adversarial attacks. Also, the selected tree, RSET_kan, can perform comparably to or even better than baselines on the test set.

**Stability**. Figure 1 compares the stability of tree models when random perturbations are added to the data. Note that only ROCT-N explicitly accounts for these perturbations during training, while all other methods are trained normally or for robustness and evaluated for stability on perturbed test data. As we can see, different representative trees from the Rashomon set (bars colored in blue palette) and CART generally perform better than other methods. Although ROCT-N is designed to optimize for robustness against perturbations, its performance on the test set does not dominate other methods. This suggests that while the Rashomon set is not explicitly designed for stability, it contains trees that perform well under perturbations. More results are in Appendix A.

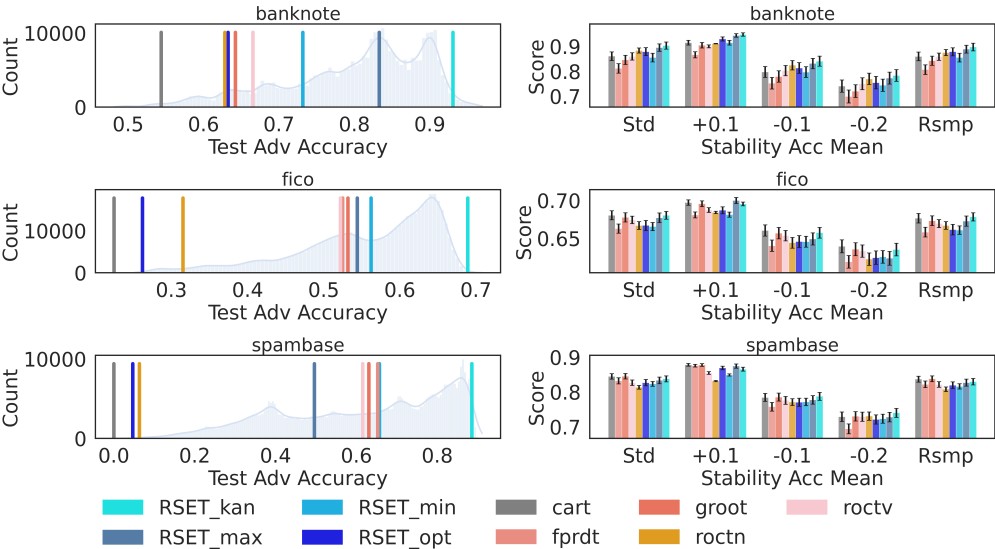

Figure 1: Left: Comparison of adversarial accuracy between trees in the Rashomon set (light blue histogram) and baseline models (vertical lines) on the test set. Most Rashomon set trees achieve higher robustness than baselines. Right: Stability comparison of different methods under random perturbations. Trees in the Rashomon set (colored in a blue palette) generally perform better than baselines.

**Membership Inference Attacks (MIAs)**. Table 1 shows the attack success rate when two different MIAs are applied to various methods. A lower value indicates that the method is more resistant to attacks. Overall, these attacks are not successful as the highest observed accuracy remains close

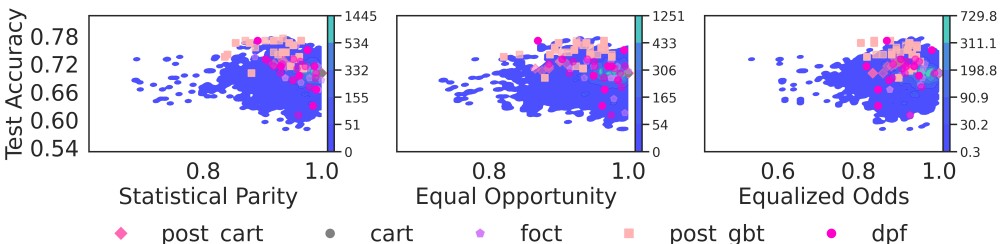

Figure 2: Test accuracy vs. fairness for trees in the Rashomon set (blue density contours) and baselines (dots) at depth 2 on german-credit dataset. Higher values on both axes indicate better performance. The colorbar displays the number of trees in the Rashomon set at each contour level.

to 0.5, meaning the attacker does not significantly outperform random guessing. Compared to 3 differential private tree methods, trees from the Rashomon set with the fewest leaves (RSET_min) achieve comparable against both attacks, despite the fact that the TreeFARMS algorithm does not incorporate any explicit randomness for privacy protection. Results for more attacks are in Appendix B.1.

Table 1: Membership inference attack success rates. Lower values indicate better resistance to attacks. Attack failed when no effective perturbation was found.

| Attack | Data | BDPT | DPLDT | PRIVA | RSET_min |
|---|---|---|---|---|---|
| Label-only | adult | **Attk Failed** | $0.513 \pm 0.012$ | **$0.498 \pm 0.010$** | $0.509 \pm 0.017$ |
| | bank | **Attk Failed** | **Attk Failed** | **$0.503 \pm 0.011$** | **Attk Failed** |
| | compas | **Attk Failed** | $0.517 \pm 0.010$ | $0.513 \pm 0.012$ | **$0.494 \pm 0.016$** |
| | credit-fusion | **Attk Failed** | **$0.504 \pm 0.010$** | $0.507 \pm 0.015$ | $0.505 \pm 0.017$ |
| | oulad | **Attk Failed** | **Attk Failed** | **Attk Failed** | **Attk Failed** |
| Shadow | adult | **$0.492 \pm 0.017$** | **$0.492 \pm 0.017$** | $0.496 \pm 0.013$ | $0.502 \pm 0.022$ |
| | bank | $0.496 \pm 0.006$ | $0.498 \pm 0.004$ | **$0.495 \pm 0.005$** | $0.508 \pm 0.014$ |
| | compas | **$0.498 \pm 0.010$** | $0.506 \pm 0.020$ | $0.501 \pm 0.011$ | $0.506 \pm 0.016$ |
| | credit-fusion | $0.506 \pm 0.003$ | $0.506 \pm 0.015$ | $0.502 \pm 0.009$ | **$0.500 \pm 0.006$** |
| | oulad | $0.503 \pm 0.007$ | $0.502 \pm 0.007$ | $0.503 \pm 0.007$ | **$0.490 \pm 0.016$** |

Table 2: Proportion of test data with different predicted labels between the unlearned and retrained models (i.e., $\hat{y}_{\text{unlearn}} \neq \hat{y}_{\text{retrain}}$) after 1% of the training data are randomly removed.

| Data | RSET | G-DaRE | G-Boosting |
|---|---|---|---|
| adult | 0 | $0.018\% \pm 0.013\%$ | $0.064\% \pm 0.027\%$ |
| bank | 0 | $0.607\% \pm 0.103\%$ | $0.888\% \pm 0.096\%$ |
| carryout | 0 | $1.294\% \pm 0.465\%$ | $5.548\% \pm 0.741\%$ |
| compas | 0 | $0.000\% \pm 0.000\%$ | $0.478\% \pm 0.543\%$ |
| restaurant | 0 | $1.902\% \pm 0.433\%$ | $4.746\% \pm 0.623\%$ |

**Machine Unlearning**. Table 2 shows the proportion of test data with different predicted labels between the unlearned and retrained models for the Rashomon set, G-DaRE, and G-Boosting after 1% of the original training data are randomly removed. Since the optimal tree after data deletion is guaranteed to be within the Rashomon set if $\epsilon$ is set appropriately, the RSET column always reports 0. In contrast, such a guarantee does not hold for other methods. More results are in Appendix B.2.

**Fairness**. Figure 2 compares the test accuracy and fairness of trees within the Rashomon set to baseline models at depth 2, with each figure representing a different fairness metric. A higher value on both axes (top-right corner) indicates better performance. The blue density contours represent the distribution of trees in the Rashomon set. Baseline models trained with different tolerance $\alpha$ are shown as dots. The results show that the contours usually overlap with the baseline points and cover the region close to the top-right corner, indicating that the Rashomon set contains trees with performance comparable to the baselines. Appendix C displays results on 6 datasets at more depths.

Table 3: Percentage of models in Rashomon set that perform better than baselines for adversarial robustness (left) and outperforming *all* selected models from baselines for fairness at depth 4 (right) on the test set.

| Robustness | FPRDT | ROCT-V | Fairness | SP | EOpp | EO |
|---|---|---|---|---|---|---|
| banknote | 95.5 ± 6.0% | 95.5 ± 6.0% | adult | 0.0 ± 0.0% | 8.7 ± 4.9% | 0.3 ± 0.3% |
| blood | 28.5 ± 39.6% | 26.9 ± 40.4% | bank | 71.0 ± 27.1% | 45.0 ± 17.5% | 49.6 ± 12.3% |
| breast | 63.0 ± 38.8% | 62.8 ± 29.8% | compas | 0.0 ± 0.0% | 1.6 ± 3.4% | 0.0 ± 0.0% |
| compas | 27.4 ± 8.1% | 68.4 ± 39.6% | german-credit | 37.8 ± 28.1% | 34.2 ± 30.8% | 20.7 ± 21.5% |
| diabetes | 51.6 ± 29.8% | 53.9 ± 27.4% | oulad | 41.2 ± 13.6% | 28.7 ± 14.1% | 60.9 ± 15.9% |
| spambase | 39.3 ± 18.9% | 45.1 ± 18.7% | student-por | 15.2 ± 15.4% | 29.6 ± 32.1% | 14.7 ± 11.4% |

Table 4: Mean training time, in seconds, for TreeFARMS and baselines averaged over eight robustness datasets (left) and six fairness datasets (right) for depth 2.

| | ROCT-V | ROCT-N | RSET | DPF | FOCT | RSET |
|---|---|---|---|---|---|---|
| **Mean Fit Time** | 1525.635 | 1828.071 | 139.45 | 0.04 | 3444.06 | 0.57 |

## 4.2 Q2. Many Near-Optimal Trees are Trustworthy

Figure 1 and 2 have shown that many trees in the Rashomon set have comparable performance in adversarial robustness and fairness to baseline models. Table 3 reports the percentage of trees within the Rashomon set that outperforms baselines. In certain datasets, more than 50% of trees in the Rashomon set achieve higher test adversarial accuracy than FPRDT and ROCT-V. Similarly, the Rashomon set contains trees that have greater fairness than all baselines, except for the adult and compas, where post-processing models and DPF can achieve higher fairness at the cost of reduced accuracy.

## 4.3 Q3. The Importance of Tree Sparsity for Trustworthy Properties

A larger Rashomon set can contain models with different complexity [74, 95]. In the case of sparse decision trees, we can measure complexity by the number of leaves, and indeed our computed Rashomon sets often contain trees with different numbers of leaves (for example, the Rashomon set of the german-credit dataset includes trees with leaves in the range from 1 to 13, Appendix C.2). Sparsity is important for trustworthy metrics, such as robustness or privacy, as it reduces the amount of information encoded in a model, limiting the risk of revealing sensitive data [56]. Our results also support this. Sparser Rashomon set trees (RSET_min) tend to perform better than more complex counterparts (RSET_max) for adversarial robustness (Figure 1 and Appendix A.1) and under the membership inference attack (Appendix B.1). Notably, we did not observe a connection between the sparsity of our models and fairness (Appendix C.2) and stability (Figure 1 and Appendix A.2).

## 4.4 Q4. TreeFARMS Training Time is Comparable to Optimal Baselines

TreeFARMS effectively optimizes the search space of trees, allowing us to find thousands or millions of near-optimal trees in seconds or minutes. Table 4 (and Appendices A.1, C.2) supports this with a summary of the training time compared to the robustness and fairness baselines that optimize for the best model averaged over datasets. Although TreeFARMS might require more time to run for deep depth, it finds significantly more trees (Appendix C.2).

Once the Rashomon set is constructed, model selection becomes a lightweight post-hoc step. Rather than retraining from scratch for each trustworthiness criterion, as required by optimization-based methods (e.g., robust or fair optimal trees), TreeFARMS allows users to evaluate and filter precomputed models according to desired constraints. Evaluating the entire set scales linearly with its size, is trivially parallelizable, and remains far cheaper than repeated retraining. In practice, enumerating and screening Rashomon sets with TreeFARMS is typically the more efficient and flexible option within an ML workflow.

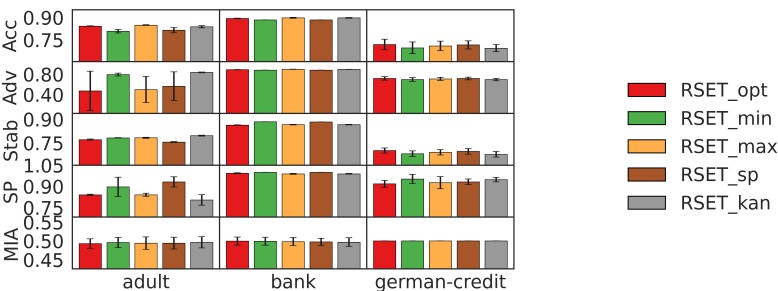

Figure 3: Evaluation of trees selected from the Rashomon set in Section 4.1 on different metrics.

## 4.5 Q5. Trees in the Rashomon Set Often Satisfy Multiple Trustworthy Properties

For different trustworthy metrics in Section 4.1, we selected trees that performed well based on the selection set. Here, in Figure 3, we further compare how these trees perform on other metrics. First of all, the accuracy of the trees is approximately the same since they all are from the same Rashomon set. All selected trees tend to be stable and perform well under membership inference attack, while the minimum complexity tree (RSET_min) is still preferred sometimes (e.g. bank). Similarly, trees selected specifically for statistical parity (e.g., german-credit and bank datasets) seem to be preferable to other models, especially optimal trees. Overall our results highlight the importance of considering multiple trustworthy metrics when selecting models, as no single tree consistently outperforms others across all criteria. Please see Appendix E for more datasets.

## 5 Conclusion, Limitations and Implications

We introduced a framework for evaluating trustworthy properties of models inside the Rashomon set of sparse decision trees. By benchmarking the Rashomon set against state-of-the-art tree baselines targeted to individual trustworthiness criteria and analyzing fairness, stability, robustness, and privacy, we provide a systematic methodology for understanding and intentionally navigating tradeoffs in high-stakes settings. Empirically, we find that Rashomon sets often contain models that are robust, stable, privacy-preserving, and fair even without explicit optimization for these properties. This reframes the Rashomon set as a resource: rather than retraining for every criterion, one can search within the set to identify models that meet desired constraints.

Our results suggest a simple selection protocol that mirrors the experiments: (i) enumerate or approximate the Rashomon set for a target loss tolerance and model class; (ii) evaluate trustworthiness metrics (fairness, stability, robustness, membership-inference privacy, and/or unlearning) for each model; (iii) filter by hard constraints (e.g., max disparity, min stability); (iv) choose model on the empirical Pareto frontier (e.g., fairness-accuracy or privacy-simplicity), or aggregate via a weighted objective when priorities are known. This procedure produces models that satisfy multiple criteria without retraining and exposes transparent tradeoffs when criteria conflict.

One limitation of our approach is that we focus on the hypothesis space of decision trees, though our methodology can be extended to other model classes (see Appendix D). Another limitation is that we evaluate membership inference using a limited number of membership inference attack algorithms for trees, as few such methods have been proposed in the literature. Nonetheless, the diversity of near-optimal trees within the Rashomon set already offers benefits: it can be leveraged for robustness via moving-target defenses, for fairness by covering different subpopulations, and for exploring explicit Pareto tradeoffs between properties such as privacy and simplicity.

Our findings motivate concrete questions for future work: When does model diversity most improve trustworthiness? Under what data/model conditions do multiple properties co-occur? How should we quantify diversity to predict trust gains? And to what extent do these effects transfer to richer model families? We hope that our evaluation frameworks can be used by data and machine learning scientists, as well as policymakers, to assess model performance across multiple trustworthiness criteria in a systematic and scalable way as well as inspire further research on model selection under larger Rashomon sets.

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

# Appendix

## Contents

**Compute Resources** All experiments were run on a Slurm cluster. Each job requested 256 GB of RAM and used Intel Xeon CPUs. We did not use GPUs for any experiments. For certain baselines, Gurobi was required and we used academic Gurobi licenses.

# A  Experiments: Robustness

Table 5: Summary of datasets used for Robustness Evaluation.

| Dataset | # Inst. | # Feat. | % Pos. |
|---|---|---|---|
| banknote [61] | 1372 | 4 | 44.5% |
| blood [96] | 748 | 4 | 23.8% |
| breast-w [91] | 683 | 9 | 35.0% |
| compas [51] | 6907 | 7 | 46.3% |
| diabetes [77] | 768 | 8 | 34.9% |
| fico [27] | 10459 | 23 | 52.2% |
| spambase [42] | 4601 | 57 | 39.4% |
| wine-quality [17] | 6497 | 11 | 63.3% |

## A.1  Adversarial Robustness

### A.1.1  Setup

We ran adversarial robustness experiments on 8 datasets listed in Table 5. Both TreeFARMS and ROCT-N solve NP-hard optimization problems, requiring preprocessing (binarization) for real-valued features. We applied GOSDT threshold guessing with n_estimator=30, max_depth=2, learning_rate=0.1, backselect=True [63]. The resulting tree is converted into a standard tree structure with appropriate features and thresholds by replacing binary splits with their corresponding threshold values. This step is necessary for adversarial attacks to generate perturbations.

We perform five-fold cross-validation with hyper-parameter tuning. For all greedy methods, we tune max depth of $[2, 3, 4]$ and regularization parameters $[0.005, 0.01, 0.015, 0.02]$ if they exist. For attack strength $\theta$, we set to $0.1$ for all datasets. To be more specific, we tune the following:

- **TreeFARMS**: For regularization, we tune TreeFARMS $\lambda = [0.005, 0.01, 0.015, 0.02]$. To select representative models for evaluation, we evaluate the optimal model (RSET_opt) and also consider the minimum- and maximum-leaf trees, choosing the highest-validation-accuracy tree among each to obtain RSET_min and RSET_max. We also use the validation set to generate adversarial examples on RSET_opt and select the best-performing model as RSET_kan.
- **GROOT**: For regularization, we tune min sample leaf to be $[0.005, 0.01, 0.015, 0.02] * n$ where $n$ is the number of samples when fitting the models. Similarly, min sample split would just be double of min sample leaf. For attack budget, we set it to $0.1$.
- **FPRDT**: Same as GROOT, we tune min sample leaf to be $[0.005, 0.01, 0.015, 0.02] * n$ where $n$ is the number of samples when fitting the models, and min sample split is double of min sample leaf. For attack budget, we set it to $0.1$.
- **ROCT-V**: We set max depth to 2, time limit to 1800, and attack budget to 0.1.
- **ROCT-N**: We set max depth to 2 and time limit to 1800. ROCT-N defines a robustness budget with a cost budget and confidence value. We set confidence value for each feature to be $90\%$, matching the 0.1 attack strength.

Note that both ROCT-V and ROCT-N are optimal algorithms with computationally expensive optimization processes, thus they are not tuned.

### A.1.2  Results

Table 6 shows the test accuracy of all models on 8 datasets we used in adversarial robustness experiments. Trees within the Rashomon set (e.g., RSET_opt and RSET_max) perform competitively and often achieve the highest test accuracy (shown in bold). In contrast, robust tree models such as GROOT, ROCT-N, and ROCT-V generally have lower accuracy as their robustness-focused optimization comes at the cost of predictive performance.

Table 7 shows accuracy on adversarial samples generated from the test dataset. The highest adversarial accuracy for each dataset is in bold. In our setup, the attack targets the optimal tree in the Rashomon

Table 6: Comparison of test accuracy across all methods. The highest accuracy for each dataset is bolded. Trees within the Rashomon set consistently achieve competitive performance.

| | CART | FPRDT | GROOT | ROCT-V | ROCT-N | RSET_opt | RSET_min | RSET_max | RSET_kan |
|---|---|---|---|---|---|---|---|---|---|
| banknote | $0.950 \pm 0.016$ | $0.908 \pm 0.009$ | $0.838 \pm 0.098$ | $0.908 \pm 0.009$ | $0.913 \pm 0.013$ | $0.953 \pm 0.029$ | $0.937 \pm 0.025$ | $0.955 \pm 0.029$ | $\mathbf{0.963 \pm 0.021}$ |
| blood | $\mathbf{0.795 \pm 0.011}$ | $0.762 \pm 0.004$ | $0.762 \pm 0.003$ | $0.766 \pm 0.004$ | $0.762 \pm 0.003$ | $0.770 \pm 0.013$ | $0.762 \pm 0.003$ | $0.765 \pm 0.054$ | $0.758 \pm 0.044$ |
| breast-w | $0.936 \pm 0.014$ | $0.946 \pm 0.013$ | $0.941 \pm 0.021$ | $0.950 \pm 0.012$ | $0.950 \pm 0.014$ | $\mathbf{0.952 \pm 0.012}$ | $0.934 \pm 0.032$ | $0.944 \pm 0.026$ | $0.941 \pm 0.021$ |
| compas | $0.666 \pm 0.014$ | $0.561 \pm 0.010$ | $0.558 \pm 0.009$ | $0.552 \pm 0.013$ | $0.653 \pm 0.015$ | $\mathbf{0.672 \pm 0.010}$ | $0.645 \pm 0.016$ | $\mathbf{0.672 \pm 0.015}$ | $0.664 \pm 0.020$ |
| diabetes | $\mathbf{0.749 \pm 0.017}$ | $0.647 \pm 0.008$ | $0.646 \pm 0.007$ | $0.650 \pm 0.005$ | $0.695 \pm 0.043$ | $0.738 \pm 0.026$ | $0.738 \pm 0.026$ | $0.728 \pm 0.040$ | $0.742 \pm 0.028$ |
| fico | $0.705 \pm 0.013$ | $0.540 \pm 0.011$ | $0.534 \pm 0.007$ | $0.522 \pm 0.000$ | $0.653 \pm 0.078$ | $0.708 \pm 0.009$ | $0.697 \pm 0.008$ | $\mathbf{0.709 \pm 0.015}$ | $0.684 \pm 0.014$ |
| spambase | $0.892 \pm 0.008$ | $0.734 \pm 0.027$ | $0.663 \pm 0.035$ | $0.648 \pm 0.028$ | $0.819 \pm 0.009$ | $0.900 \pm 0.004$ | $0.868 \pm 0.020$ | $\mathbf{0.905 \pm 0.005}$ | $0.895 \pm 0.009$ |
| wine-quality | $\mathbf{0.744 \pm 0.007}$ | $0.635 \pm 0.001$ | $0.633 \pm 0.000$ | $0.634 \pm 0.002$ | $0.633 \pm 0.000$ | $0.731 \pm 0.009$ | $0.730 \pm 0.006$ | $\mathbf{0.744 \pm 0.013}$ | $0.735 \pm 0.004$ |

Table 7: Comparison of test accuracy on adversarial samples. The highest accuracy for each dataset is in bold. The optimal tree RSET_opt in the Rashomon set is attacked and RSET_kan is the model within the Rashomon set selected from the selection set.

| | CART | FPRDT | GROOT | ROCT-V | ROCT-N | RSET_opt | RSET_min | RSET_max | RSET_kan |
|---|---|---|---|---|---|---|---|---|---|
| banknote | $0.544 \pm 0.050$ | $0.665 \pm 0.023$ | $0.642 \pm 0.055$ | $0.665 \pm 0.023$ | $0.612 \pm 0.071$ | $0.598 \pm 0.033$ | $0.785 \pm 0.110$ | $0.814 \pm 0.088$ | $\mathbf{0.938 \pm 0.016}$ |
| blood | $0.385 \pm 0.041$ | $0.759 \pm 0.004$ | $\mathbf{0.762 \pm 0.003}$ | $0.761 \pm 0.007$ | $\mathbf{0.762 \pm 0.003}$ | $0.544 \pm 0.173$ | $\mathbf{0.762 \pm 0.003}$ | $0.661 \pm 0.143$ | $0.739 \pm 0.035$ |
| breast-w | $0.833 \pm 0.027$ | $0.931 \pm 0.012$ | $0.930 \pm 0.018$ | $0.937 \pm 0.017$ | $0.900 \pm 0.017$ | $0.893 \pm 0.043$ | $0.912 \pm 0.041$ | $0.930 \pm 0.036$ | $\mathbf{0.940 \pm 0.020}$ |
| compas | $0.184 \pm 0.030$ | $0.543 \pm 0.010$ | $0.558 \pm 0.009$ | $0.326 \pm 0.209$ | $0.193 \pm 0.154$ | $0.195 \pm 0.015$ | $0.583 \pm 0.019$ | $0.485 \pm 0.103$ | $\mathbf{0.667 \pm 0.022}$ |
| diabetes | $0.477 \pm 0.151$ | $0.645 \pm 0.013$ | $0.642 \pm 0.015$ | $0.637 \pm 0.015$ | $0.540 \pm 0.126$ | $0.439 \pm 0.098$ | $0.719 \pm 0.026$ | $0.651 \pm 0.066$ | $\mathbf{0.728 \pm 0.031}$ |
| fico | $0.224 \pm 0.024$ | $0.525 \pm 0.005$ | $0.532 \pm 0.007$ | $0.522 \pm 0.000$ | $0.394 \pm 0.083$ | $0.263 \pm 0.010$ | $0.561 \pm 0.082$ | $0.500 \pm 0.028$ | $\mathbf{0.680 \pm 0.010}$ |
| spambase | $0.000 \pm 0.000$ | $0.653 \pm 0.015$ | $0.631 \pm 0.015$ | $0.616 \pm 0.010$ | $0.031 \pm 0.040$ | $0.083 \pm 0.106$ | $0.634 \pm 0.116$ | $0.490 \pm 0.091$ | $\mathbf{0.894 \pm 0.007}$ |
| wine-q | $0.198 \pm 0.018$ | $0.632 \pm 0.002$ | $0.633 \pm 0.000$ | $0.633 \pm 0.000$ | $0.633 \pm 0.000$ | $0.355 \pm 0.030$ | $0.487 \pm 0.165$ | $0.647 \pm 0.034$ | $\mathbf{0.719 \pm 0.032}$ |

set, so it often has lower adversarial accuracy. However, many other trees within the Rashomon set remain robust. RSET_kan achieves the highest accuracy on most datasets, indicating that the selected trees can provide both robustness and accurate predictions.

Table 8: Adversarial accuracy of different Rashomon set trees when the attack targets a randomly selected tree instead of the optimal tree.

| Dataset | RSET_opt | RSET_min | RSET_max | RSET_kan |
|---|---|---|---|---|
| banknote | $0.772 \pm 0.066$ | $0.767 \pm 0.076$ | $0.794 \pm 0.08$ | $\mathbf{0.812 \pm 0.046}$ |
| blood | $0.71 \pm 0.089$ | $\mathbf{0.762 \pm 0.003}$ | $0.485 \pm 0.076$ | $0.639 \pm 0.085$ |
| breast-w | $0.726 \pm 0.266$ | $0.918 \pm 0.027$ | $\mathbf{0.946 \pm 0.018}$ | $0.911 \pm 0.034$ |
| compas | $\mathbf{0.529 \pm 0.081}$ | $0.389 \pm 0.152$ | $0.503 \pm 0.052$ | $0.472 \pm 0.087$ |
| diabetes | $0.71 \pm 0.037$ | $\mathbf{0.716 \pm 0.054}$ | $0.693 \pm 0.063$ | $0.706 \pm 0.061$ |
| fico | $0.564 \pm 0.054$ | $0.547 \pm 0.044$ | $0.576 \pm 0.079$ | $\mathbf{0.64 \pm 0.043}$ |
| spambase | $0.764 \pm 0.119$ | $\mathbf{0.853 \pm 0.013}$ | $0.708 \pm 0.112$ | $0.665 \pm 0.121$ |
| wine-quality | $0.604 \pm 0.059$ | $\mathbf{0.609 \pm 0.056}$ | $0.583 \pm 0.078$ | $0.57 \pm 0.072$ |

We also investigate how different representative trees within the Rashomon set perform when the attack targets a randomly selected tree rather than the optimal tree. Table 8 shows that attacking different random trees from the Rashomon set leads to different robustness performances of representative trees. In some cases, RSET_kan, the model chosen from the selection set, achieves the highest accuracy. However, in most cases, RSET_min, the tree with the fewest leaves, outperforms the others. This finding suggests that sparser trees tend to be more robust against adversarial attacks compared to their more complex counterparts, such as RSET_max, as we discussed in Section 4.3.

Figure 4 compares the adversarial accuracy of trees within the Rashomon set to various baseline models on the test set. This figure expands on the discussion in Section 4.1. The light blue density plot represents the distribution of adversarial accuracy for all trees in the Rashomon set or a random subsample of 100,000 trees of the Rashomon set, whichever is smaller. Vertical lines indicate the accuracy of baseline models, including CART, FPRDT, GROOT, ROCT-V, and ROCT-N, as well as specific Rashomon set trees such as the optimal tree (RSET_opt), the sparsest tree (RSET_min), the most complex tree (RSET_max), and RSET_kan, a tree selected from the held-out validation set.

As we can see, many trees within the Rashomon set achieve higher accuracy than the baselines, indicating their robustness against adversarial attacks. This suggests that careful selection within the Rashomon set can yield robust models for critical tasks facing adversarial attacks, as attackers cannot possibly generate 100,000 distinct adversarial examples. Additionally, these density plots

Table 9: Percentage of RSET trees outperforming baseline models for Test Adv Accuracy (mean ± std across folds)

| dataset | ALL | FPRDT | ROCT-V |
|---|---|---|---|
| banknote | 95.5 ± 6.0% | 95.5 ± 6.0% | 95.5 ± 6.0% |
| blood | 26.9 ± 40.4% | 28.5 ± 39.6% | 26.9 ± 40.4% |
| breast-w | 54.9 ± 35.9% | 63.0 ± 38.8% | 62.8 ± 29.8% |
| compas | 27.4 ± 8.2% | 27.4 ± 8.1% | 68.4 ± 39.6% |
| diabetes | 50.3 ± 30.2% | 51.6 ± 29.8% | 53.9 ± 27.4% |
| fico | 59.2 ± 12.6% | 59.5 ± 12.1% | 60.5 ± 13.8% |
| spambase | 39.3 ± 18.9% | 39.3 ± 18.9% | 45.1 ± 18.7% |
| wine-quality | 45.4 ± 15.5% | 46.2 ± 15.9% | 45.5 ± 15.6% |

highlight areas for improving our selection metric. For instance, while the selected Rashomon set tree performs best on certain datasets, it does not always achieve the best performance in others (e.g., breast cancer). This finding might inspire future research on developing better selection criteria.

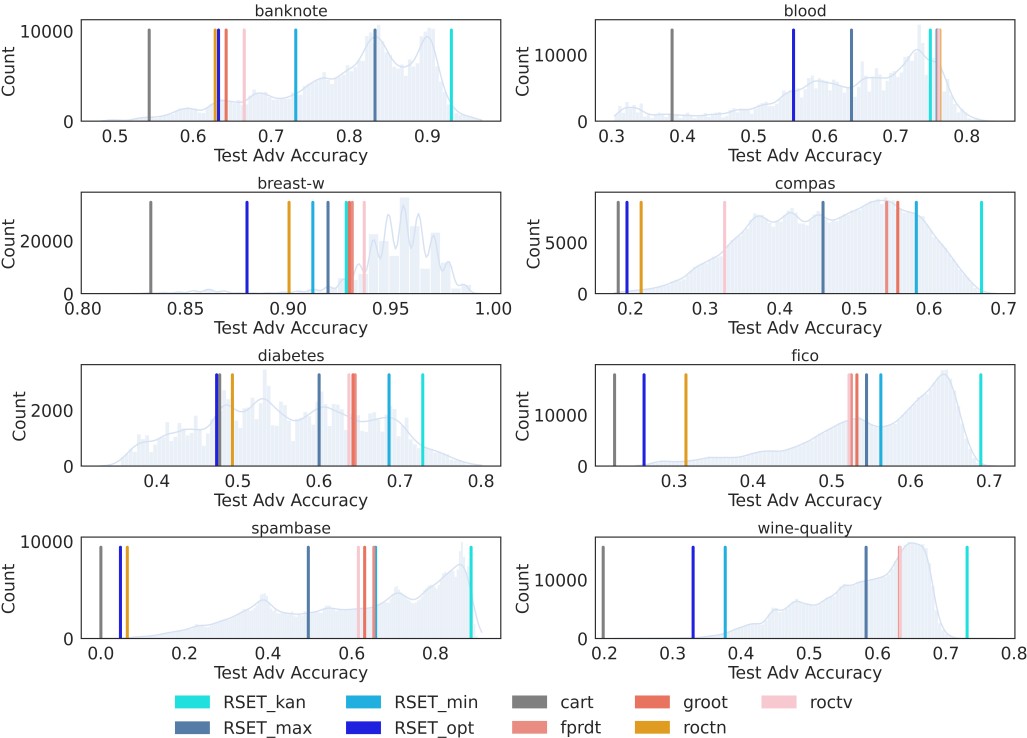

Figure 4: Comparison of adversarial accuracy between trees in the Rashomon set (light blue histogram) and baseline models (vertical lines) on the test set. Most Rashomon set trees achieve higher robustness than baselines.

Table 9 extends Table 3 in Section 4.2, reporting the percentage of trees within the Rashomon set that outperform the baseline models on the test set. Across all datasets, the Rashomon set includes trees that achieve higher adversarial accuracy than the baselines. In certain cases, such as banknote and fico datasets, nearly 80% of trees in the Rashomon set achieve higher test adversarial accuracy than FPRDT and ROCT-V.

## A.2 Stability

### A.2.1 Setup

To ensure stability, we evaluate our models on the same 8 datasets used in the adversarial robustness experiments (Table 5). We consider five different noise perturbations, briefly introduced in Section

Table 10: Training Time for RSET and Optimal Robust Tree Baselines.

| | ROCT-V | ROCT-N | RSET |
|---|---|---|---|
| banknote | $1800.781 \pm 0.035$ | $1810.124 \pm 6.608$ | $\mathbf{17.895 \pm 12.102}$ |
| blood | $977.194 \pm 295.281$ | $1809.429 \pm 3.032$ | $\mathbf{2.782 \pm 2.876}$ |
| breast-w | $352.586 \pm 76.484$ | $1810.625 \pm 11.337$ | $\mathbf{16.391 \pm 21.468}$ |
| compas | $1802.747 \pm 0.247$ | $1840.383 \pm 15.924$ | $\mathbf{29.072 \pm 15.638}$ |
| diabetes | $1800.526 \pm 0.042$ | $1817.740 \pm 11.857$ | $\mathbf{68.044 \pm 69.641}$ |
| fico | $1806.438 \pm 0.474$ | $1863.387 \pm 37.000$ | $\mathbf{481.208 \pm 280.406}$ |
| spambase | $1861.620 \pm 77.444$ | $1842.396 \pm 21.302$ | $\mathbf{365.972 \pm 255.280}$ |
| wine-quality | $1803.188 \pm 0.334$ | $1830.481 \pm 24.930$ | $\mathbf{134.246 \pm 123.406}$ |

3.2. Additionally, threshold guessing is applied to all datasets to generate a binary dataset. Same hyperparameter tuning as the adversarial robustness section is applied.

For baselines that require an attack strength specification, we set their budget to 10% by default. This choice is justified by the expected perturbation value for noise, which is approximately 11%. This value is derived from the expectation of a geometric distribution, $\frac{1}{q_t^j}$, where the expected value of $q_t^j$ corresponds to the mean of our normal distribution, which is 0.9. Thus, using a 10% budget ensures consistency with the setup used in other sections.

For each perturbation, we conduct 5000 repeated trials, resampling the noise in each iteration while maintaining the fixed confidence level. We compute the average and worst-case scores and record the standard deviation of the results. This process is repeated across five folds.

### A.2.2 Results

Figure 5 and Figure 6 visualize the stability performance of trees across 8 datasets under five different types of noise perturbations. Note that only ROCT-N explicitly accounts for these perturbations during training. As shown in the figures, different representative trees from the Rashomon set (bars colored in blue palette) are generally comparable to other baselines. This indicates that while the Rashomon set is not explicitly designed for stability, it contains trees that perform well under perturbations. Interestingly, CART also performs well for stability. We believe that the similar stability performance may be due to intrinsic tree properties.

**Training Time:** Time consumption is an important metric to consider when evaluating model performance. We investigate whether TreeFARMS' consistently strong results come at the cost of significantly longer training times. Table 10 reports the training time (in seconds) for each method in the adversarial robustness setting. Greedy methods are not included as CART, FPRDT, and GROOT complete training quickly due to their heuristic-based construction. In contrast, ROCT-N, ROCT-V, and TreeFARMS aim to find globally optimal solutions, which are NP-hard problems. As a result, ROCT-N and ROCT-V often reach or nearly approach the 1,800-second time limit. TreeFARMS (RSET) usually completes training within a reasonable time frame. Note that the table shows training latency – for example, ROCT-N continues processing beyond the time limit before terminating.

## B Experiments: Privacy

### B.1 Membership Inference Attack

#### B.1.1 Setup

We evaluate the membership inference attack on seven datasets, see Table 11 for details. As before, we apply GOSDT threshold guessing to binarize continuous dataset using `n_estimator=30`, `max_depth=2`, and `learning_rate=0.1` with back select.

We perform five-fold cross-validation with hyper-parameter tuning. For all methods, we tune max depth of $[2, 3, 4]$ and regularization parameters $[0.005, 0.01, 0.015, 0.02]$. To be more specific, we tune the following:

- **TreeFARMS**: For regularization, we tune TreeFARMS $\lambda = [0.005, 0.01, 0.015, 0.02]$. To select representative models for evaluation, we evaluate the optimal model (RSET_opt) and also consider

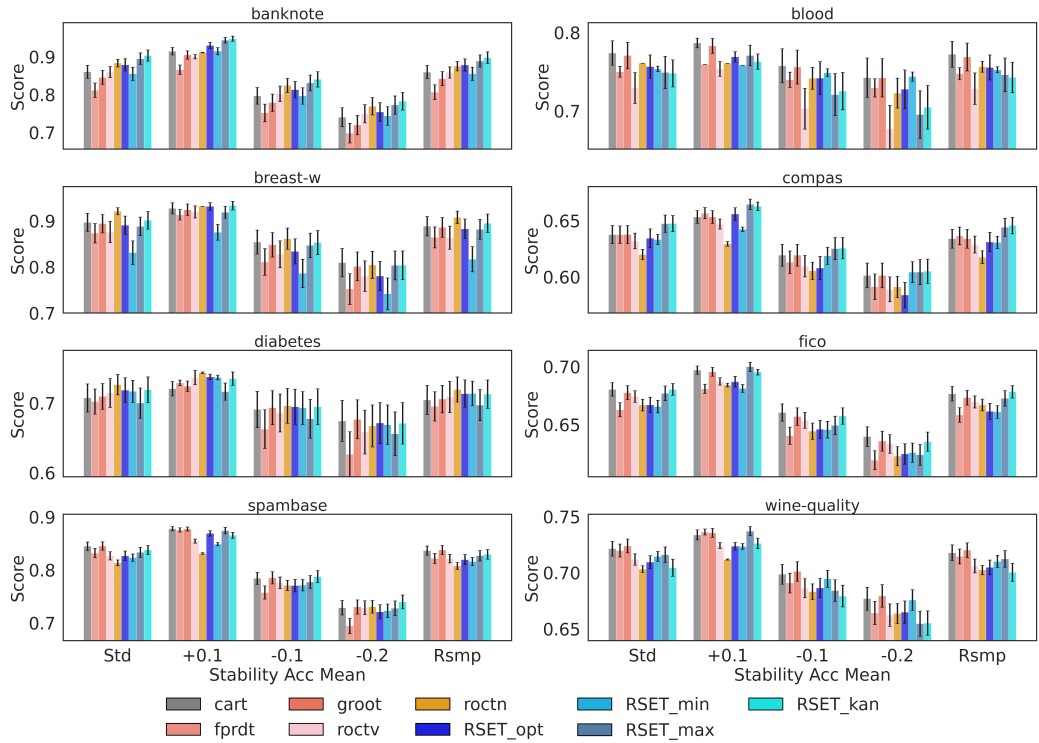

Figure 5: Comparison of average stability accuracy across different methods under 5 perturbation types. Error bars represent the average standard deviation across 5000 trials over five folds.

Table 11: Summary of datasets used for Membership Inference Attack Evaluation.

| Dataset | # Inst. | # Feat. | % Pos. |
|---|---|---|---|
| adult [23] | 45222 | 18 | 24.8% |
| bank [66] | 45211 | 47 | 11.7% |
| cal-h [69] | 20634 | 8 | 50.0% |
| compas [51] | 6172 | 10 | 54.5% |
| credit-fusion [31] | 16714 | 10 | 50.0% |
| fico [27] | 10459 | 23 | 52.2% |
| oulad [50] | 21562 | 46 | 68.0% |

the minimum- and maximum-leaf trees, choosing the highest-validation-accuracy tree among each to obtain RSET_min and RSET_max.

- **PRIVA**: For regularization, we tune min sample leaf to be $[0.005, 0.01, 0.015, 0.02] * n$ where $n$ is the number of samples when fitting the models. Similarly, min sample split would just be double of min sample leaf. For privacy threshold $\eta_1$, we set it to $0.1$.

- **BDPT**: We tune min sample split as $[0.005, 0.01, 0.015, 0.02] * 2n$. We further set the privacy threshold $\eta_1$ to $0.1$.

- **DPLDT**: We set the privacy $\eta_1$ threshold to $0.1$. DPLDT does not have a regularization parameter.

Both DPLDT and BDPT were designed as base estimators for random forest, but we evaluate whether a single estimator can provide sufficient privacy against membership inference attacks.

### B.1.2 Results

Table 12 presents the standard test accuracy of different methods, including greedy tree CART, differentially private trees BDPT, DPLDT, and PRIVA, and representative trees within the Rashomon set. The highest accuracy for each dataset is bolded. The results show that trees within the Rashomon set (especially RSET_max) usually achieve higher test accuracy compared to all baselines. Differentially private trees often underperform. This is likely due to their reliance on randomized splitting or noise injection during training, which introduces additional variability and reduces accuracy.

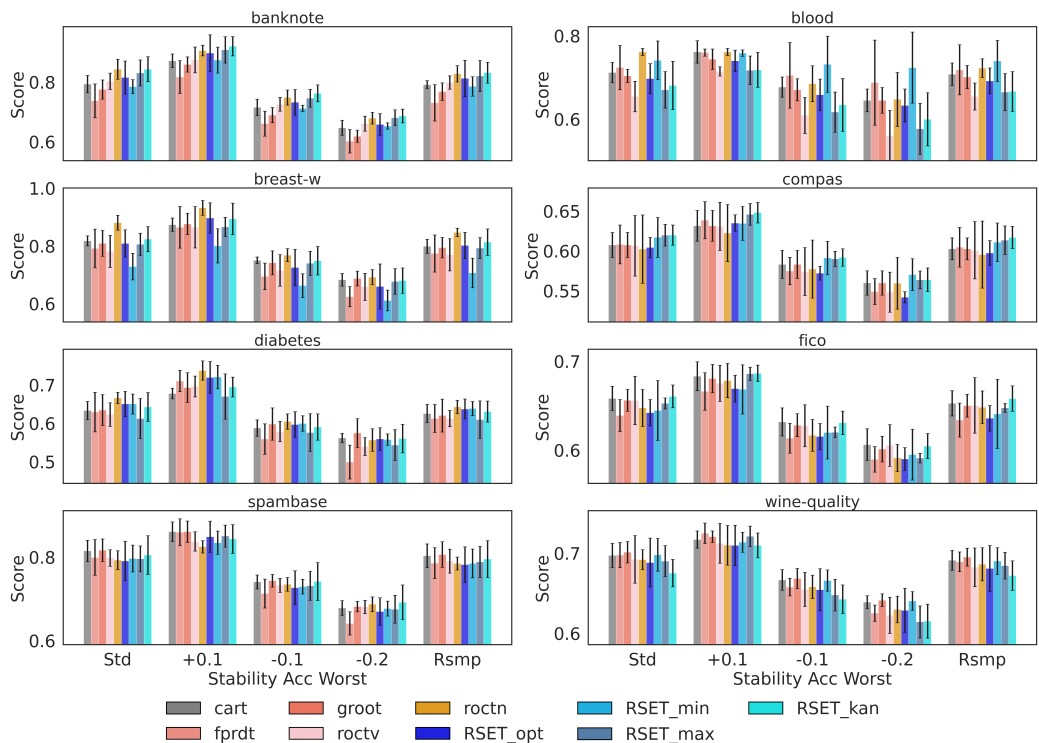

Figure 6: Comparison of worst-case stability accuracy across different methods under 5 perturbation types. Error bars represent the standard deviation across five folds.

Table 12: Comparison of test accuracy between differentially private tree models and trees within the Rashomon set.

|  | CART | BDPT | DPLDT | PRIVA | RSET_opt | RSET_min | RSET_max |
|---|---|---|---|---|---|---|---|
| adult | $0.832 \pm 0.009$ | $0.752 \pm 0.000$ | $0.752 \pm 0.000$ | $0.789 \pm 0.028$ | $0.837 \pm 0.005$ | $0.794 \pm 0.002$ | $\mathbf{0.838 \pm 0.005}$ |
| bank | $0.894 \pm 0.003$ | $0.883 \pm 0.000$ | $0.883 \pm 0.000$ | $0.884 \pm 0.007$ | $0.893 \pm 0.002$ | $0.883 \pm 0.000$ | $\mathbf{0.897 \pm 0.003}$ |
| cal-h | $0.780 \pm 0.004$ | $0.500 \pm 0.000$ | $0.763 \pm 0.013$ | $0.716 \pm 0.029$ | $0.816 \pm 0.010$ | $0.769 \pm 0.013$ | $\mathbf{0.825 \pm 0.013}$ |
| compas | $0.663 \pm 0.007$ | $0.545 \pm 0.000$ | $0.641 \pm 0.011$ | $0.606 \pm 0.040$ | $0.663 \pm 0.006$ | $0.643 \pm 0.017$ | $\mathbf{0.666 \pm 0.005}$ |
| credit | $0.750 \pm 0.005$ | $0.500 \pm 0.000$ | $0.722 \pm 0.006$ | $0.701 \pm 0.028$ | $\mathbf{0.762 \pm 0.006}$ | $0.729 \pm 0.014$ | $0.760 \pm 0.007$ |
| fico | $0.711 \pm 0.007$ | $0.522 \pm 0.000$ | $0.687 \pm 0.013$ | $0.675 \pm 0.030$ | $0.706 \pm 0.006$ | $0.699 \pm 0.005$ | $\mathbf{0.714 \pm 0.010}$ |
| oulad | $0.681 \pm 0.002$ | $0.673 \pm 0.000$ | $0.673 \pm 0.000$ | $0.672 \pm 0.001$ | $0.673 \pm 0.000$ | $0.673 \pm 0.000$ | $\mathbf{0.686 \pm 0.004}$ |

Tables 13 - 16 report the accuracy of four different attacks, ordered from weakest to strongest: baseline attack, label-only inference attacks, label-sup inference attack, and shadow model attack. A lower value indicates that the method is more resistant to attacks.

Table 13: Rule-based MIA success rates for different methods. Lower values imply better privacy.

|  | CART | BDPT | DPLDT | PRIVA | RSET_opt | RSET_min | RSET_max |
|---|---|---|---|---|---|---|---|
| adult | $0.506 \pm 0.014$ | $0.513 \pm 0.012$ | $0.513 \pm 0.012$ | $0.505 \pm 0.018$ | $0.502 \pm 0.012$ | $0.509 \pm 0.017$ | $\mathbf{0.502 \pm 0.011}$ |
| bank | $0.499 \pm 0.004$ | $0.499 \pm 0.008$ | $\mathbf{0.498 \pm 0.007}$ | $0.499 \pm 0.011$ | $0.511 \pm 0.019$ | $0.508 \pm 0.015$ | $0.510 \pm 0.015$ |
| cal-h | $0.505 \pm 0.016$ | $0.510 \pm 0.014$ | $0.504 \pm 0.013$ | $0.499 \pm 0.011$ | $0.502 \pm 0.009$ | $\mathbf{0.499 \pm 0.006}$ | $0.504 \pm 0.009$ |
| compas | $0.517 \pm 0.006$ | $0.507 \pm 0.006$ | $0.514 \pm 0.010$ | $0.515 \pm 0.011$ | $0.500 \pm 0.010$ | $\mathbf{0.494 \pm 0.016}$ | $0.501 \pm 0.014$ |
| credit | $0.514 \pm 0.008$ | $\mathbf{0.496 \pm 0.006}$ | $0.509 \pm 0.012$ | $0.510 \pm 0.013$ | $0.503 \pm 0.009$ | $0.505 \pm 0.017$ | $0.502 \pm 0.014$ |
| fico | $0.489 \pm 0.009$ | $\mathbf{0.483 \pm 0.008}$ | $0.486 \pm 0.011$ | $0.490 \pm 0.014$ | $0.504 \pm 0.017$ | $0.502 \pm 0.012$ | $0.506 \pm 0.018$ |
| oulad | $0.500 \pm 0.006$ | $\mathbf{0.496 \pm 0.006}$ | $\mathbf{0.496 \pm 0.006}$ | $\mathbf{0.496 \pm 0.006}$ | $0.502 \pm 0.019$ | $0.502 \pm 0.019$ | $0.498 \pm 0.018$ |

Overall, these attacks are not successful as the highest observed accuracy remains close to 0.5 in all four tables, meaning the attacker does not significantly outperform random guessing. Compared to 3 differential private tree methods, trees from the Rashomon set with the fewest leaves (RSET_min) achieve comparable or even better resistance against both attacks, despite the fact that the TreeFARMS algorithm does not incorporate any explicit randomness for privacy protection.

Table 14: Label-only unsupervised MIA success rates for different methods. Lower values imply better privacy. Attack failed when no effective perturbation was found.

|        | CART            | BDPT        | DPLDT           | PRIVA           | RSET_opt        | RSET_min        | RSET_max        |
|--------|-----------------|-------------|-----------------|-----------------|-----------------|-----------------|-----------------|
| adult  | $0.506 \pm 0.014$ | **Attk Failed** | $0.513 \pm 0.012$ | $0.498 \pm 0.010$ | $0.502 \pm 0.012$ | $0.509 \pm 0.017$ | $0.502 \pm 0.011$ |
| bank   | $0.499 \pm 0.004$ | **Attk Failed** | **Attk Failed** | $0.503 \pm 0.011$ | $0.511 \pm 0.019$ | **Attk Failed** | $0.510 \pm 0.015$ |
| cal-h  | $0.505 \pm 0.016$ | **Attk Failed** | $0.504 \pm 0.011$ | $0.497 \pm 0.013$ | $0.502 \pm 0.009$ | $0.499 \pm 0.006$ | $0.504 \pm 0.009$ |
| compas | $0.517 \pm 0.006$ | **Attk Failed** | $0.517 \pm 0.010$ | $0.513 \pm 0.012$ | $0.500 \pm 0.010$ | $0.494 \pm 0.016$ | $0.501 \pm 0.014$ |
| credit | $0.514 \pm 0.008$ | **Attk Failed** | $0.504 \pm 0.010$ | $0.507 \pm 0.015$ | $0.503 \pm 0.009$ | $0.505 \pm 0.017$ | $0.502 \pm 0.014$ |
| fico   | $0.489 \pm 0.009$ | **Attk Failed** | $0.501 \pm 0.012$ | $0.500 \pm 0.009$ | $0.504 \pm 0.017$ | $0.502 \pm 0.012$ | $0.506 \pm 0.018$ |
| oulad  | $0.500 \pm 0.006$ | **Attk Failed** | **Attk Failed** | **Attk Failed** | **Attk Failed** | **Attk Failed** | $0.498 \pm 0.018$ |

Table 15: Label-only supervised MIA success rates for different methods. Lower values imply better privacy.

|        | CART            | BDPT            | DPLDT               | PRIVA               | RSET_opt        | RSET_min            | RSET_max            |
|--------|-----------------|-----------------|---------------------|---------------------|-----------------|---------------------|---------------------|
| adult  | $0.505 \pm 0.009$ | $0.500 \pm 0.000$ | $0.512 \pm 0.020$     | **0.494 $\pm$ 0.011**   | $0.503 \pm 0.007$ | $0.509 \pm 0.017$     | $0.503 \pm 0.005$     |
| bank   | $0.501 \pm 0.007$ | $0.500 \pm 0.000$ | $0.499 \pm 0.007$     | **0.494 $\pm$ 0.006**   | $0.511 \pm 0.019$ | $0.500 \pm 0.000$     | $0.507 \pm 0.012$     |
| cal-h  | $0.505 \pm 0.013$ | $0.500 \pm 0.000$ | **0.497 $\pm$ 0.012** | $0.504 \pm 0.010$     | $0.499 \pm 0.008$ | **0.496 $\pm$ 0.004** | $0.503 \pm 0.004$     |
| compas | $0.504 \pm 0.009$ | $0.500 \pm 0.000$ | $0.516 \pm 0.013$     | $0.502 \pm 0.008$     | $0.497 \pm 0.006$ | **0.492 $\pm$ 0.007** | $0.494 \pm 0.012$     |
| credit | $0.505 \pm 0.009$ | $0.500 \pm 0.000$ | $0.502 \pm 0.007$     | $0.509 \pm 0.018$     | $0.500 \pm 0.007$ | $0.502 \pm 0.017$     | **0.498 $\pm$ 0.011** |
| fico   | $0.500 \pm 0.011$ | $0.500 \pm 0.000$ | **0.486 $\pm$ 0.005** | $0.491 \pm 0.014$     | $0.512 \pm 0.007$ | $0.502 \pm 0.012$     | $0.508 \pm 0.008$     |
| oulad  | $0.499 \pm 0.006$ | $0.500 \pm 0.000$ | $0.500 \pm 0.000$     | $0.503 \pm 0.004$     | $0.500 \pm 0.000$ | $0.500 \pm 0.000$     | **0.492 $\pm$ 0.014** |

## B.2   Machine Unlearning

### B.2.1   Setup

We evaluate TreeFARMS, DaRE, and GBDT unlearning on 5 datasets: adult, bank, carryout, compas, and restaurant. Note that adult, compas, and bank datasets are binarized, while carryout and restaurant are real-valued. Details are shown in Table 17. The datasets are split into training and test sets using an 80-20 split. All methods are fitted on the training set after hyperparameters have been tuned based on the configurations described:

- **TreeFARMS**: We tune depth=[2,3,4] and $\lambda = [0.01, 0.005, 0.001]$.
- **DaRE**: We first fix the number of random layers to 0 and tune G-DaRE, considering the maximum tree depth [1,3,5,10,20], the number of trees [10,25,50,100,250], and the number of threshold values per attribute [5,10,25,50]. After identifying the best configuration for G-DaRE, we tune the number of random layers from 1 to 10, stopping when the cross-validation score exceeds a 0.5% tolerance compared to the greedy model for R-DaRE.
- **GBDT unlearning**: Similar to DaRE, we first ignore random layers and tune the maximum number of leaves [5,10,15,20] and feature sampling rate [0.05, 0.1, 0.5, 1] for G-Boosting. We then tune the number of random layers from 1 to 4.

Note that constructing the Rashomon set is NP-hard, so we apply GOSDT threshold guessing with n_estimator=30, max_depth=2, learning_rate=0.1, backselect=True [63] when fitting TreeFARMS on carryout and restaurant datasets.

We randomly remove 0.5%, 1%, and 2% of training samples 10 times and compare the results of unlearning with those of retraining for each method. Note that we only need to construct one Rashomon set from TreeFARMS by setting $\epsilon = 0.04$.

Table 16: Shadow MIA success rates for different methods. Lower values imply better privacy.

|        | CART                | BDPT                | DPLDT           | PRIVA           | RSET_opt            | RSET_min            | RSET_max        |
|--------|---------------------|---------------------|-----------------|-----------------|---------------------|---------------------|-----------------|
| adult  | **0.492 $\pm$ 0.012** | $0.492 \pm 0.017$     | $0.492 \pm 0.017$ | $0.496 \pm 0.013$ | $0.500 \pm 0.018$     | $0.502 \pm 0.022$     | $0.501 \pm 0.020$ |
| bank   | **0.490 $\pm$ 0.004** | $0.496 \pm 0.006$     | $0.498 \pm 0.004$ | $0.495 \pm 0.005$ | $0.505 \pm 0.009$     | $0.508 \pm 0.014$     | $0.509 \pm 0.015$ |
| cal-h  | $0.499 \pm 0.019$     | **0.489 $\pm$ 0.011** | $0.491 \pm 0.018$ | $0.495 \pm 0.013$ | $0.503 \pm 0.020$     | $0.491 \pm 0.020$     | $0.504 \pm 0.021$ |
| compas | $0.500 \pm 0.017$     | **0.498 $\pm$ 0.010** | $0.506 \pm 0.020$ | $0.501 \pm 0.011$ | $0.504 \pm 0.020$     | $0.506 \pm 0.016$     | $0.508 \pm 0.015$ |
| credit | $0.504 \pm 0.019$     | $0.506 \pm 0.003$     | $0.506 \pm 0.015$ | $0.502 \pm 0.009$ | $0.501 \pm 0.005$     | **0.500 $\pm$ 0.006** | $0.503 \pm 0.004$ |
| fico   | **0.489 $\pm$ 0.010** | $0.496 \pm 0.020$     | $0.493 \pm 0.011$ | $0.490 \pm 0.011$ | $0.498 \pm 0.020$     | $0.509 \pm 0.015$     | $0.492 \pm 0.017$ |
| oulad  | $0.508 \pm 0.022$     | $0.503 \pm 0.007$     | $0.502 \pm 0.007$ | $0.503 \pm 0.007$ | **0.490 $\pm$ 0.016** | **0.490 $\pm$ 0.016** | $0.492 \pm 0.014$ |

Table 17: Summary of datasets used for Machine Unlearning Evaluation.

| Dataset | # Inst. | # Feat. | % Pos. |
|---|---|---|---|
| adult [23] | 45222 | 18 | 24.8% |
| bank [66] | 45211 | 47 | 11.7% |
| carryout [86] | 2280 | 22 | 73.77% |
| compas [51] | 6172 | 10 | 54.5% |
| restaurant [86] | 2653 | 22 | 70.90% |

### B.2.2 Results

Table 18 is a complete version of Table 2. It shows the proportion of test data with different predicted labels between the unlearned and retrained models for the Rashomon set, G-DaRE, R-DaRE, G-Boosting, and R-Boosting. Since the Rashomon set has a theoretical guarantee that the optimal tree for the reduced dataset remains within the set after a subset of the training data is removed, the RSET column always reports 0. However, other methods do not have this guarantee.

Table 18: Proportion of test data with different predicted labels between the unlearned and retrained models (i.e., $\hat{y}_{\text{unlearn}} \neq \hat{y}_{\text{retrain}}$) after a subset of the original data is randomly removed. The Rashomon set achieves the lowest mismatch loss, as it is guaranteed to contain the optimal tree trained after a certain proportion of samples are removed.

| Dataset | unlearn size | RSET | G-DARE | R-DARE | G-Boosting | R-Boosting |
|---|---|---|---|---|---|---|
| carryout | 0.5% | 0% | 1.272% ± 0.611% | 1.601% ± 0.529% | 5.680% ± 0.385% | 2.961% ± 0.660% |
| | 1% | 0% | 1.294% ± 0.465% | 1.601% ± 0.510% | 5.548% ± 0.741% | 3.224% ± 0.844% |
| | 2% | 0% | 1.469% ± 0.501% | 1.689% ± 0.393% | 4.978% ± 1.334% | 3.662% ± 1.233% |
| restaurant | 0.5% | 0% | 2.015% ± 0.267% | 1.620% ± 0.388% | 4.708% ± 0.834% | 4.087% ± 0.513% |
| | 1% | 0% | 1.902% ± 0.433% | 1.808% ± 0.493% | 4.746% ± 0.623% | 3.691% ± 0.847% |
| | 2% | 0% | 2.147% ± 0.349% | 1.846% ± 0.582% | 4.501% ± 0.755% | 3.974% ± 0.865% |
| adult | 0.5% | 0% | 0.008% ± 0.007% | 0.066% ± 0.037% | 0.052% ± 0.019% | 0.170% ± 0.170% |
| | 1% | 0% | 0.018% ± 0.013% | 0.067% ± 0.047% | 0.064% ± 0.027% | 0.175% ± 0.165% |
| | 2% | 0% | 0.010% ± 0.014% | 0.060% ± 0.036% | 0.051% ± 0.024% | 0.238% ± 0.166% |
| compas | 0.5% | 0% | 0.000% ± 0.000% | 0.000% ± 0.000% | 0.470% ± 0.198% | 1.741% ± 1.599% |
| | 1% | 0% | 0.000% ± 0.000% | 0.000% ± 0.000% | 0.478% ± 0.543% | 1.895% ± 1.422% |
| | 2% | 0% | 0.000% ± 0.000% | 0.000% ± 0.000% | 0.810% ± 0.871% | 1.879% ± 1.608% |
| bank | 0.5% | 0% | 0.587% ± 0.117% | 0.201% ± 0.040% | 0.911% ± 0.099% | 1.081% ± 0.157% |
| | 1% | 0% | 0.607% ± 0.103% | 0.248% ± 0.033% | 0.888% ± 0.096% | 1.006% ± 0.105% |
| | 2% | 0% | 0.631% ± 0.077% | 0.289% ± 0.055% | 0.850% ± 0.139% | 1.066% ± 0.124% |

Another important metric in machine unlearning experiments is time consumption. Table 19 compares the unlearning time and retraining time for each method after different proportions of the training data are removed in seconds. In general, unlearning is faster than retraining for all methods, and the timing is fairly comparable across different methods. Additionally, removing fewer samples results in faster unlearning times. In some cases, retraining may be faster than unlearning for the Rashomon set. This is because unlearning in the Rashomon set is actually a search process. When the Rashomon set is large, and the proportion of removed data is high, more models within the set may need to be evaluated to identify the optimal one, leading to a longer learning time.

Table 19: Comparison of unlearning time and retraining time (in seconds). Unlearning is usually faster than retraining across all methods.

| Dataset | unlearn size | RSET | | G-DARE | | R-DARE | | G-Boosting | | R-Boosting | |
|---|---|---|---|---|---|---|---|---|---|---|---|
| | | unlearn time | retrain time | unlearn time | retrain time | unlearn time | retrain time | unlearn time | retrain time | unlearn time | retrain time |
| carryout | 0.5% | $0.093 \pm 0.029$ | $0.087 \pm 0.001$ | $0.187 \pm 0.029$ | $0.784 \pm 0.01$ | $0.133 \pm 0.016$ | $0.738 \pm 0.013$ | $0.679 \pm 0.066$ | $0.548 \pm 0.004$ | $0.605 \pm 0.054$ | $0.533 \pm 0.007$ |
| | 1% | $0.849 \pm 0.098$ | $0.088 \pm 0.002$ | $0.277 \pm 0.022$ | $0.785 \pm 0.004$ | $0.208 \pm 0.017$ | $0.742 \pm 0.012$ | $0.661 \pm 0.007$ | $0.550 \pm 0.006$ | $0.611 \pm 0.054$ | $0.532 \pm 0.007$ |
| | 2% | $22.021 \pm 7.462$ | $0.090 \pm 0.001$ | $0.363 \pm 0.028$ | $0.760 \pm 0.004$ | $0.289 \pm 0.017$ | $0.717 \pm 0.010$ | $0.690 \pm 0.063$ | $0.543 \pm 0.004$ | $0.616 \pm 0.056$ | $0.528 \pm 0.004$ |
| restaurant | 0.5% | $0.018 \pm 0.011$ | $0.156 \pm 0.001$ | $0.064 \pm 0.011$ | $0.346 \pm 0.006$ | $0.044 \pm 0.006$ | $0.303 \pm 0.004$ | $1.143 \pm 0.126$ | $0.887 \pm 0.005$ | $1.104 \pm 0.114$ | $0.901 \pm 0.006$ |
| | 1% | $0.176 \pm 0.059$ | $0.156 \pm 0.002$ | $0.094 \pm 0.018$ | $0.346 \pm 0.001$ | $0.069 \pm 0.010$ | $0.303 \pm 0.001$ | $1.125 \pm 0.033$ | $0.890 \pm 0.007$ | $1.080 \pm 0.006$ | $0.905 \pm 0.013$ |
| | 2% | $8.699 \pm 6.793$ | $0.158 \pm 0.002$ | $0.127 \pm 0.013$ | $0.335 \pm 0.001$ | $0.098 \pm 0.008$ | $0.293 \pm 0.002$ | $1.156 \pm 0.107$ | $0.888 \pm 0.009$ | $1.140 \pm 0.120$ | $0.901 \pm 0.005$ |
| adult | 0.5% | $0.006 \pm 0.001$ | $1.342 \pm 0.027$ | $0.273 \pm 0.023$ | $7.306 \pm 0.517$ | $0.088 \pm 0.005$ | $2.090 \pm 0.067$ | $4.425 \pm 0.436$ | $3.580 \pm 0.014$ | $4.150 \pm 0.457$ | $3.724 \pm 0.008$ |
| | 1% | $0.017 \pm 0.001$ | $1.329 \pm 0.029$ | $0.412 \pm 0.076$ | $7.840 \pm 0.630$ | $0.129 \pm 0.006$ | $2.128 \pm 0.075$ | $4.315 \pm 0.030$ | $3.603 \pm 0.090$ | $4.041 \pm 0.012$ | $3.700 \pm 0.014$ |
| | 2% | $0.218 \pm 0.018$ | $1.313 \pm 0.038$ | $0.613 \pm 0.075$ | $7.959 \pm 0.593$ | $0.193 \pm 0.007$ | $2.051 \pm 0.063$ | $4.566 \pm 0.527$ | $3.580 \pm 0.030$ | $4.246 \pm 0.370$ | $3.691 \pm 0.015$ |
| compas | 0.5% | $0.012 \pm 0.004$ | $0.045 \pm 0.002$ | $0.001 \pm 0.000$ | $0.011 \pm 0.000$ | $0.001 \pm 0.000$ | $0.009 \pm 0.000$ | $0.581 \pm 0.057$ | $0.465 \pm 0.005$ | $0.559 \pm 0.052$ | $0.474 \pm 0.019$ |
| | 1% | $0.067 \pm 0.005$ | $0.045 \pm 0.002$ | $0.001 \pm 0.000$ | $0.011 \pm 0.000$ | $0.001 \pm 0.000$ | $0.009 \pm 0.000$ | $0.567 \pm 0.007$ | $0.462 \pm 0.005$ | $0.541 \pm 0.005$ | $0.464 \pm 0.004$ |
| | 2% | $0.296 \pm 0.027$ | $0.044 \pm 0.002$ | $0.001 \pm 0.000$ | $0.010 \pm 0.000$ | $0.001 \pm 0.000$ | $0.009 \pm 0.000$ | $0.588 \pm 0.056$ | $0.470 \pm 0.032$ | $0.563 \pm 0.048$ | $0.462 \pm 0.005$ |
| bank | 0.5% | $0.006 \pm 0.001$ | $1.082 \pm 0.023$ | $0.165 \pm 0.056$ | $2.796 \pm 0.059$ | $0.079 \pm 0.022$ | $1.491 \pm 0.009$ | $5.721 \pm 0.515$ | $4.606 \pm 0.032$ | $5.577 \pm 0.483$ | $4.648 \pm 0.010$ |
| | 1% | $0.032 \pm 0.000$ | $1.076 \pm 0.017$ | $0.246 \pm 0.035$ | $2.803 \pm 0.062$ | $0.129 \pm 0.029$ | $1.480 \pm 0.009$ | $5.687 \pm 0.019$ | $4.579 \pm 0.012$ | $5.623 \pm 0.066$ | $4.618 \pm 0.005$ |
| | 2% | $0.129 \pm 0.000$ | $1.063 \pm 0.021$ | $0.441 \pm 0.062$ | $3.007 \pm 0.036$ | $0.214 \pm 0.027$ | $1.507 \pm 0.008$ | $5.980 \pm 0.414$ | $4.551 \pm 0.011$ | $5.971 \pm 0.466$ | $4.591 \pm 0.010$ |

# C  Experiments: Fairness

Table 20: Summary of datasets used for Fairness Evaluation.

| Dataset | # Inst. | # Feat. | % Pos. | Sens. Feat. | Sens. % |
|---|---|---|---|---|---|
| adult [23] | 45222 | 18 | 24.8% | Sex | 32.5% |
| bank [66] | 45211 | 47 | 11.7% | marital | 39.8% |
| compas [51] | 6172 | 10 | 54.5% | Race | 65.9% |
| german-credit [23] | 1000 | 70 | 70.0% | Gender | 31.0% |
| oulad [50] | 21562 | 46 | 68.0% | gender:M | 46.4% |
| student-por [16] | 649 | 56 | 84.6% | Sex | 59.0% |

## C.1  Setup

We ran fairness experiments on 6 datasets: adult, bank, compas, german-credit, oulad, and student-por. Details are shown in Table 20. They are all binarized datasets. The datasets are split into training, selection, and test sets using a 70-10-20 split. One note is that we follow the convention of dropping the sensitive feature before training on TreeFARMS. The selection set is used to (1) find representative trees from the Rashomon set under different criteria (2) apply the post-processing step for PostCART and PostGBT.

Note that FOCT and post-processing methods optimize all three fairness metrics, while DPF works only with statistical parity. We ran experiments 5 times and reported the mean and standard deviation for every method and dataset.

We configure each method as follows:

- **TreeFARMS**: We construct the Rashomon set with depth=[2,3,4], $\lambda = 0.01, \epsilon = 0.05$ and select the best tree that maximizes the fairness parity score and validation accuracy.
- **DPF**: We train trees with depth=[2,3,4].
- **FOCT**: We train trees with depth=[2,3,4] and regularization $\lambda = 0.01$.
- **PostCART**: We train CART with depth=[2,3,4] and the minimum sample per leaf is set to 1% of the training sample size.
- **PostGBT**: We train GBT with depth=[2,3,4] and minimum sample per leaf=1% of the training sample size.

For fairness parameters, we sweep through values between $[0,1]$ roughly in a logarithmic scale. For post-processing methods and DPF, we consider these values: $[0.001, 0.005, 0.01, 0.02, 0.03, 0.04, 0.05, 0.07, 0.09, 0.1, 0.3, 0.5, 1]$. For FOCT, due to its lengthy optimization process, and its optimal behavior, close parameters lead to the same tree, so we only sweep through $[0.001, 0.01, 0.1, 1]$.

## C.2  Results

For each dataset, depth, and fairness metric, we display the performance of the entire Rashomon set and baseline trees evaluated on the test set (Figures 7-9). For fairness baselines, the models across five folds are plotted as scatter plots with different markers. For TreeFARMS, models across five folds are plotted using kernel density estimation to generate the blue contours of the distribution of the Rashomon set. On the x-axis, we plot fairness parity, and on the y-axis, we plot accuracy.

While the Rashomon set typically contains trees generated by baseline methods, we observe cases where it fails to capture all baseline models. We identify two scenarios where this occurs. First, the Rashomon set may exclude models that sacrifice significant accuracy to satisfy fairness constraints. This is evident in the compas dataset, where optimal fair trees (such as DPF and FOCT) successfully navigate this tradeoff but fall outside the Rashomon set.

Second, at depth 2, the Rashomon set appears too constrained to include the higher-performing models produced by postGBT. We observe this in the adult and oulad datasets, where postGBT's complex architecture and ensemble nature allow it to outperform the simpler trees within the set. In both cases, these exclusions illustrate fundamental tradeoffs in machine learning: the tension between fairness and accuracy, and the balance between complexity and interpretability.

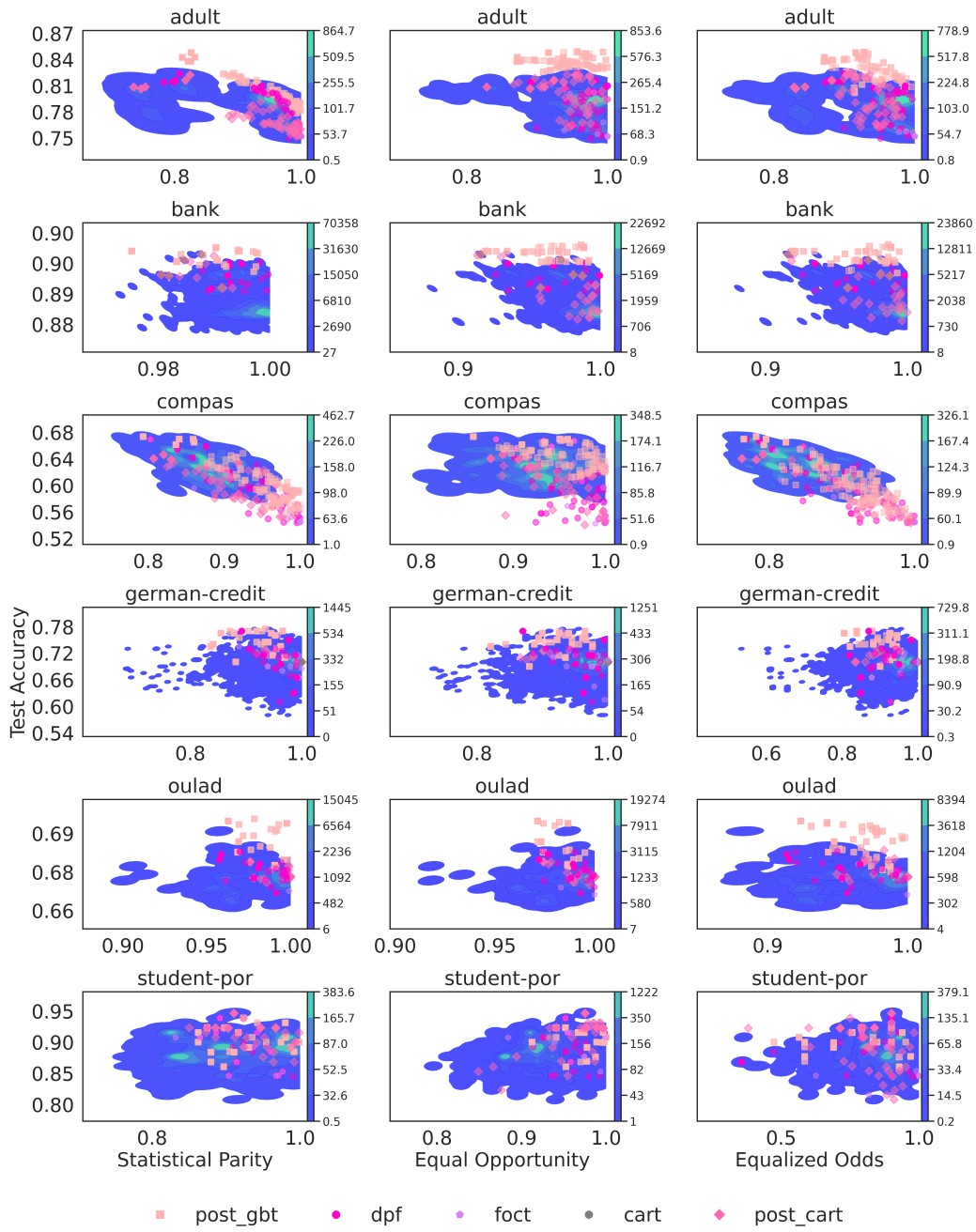

Figure 7: Comparison of test accuracy and fairness between trees in the Rashomon set at depth 2

Table 21: Percentage of RSET trees performing equally or better than ALL baseline models at depth 2 (mean ± std across folds)

| Dataset | Statistical Parity | Equal Opportunity | Equalized Odds |
|---|---|---|---|
| adult | 4.2 ± 3.5% | 3.5 ± 3.6% | 1.0 ± 0.1% |
| bank | 0.8 ± 0.5% | 1.3 ± 0.6% | 0.8 ± 0.5% |
| compas | 0.0 ± 0.0% | 6.7 ± 10.4% | 0.0 ± 0.0% |
| german-credit | 9.1 ± 9.7% | 13.7 ± 12.3% | 11.3 ± 14.5% |
| oulad | 6.6 ± 3.6% | 7.1 ± 3.6% | 6.6 ± 3.6% |
| student-por | 34.0 ± 40.9% | 3.5 ± 2.1% | 22.6 ± 27.4% |

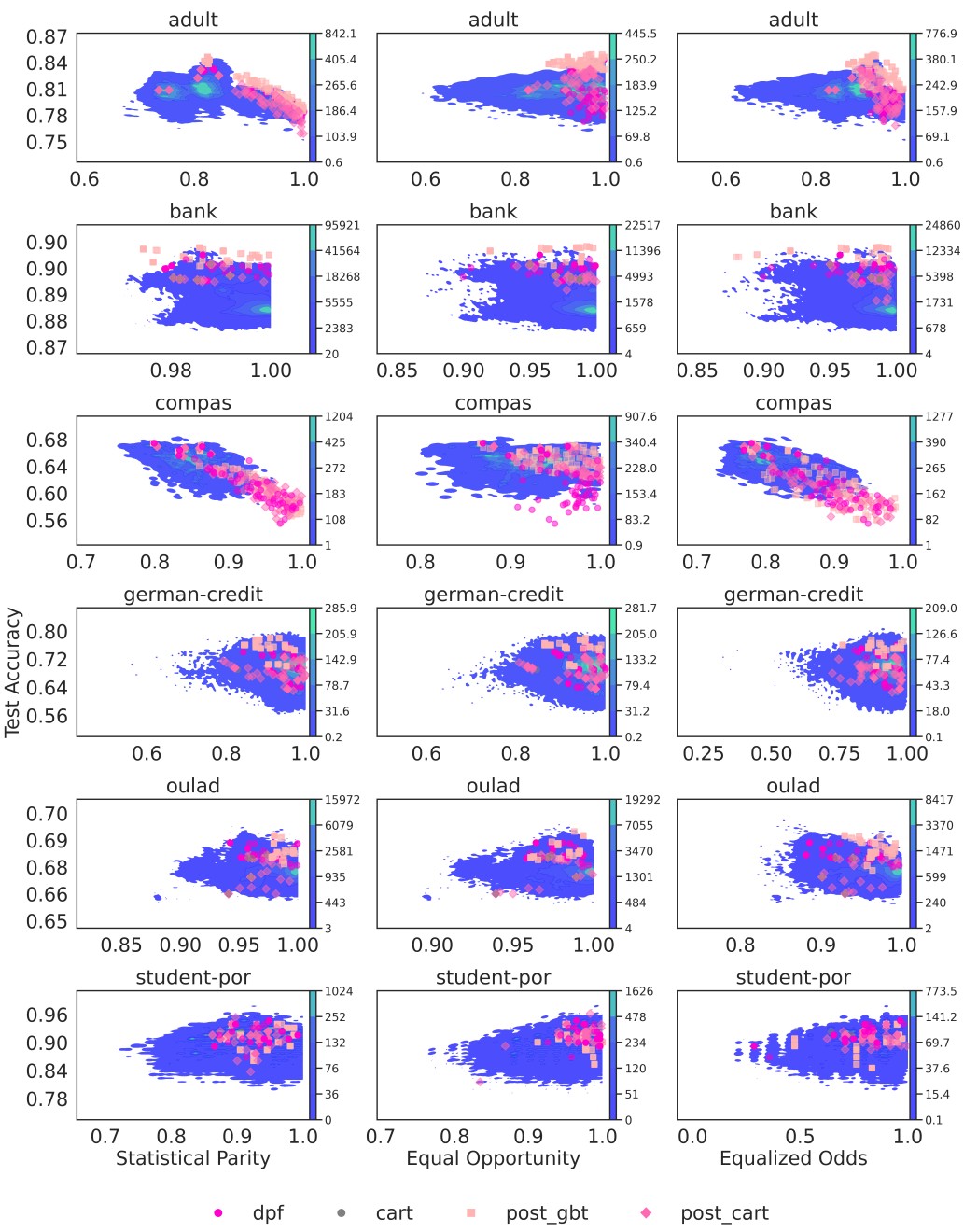

Figure 8: Comparison of test accuracy and fairness between trees in the Rashomon set at depth 3

Table 22: Percentage of RSET trees performing equally or better than ALL baseline models at depth 3 (mean ± std across folds)

| Dataset | Statistical Parity | Equal Opportunity | Equalized Odds |
|---|---|---|---|
| adult | 0.5 ± 0.4% | 24.3 ± 12.6% | 9.7 ± 5.4% |
| bank | 96.3 ± 2.7% | 54.3 ± 29.5% | 60.0 ± 18.6% |
| compas | 0.1 ± 0.1% | 1.4 ± 2.3% | 3.3 ± 4.4% |
| german-credit | 45.1 ± 30.5% | 43.0 ± 24.8% | 24.6 ± 15.8% |
| oulad | 60.7 ± 19.5% | 63.3 ± 20.2% | 64.3 ± 6.3% |
| student-por | 27.2 ± 24.0% | 33.7 ± 21.3% | 38.0 ± 21.8% |

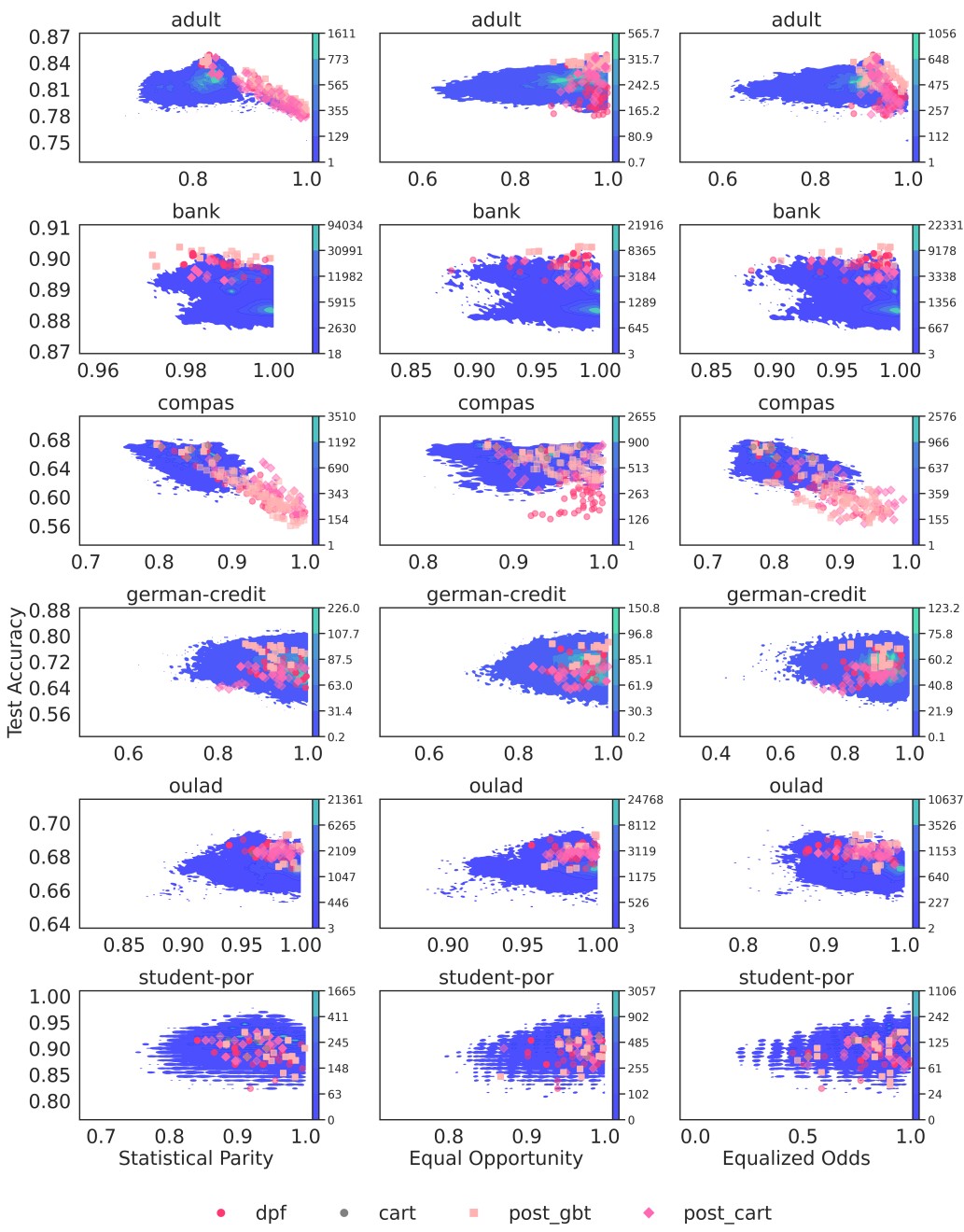

Figure 9: Comparison of test accuracy and fairness between trees in the Rashomon set at depth 4

Table 23: Percentage of RSET trees performing equally or better than ALL baseline models at depth 4 (mean ± std across folds)

| Dataset | Statistical Parity | Equal Opportunity | Equalized Odds |
|---|---|---|---|
| adult | 0.0 ± 0.0% | 10.6 ± 9.1% | 0.4 ± 0.3% |
| bank | 98.1 ± 1.1% | 44.8 ± 11.3% | 46.3 ± 10.6% |
| compas | 0.0 ± 0.0% | 2.5 ± 5.5% | 0.0 ± 0.0% |
| german-credit | 43.4 ± 31.1% | 37.1 ± 31.5% | 30.5 ± 22.9% |
| oulad | 48.3 ± 22.1% | 39.5 ± 20.5% | 73.8 ± 3.4% |
| student-por | 25.2 ± 32.5% | 34.4 ± 17.0% | 14.4 ± 13.2% |

Besides visualization, we also show some quantitative results. Tables 21, 22, 23 display the percentage of trees in the Rashomon set that perform equally or better than all baselines at different configurations on the test data. We notice that other than adult and compas datasets, a sizable portion of trees within the Rashomon sets outperform all baselines at depth 3 and 4. We also observe that as we increase the depth of trees, the Rashomon set is able to generate superior trees more consistently. However, note that some of these performances have high variance. Nevertheless, for many datasets the Rashomon set contains trees that are fairer than baselines.

We also show examples of comparing model-to-model here. Following the settings in [79], we use alpha of 0.3 for post-processing methods and delta of 0.01 for DPF and FOCT. For TreeFARMS, we select the fair tree by maximizing accuracy $+ 0.3*$ fairness parity on the validation set. We also show the optimal tree in comparison.

We show the statistical parity and accuracy on the test set in Table 24 for depth 2 and Table 25 and 26 for depth 3 and 4. We also show this for equal opportunity in Table 30-32, and equalized odds in Table 27-29.

Table 24: Test parity score in statistical parity (SP) and accuracy (Acc) for the sparse decision trees with depth limit 2.

| Model | CART | | POST_CART | | POST_GBT | | DPF | | FOCT | | RSET_sp | | RSET_opt | |
| Metric | SP | Acc | SP | Acc | SP | Acc | SP | Acc | SP | Acc | SP | Acc | SP | Acc |
|---|---|---|---|---|---|---|---|---|---|---|---|---|---|---|
| adult | 0.776 ± 0.042 | **0.812 ± 0.007** | 0.917 ± 0.018 | 0.778 ± 0.006 | 0.924 ± 0.009 | 0.811 ± 0.004 | 0.927 ± 0.004 | 0.793 ± 0.002 | **0.986 ± 0.027** | 0.330 ± 0.077 | 0.949 ± 0.014 | 0.801 ± 0.002 | 0.812 ± 0.008 | 0.808 ± 0.003 |
| bank | 0.985 ± 0.004 | 0.894 ± 0.004 | 0.985 ± 0.004 | 0.894 ± 0.003 | 0.988 ± 0.007 | **0.898 ± 0.002** | 0.993 ± 0.002 | 0.894 ± 0.002 | **1.000 ± 0.000** | 0.117 ± 0.000 | 0.992 ± 0.003 | 0.895 ± 0.002 | 0.996 ± 0.005 | 0.886 ± 0.005 |
| compas | 0.851 ± 0.046 | 0.638 ± 0.007 | 0.912 ± 0.034 | 0.602 ± 0.007 | 0.930 ± 0.011 | 0.613 ± 0.007 | 0.926 ± 0.017 | 0.605 ± 0.006 | **0.939 ± 0.016** | 0.585 ± 0.011 | 0.855 ± 0.033 | 0.650 ± 0.014 | 0.844 ± 0.027 | **0.663 ± 0.006** |
| german-credit | **0.987 ± 0.029** | 0.705 ± 0.011 | **0.987 ± 0.029** | 0.705 ± 0.011 | 0.924 ± 0.018 | **0.745 ± 0.024** | 0.949 ± 0.038 | 0.711 ± 0.043 | 0.965 ± 0.018 | 0.707 ± 0.027 | 0.973 ± 0.017 | 0.685 ± 0.026 | 0.960 ± 0.023 | 0.711 ± 0.040 |
| oulad | 0.991 ± 0.013 | 0.674 ± 0.003 | 0.992 ± 0.011 | 0.674 ± 0.002 | 0.979 ± 0.014 | **0.687 ± 0.007** | 0.977 ± 0.008 | 0.677 ± 0.003 | 0.994 ± 0.011 | 0.641 ± 0.044 | 0.983 ± 0.011 | 0.676 ± 0.002 | **1.000 ± 0.000** | 0.673 ± 0.000 |
| student-por | 0.920 ± 0.037 | **0.928 ± 0.010** | 0.927 ± 0.040 | 0.918 ± 0.012 | 0.932 ± 0.046 | 0.903 ± 0.015 | 0.924 ± 0.033 | 0.910 ± 0.012 | 0.923 ± 0.039 | 0.901 ± 0.007 | **0.934 ± 0.052** | 0.897 ± 0.017 | 0.920 ± 0.037 | **0.928 ± 0.010** |

Table 25: Test parity score in statistical parity (SP) and accuracy (Acc) for the sparse decision trees with depth limit 3.

| Model | CART | | POST_CART | | POST_GBT | | DPF | | RSET_sp | | RSET_opt | |
| Metric | SP | Acc | SP | Acc | SP | Acc | SP | Acc | SP | Acc | SP | Acc |
|---|---|---|---|---|---|---|---|---|---|---|---|---|
| adult | 0.790 ± 0.045 | 0.819 ± 0.010 | 0.911 ± 0.018 | 0.797 ± 0.006 | 0.925 ± 0.010 | 0.810 ± 0.003 | **0.930 ± 0.004** | 0.804 ± 0.003 | 0.906 ± 0.058 | 0.813 ± 0.013 | 0.803 ± 0.008 | **0.823 ± 0.005** |
| bank | 0.985 ± 0.003 | 0.894 ± 0.002 | 0.986 ± 0.005 | 0.894 ± 0.006 | 0.984 ± 0.006 | **0.900 ± 0.002** | 0.986 ± 0.004 | 0.896 ± 0.002 | 0.993 ± 0.005 | 0.896 ± 0.003 | **0.996 ± 0.005** | 0.886 ± 0.005 |
| compas | 0.844 ± 0.027 | **0.663 ± 0.006** | **0.926 ± 0.020** | 0.615 ± 0.009 | 0.924 ± 0.015 | 0.620 ± 0.008 | 0.918 ± 0.020 | 0.617 ± 0.009 | 0.849 ± 0.010 | 0.651 ± 0.022 | 0.844 ± 0.027 | **0.663 ± 0.006** |
| german-credit | 0.941 ± 0.066 | 0.675 ± 0.034 | 0.933 ± 0.054 | 0.677 ± 0.031 | 0.904 ± 0.030 | **0.748 ± 0.027** | 0.946 ± 0.027 | 0.702 ± 0.038 | **0.965 ± 0.029** | 0.692 ± 0.022 | 0.951 ± 0.054 | 0.713 ± 0.031 |
| oulad | 0.959 ± 0.012 | 0.674 ± 0.009 | 0.964 ± 0.009 | 0.674 ± 0.009 | 0.984 ± 0.007 | **0.685 ± 0.005** | 0.968 ± 0.008 | 0.683 ± 0.004 | 0.984 ± 0.012 | 0.678 ± 0.007 | **1.000 ± 0.000** | 0.673 ± 0.000 |
| student-por | 0.912 ± 0.025 | 0.923 ± 0.009 | 0.921 ± 0.025 | 0.910 ± 0.018 | **0.943 ± 0.033** | 0.910 ± 0.025 | 0.936 ± 0.032 | 0.916 ± 0.016 | 0.933 ± 0.051 | 0.898 ± 0.025 | 0.920 ± 0.037 | **0.928 ± 0.010** |

Table 26: Test parity score in statistical parity (SP) and accuracy (Acc) for the sparse decision trees with depth limit 4.

| Model | CART | | POST_CART | | POST_GBT | | DPF | | RSET_sp | | RSET_opt | |
| Metric | SP | Acc | SP | Acc | SP | Acc | SP | Acc | SP | Acc | SP | Acc |
|---|---|---|---|---|---|---|---|---|---|---|---|---|
| adult | 0.840 ± 0.006 | 0.832 ± 0.009 | 0.920 ± 0.006 | 0.806 ± 0.007 | 0.925 ± 0.010 | 0.811 ± 0.003 | **0.926 ± 0.003** | 0.809 ± 0.003 | 0.891 ± 0.052 | 0.820 ± 0.013 | 0.833 ± 0.006 | **0.837 ± 0.005** |
| bank | 0.984 ± 0.003 | 0.896 ± 0.002 | 0.984 ± 0.003 | 0.896 ± 0.003 | 0.981 ± 0.005 | **0.901 ± 0.002** | 0.986 ± 0.003 | 0.899 ± 0.001 | 0.990 ± 0.004 | 0.896 ± 0.001 | **0.996 ± 0.005** | 0.886 ± 0.005 |
| compas | 0.846 ± 0.029 | **0.663 ± 0.008** | **0.927 ± 0.009** | 0.620 ± 0.009 | 0.919 ± 0.016 | 0.617 ± 0.008 | 0.913 ± 0.024 | 0.618 ± 0.007 | 0.843 ± 0.015 | 0.655 ± 0.017 | 0.844 ± 0.027 | **0.663 ± 0.006** |
| german-credit | 0.924 ± 0.053 | 0.682 ± 0.017 | 0.915 ± 0.035 | 0.682 ± 0.021 | 0.925 ± 0.021 | **0.746 ± 0.026** | 0.943 ± 0.044 | 0.709 ± 0.037 | **0.979 ± 0.022** | 0.687 ± 0.036 | 0.948 ± 0.059 | 0.705 ± 0.031 |
| oulad | 0.971 ± 0.013 | 0.682 ± 0.001 | 0.978 ± 0.009 | 0.682 ± 0.002 | 0.985 ± 0.007 | **0.685 ± 0.007** | 0.969 ± 0.010 | 0.683 ± 0.003 | 0.982 ± 0.014 | 0.682 ± 0.007 | **1.000 ± 0.000** | 0.673 ± 0.000 |
| student-por | 0.923 ± 0.023 | 0.908 ± 0.016 | 0.924 ± 0.031 | 0.905 ± 0.016 | **0.947 ± 0.041** | 0.901 ± 0.023 | 0.917 ± 0.035 | 0.889 ± 0.015 | 0.898 ± 0.039 | 0.920 ± 0.021 | 0.920 ± 0.037 | **0.928 ± 0.010** |

Table 27: Test parity score in equalized odds (EO) and accuracy (Acc) for the sparse decision trees with depth limit 2.

| Model | CART | | POST_CART | | POST_GBT | | DPF | | FOCT | | RSET_eo | | RSET_opt | |
| Metric | EOdds | Acc | EOdds | Acc | EOdds | Acc | EOdds | Acc | EOdds | Acc | EOdds | Acc | EOdds | Acc |
|---|---|---|---|---|---|---|---|---|---|---|---|---|---|---|
| adult | 0.858 ± 0.033 | 0.812 ± 0.007 | 0.918 ± 0.007 | 0.797 ± 0.003 | 0.925 ± 0.016 | **0.830 ± 0.003** | 0.950 ± 0.001 | 0.793 ± 0.002 | **0.995 ± 0.006** | 0.554 ± 0.171 | 0.960 ± 0.042 | 0.806 ± 0.006 | 0.890 ± 0.011 | 0.808 ± 0.003 |
| bank | 0.963 ± 0.029 | 0.894 ± 0.004 | 0.967 ± 0.028 | 0.892 ± 0.004 | 0.966 ± 0.025 | **0.898 ± 0.002** | 0.969 ± 0.025 | 0.894 ± 0.002 | **1.000 ± 0.000** | 0.117 ± 0.000 | 0.966 ± 0.021 | 0.894 ± 0.003 | 0.993 ± 0.013 | 0.886 ± 0.005 |
| compas | 0.815 ± 0.051 | 0.638 ± 0.007 | 0.909 ± 0.023 | 0.593 ± 0.010 | 0.887 ± 0.017 | 0.615 ± 0.011 | 0.889 ± 0.020 | 0.605 ± 0.006 | **0.924 ± 0.013** | 0.590 ± 0.005 | 0.860 ± 0.056 | 0.636 ± 0.021 | 0.798 ± 0.030 | **0.663 ± 0.006** |
| german-credit | **0.980 ± 0.044** | 0.705 ± 0.011 | 0.976 ± 0.055 | 0.703 ± 0.007 | 0.894 ± 0.027 | **0.744 ± 0.025** | 0.890 ± 0.051 | 0.711 ± 0.043 | 0.920 ± 0.036 | 0.705 ± 0.031 | 0.955 ± 0.028 | 0.701 ± 0.016 | 0.917 ± 0.031 | 0.711 ± 0.040 |
| oulad | 0.984 ± 0.022 | 0.674 ± 0.003 | 0.986 ± 0.021 | 0.674 ± 0.003 | 0.961 ± 0.014 | **0.686 ± 0.007** | 0.950 ± 0.017 | 0.677 ± 0.003 | 0.993 ± 0.007 | 0.645 ± 0.039 | 0.995 ± 0.005 | 0.673 ± 0.002 | **1.000 ± 0.000** | 0.673 ± 0.000 |
| student-por | 0.817 ± 0.069 | **0.928 ± 0.010** | 0.825 ± 0.040 | 0.900 ± 0.013 | 0.785 ± 0.128 | 0.902 ± 0.014 | 0.792 ± 0.106 | 0.910 ± 0.012 | **0.865 ± 0.061** | 0.884 ± 0.007 | 0.809 ± 0.132 | 0.888 ± 0.009 | 0.817 ± 0.069 | **0.928 ± 0.010** |

Table 28: Test parity score in equalized odds (EO) and accuracy (Acc) for the sparse decision trees with depth limit 3.

| Model | CART | | POST_CART | | POST_GBT | | DPF | | RSET_eo | | RSET_opt | |
| Metric | EOdds | Acc | EOdds | Acc | EOdds | Acc | EOdds | Acc | EOdds | Acc | EOdds | Acc |
|---|---|---|---|---|---|---|---|---|---|---|---|---|
| adult | 0.875 ± 0.039 | 0.819 ± 0.010 | 0.933 ± 0.008 | 0.806 ± 0.003 | 0.929 ± 0.020 | 0.831 ± 0.004 | **0.950 ± 0.003** | 0.804 ± 0.003 | 0.926 ± 0.009 | **0.832 ± 0.004** | 0.890 ± 0.011 | 0.823 ± 0.005 |
| bank | 0.975 ± 0.016 | 0.894 ± 0.002 | 0.977 ± 0.014 | 0.893 ± 0.002 | 0.968 ± 0.031 | **0.900 ± 0.002** | 0.975 ± 0.016 | 0.896 ± 0.002 | 0.969 ± 0.019 | 0.894 ± 0.003 | **0.993 ± 0.013** | 0.886 ± 0.005 |
| compas | 0.798 ± 0.030 | 0.663 ± 0.006 | 0.877 ± 0.047 | 0.615 ± 0.017 | **0.883 ± 0.029** | 0.614 ± 0.011 | 0.876 ± 0.022 | 0.617 ± 0.009 | 0.839 ± 0.043 | 0.645 ± 0.025 | 0.798 ± 0.030 | **0.663 ± 0.006** |
| german-credit | 0.923 ± 0.065 | 0.675 ± 0.034 | 0.901 ± 0.066 | 0.678 ± 0.028 | 0.893 ± 0.041 | **0.745 ± 0.026** | 0.902 ± 0.023 | 0.702 ± 0.038 | 0.912 ± 0.069 | 0.694 ± 0.015 | **0.925 ± 0.051** | 0.713 ± 0.031 |
| oulad | 0.930 ± 0.023 | 0.674 ± 0.009 | 0.939 ± 0.026 | 0.674 ± 0.009 | 0.962 ± 0.016 | **0.685 ± 0.005** | 0.931 ± 0.026 | 0.683 ± 0.004 | 0.978 ± 0.015 | 0.677 ± 0.006 | **1.000 ± 0.000** | 0.673 ± 0.000 |
| student-por | 0.782 ± 0.098 | **0.923 ± 0.009** | 0.825 ± 0.057 | 0.913 ± 0.006 | 0.795 ± 0.190 | 0.903 ± 0.023 | 0.745 ± 0.068 | 0.916 ± 0.016 | 0.757 ± 0.062 | 0.900 ± 0.023 | 0.817 ± 0.069 | **0.928 ± 0.010** |

Table 29: Test parity score in equalized odds (EO) and accuracy (Acc) for the sparse decision trees with depth limit 4.

| Model | CART | | POST_CART | | POST_GBT | | DPF | | RSET_eo | | RSET_opt | |
| Metric | EOdds | Acc | EOdds | Acc | EOdds | Acc | EOdds | Acc | EOdds | Acc | EOdds | Acc |
|---|---|---|---|---|---|---|---|---|---|---|---|---|
| adult | 0.925 ± 0.008 | 0.832 ± 0.009 | 0.942 ± 0.002 | 0.814 ± 0.004 | 0.932 ± 0.012 | 0.831 ± 0.003 | **0.950 ± 0.009** | 0.809 ± 0.003 | 0.927 ± 0.009 | **0.836 ± 0.008** | 0.924 ± 0.007 | **0.837 ± 0.005** |
| bank | 0.973 ± 0.027 | 0.896 ± 0.002 | 0.973 ± 0.024 | 0.895 ± 0.003 | 0.978 ± 0.018 | **0.901 ± 0.002** | 0.976 ± 0.010 | 0.899 ± 0.001 | 0.964 ± 0.018 | 0.894 ± 0.004 | **0.993 ± 0.013** | 0.886 ± 0.005 |
| compas | 0.804 ± 0.033 | **0.663 ± 0.008** | **0.892 ± 0.018** | 0.616 ± 0.010 | 0.874 ± 0.034 | 0.615 ± 0.010 | 0.866 ± 0.023 | 0.618 ± 0.007 | 0.836 ± 0.041 | 0.644 ± 0.025 | 0.798 ± 0.030 | **0.663 ± 0.006** |
| german-credit | 0.901 ± 0.049 | 0.682 ± 0.017 | 0.873 ± 0.045 | 0.678 ± 0.021 | 0.898 ± 0.040 | **0.749 ± 0.028** | 0.897 ± 0.045 | 0.709 ± 0.037 | 0.883 ± 0.064 | 0.696 ± 0.022 | **0.917 ± 0.064** | 0.705 ± 0.031 |
| oulad | 0.931 ± 0.030 | 0.682 ± 0.001 | 0.950 ± 0.020 | 0.682 ± 0.002 | 0.955 ± 0.012 | **0.685 ± 0.007** | 0.931 ± 0.024 | 0.683 ± 0.003 | 0.970 ± 0.019 | 0.678 ± 0.005 | **1.000 ± 0.000** | 0.673 ± 0.000 |
| student-por | 0.793 ± 0.180 | **0.908 ± 0.016** | 0.811 ± 0.134 | 0.901 ± 0.013 | 0.817 ± 0.145 | 0.894 ± 0.024 | 0.808 ± 0.073 | 0.889 ± 0.015 | **0.852 ± 0.058** | 0.898 ± 0.017 | 0.817 ± 0.069 | **0.928 ± 0.010** |

Table 30: Test parity score in equal opportunity (EOpp) and accuracy (Acc) for the sparse decision trees with depth limit 2.

| Model | CART | | POST_CART | | POST_GBT | | DPF | | FOCT | | RSET_eopp | | RSET_opt | |
|---|---|---|---|---|---|---|---|---|---|---|---|---|---|---|
| Metric | EOpp | Acc | EOpp | Acc | EOpp | Acc | EOpp | Acc | EOpp | Acc | EOpp | Acc | EOpp | Acc |
| adult | 0.901 ± 0.047 | 0.812 ± 0.007 | 0.937 ± 0.015 | 0.805 ± 0.007 | 0.932 ± 0.017 | **0.838 ± 0.003** | 0.962 ± 0.011 | 0.793 ± 0.002 | **0.992 ± 0.007** | 0.725 ± 0.050 | 0.958 ± 0.040 | 0.814 ± 0.006 | 0.949 ± 0.028 | 0.808 ± 0.003 |
| bank | 0.963 ± 0.029 | 0.894 ± 0.004 | 0.968 ± 0.029 | 0.892 ± 0.004 | 0.964 ± 0.023 | **0.898 ± 0.002** | 0.970 ± 0.025 | 0.894 ± 0.002 | **1.000 ± 0.000** | 0.117 ± 0.000 | 0.971 ± 0.022 | 0.893 ± 0.004 | **0.994 ± 0.013** | 0.886 ± 0.005 |
| compas | 0.916 ± 0.045 | 0.638 ± 0.007 | 0.946 ± 0.025 | 0.614 ± 0.009 | 0.943 ± 0.035 | 0.652 ± 0.007 | **0.960 ± 0.020** | 0.605 ± 0.006 | 0.955 ± 0.021 | 0.625 ± 0.007 | 0.918 ± 0.025 | 0.660 ± 0.012 | 0.925 ± 0.033 | **0.663 ± 0.006** |
| german-credit | **0.980 ± 0.044** | 0.705 ± 0.011 | 0.979 ± 0.047 | 0.704 ± 0.009 | 0.924 ± 0.020 | **0.744 ± 0.023** | 0.948 ± 0.041 | 0.711 ± 0.043 | 0.957 ± 0.018 | 0.703 ± 0.035 | 0.932 ± 0.061 | 0.680 ± 0.025 | 0.933 ± 0.027 | 0.711 ± 0.040 |
| oulad | 0.992 ± 0.011 | 0.674 ± 0.003 | 0.993 ± 0.010 | 0.674 ± 0.002 | 0.979 ± 0.008 | **0.687 ± 0.007** | 0.986 ± 0.005 | 0.677 ± 0.003 | 0.997 ± 0.004 | 0.656 ± 0.037 | 0.990 ± 0.012 | 0.675 ± 0.004 | **1.000 ± 0.000** | 0.673 ± 0.000 |
| student-por | **0.976 ± 0.021** | **0.928 ± 0.010** | 0.966 ± 0.019 | 0.917 ± 0.011 | 0.969 ± 0.024 | 0.905 ± 0.016 | 0.967 ± 0.018 | 0.910 ± 0.012 | 0.970 ± 0.022 | 0.910 ± 0.015 | 0.959 ± 0.029 | 0.888 ± 0.009 | **0.976 ± 0.021** | **0.928 ± 0.010** |

Table 31: Test parity score in equal opportunity (EOpp) and accuracy (Acc) for the sparse decision trees with depth limit 3.

| Model | CART | | POST_CART | | POST_GBT | | DPF | | RSET_eopp | | RSET_opt | |
|---|---|---|---|---|---|---|---|---|---|---|---|---|
| Metric | EOpp | Acc | EOpp | Acc | EOpp | Acc | EOpp | Acc | EOpp | Acc | EOpp | Acc |
| adult | 0.927 ± 0.060 | 0.819 ± 0.010 | 0.938 ± 0.044 | 0.818 ± 0.008 | 0.940 ± 0.025 | **0.839 ± 0.003** | 0.963 ± 0.008 | 0.804 ± 0.003 | 0.962 ± 0.030 | 0.830 ± 0.006 | **0.968 ± 0.009** | 0.823 ± 0.005 |
| bank | 0.975 ± 0.016 | 0.894 ± 0.002 | 0.976 ± 0.017 | 0.894 ± 0.004 | 0.968 ± 0.029 | **0.901 ± 0.002** | 0.976 ± 0.017 | 0.896 ± 0.002 | 0.972 ± 0.017 | 0.895 ± 0.003 | **0.994 ± 0.013** | 0.886 ± 0.005 |
| compas | 0.925 ± 0.033 | 0.663 ± 0.006 | 0.950 ± 0.025 | 0.648 ± 0.004 | 0.944 ± 0.035 | 0.655 ± 0.006 | **0.957 ± 0.017** | 0.617 ± 0.009 | 0.917 ± 0.030 | 0.661 ± 0.013 | 0.925 ± 0.033 | **0.663 ± 0.006** |
| german-credit | 0.943 ± 0.055 | 0.675 ± 0.034 | 0.941 ± 0.064 | 0.678 ± 0.032 | 0.920 ± 0.027 | **0.747 ± 0.026** | 0.947 ± 0.018 | 0.702 ± 0.038 | **0.952 ± 0.058** | 0.684 ± 0.026 | 0.939 ± 0.044 | 0.713 ± 0.031 |
| oulad | 0.968 ± 0.018 | 0.674 ± 0.009 | 0.971 ± 0.017 | 0.674 ± 0.009 | 0.982 ± 0.011 | **0.685 ± 0.005** | 0.977 ± 0.004 | 0.683 ± 0.004 | 0.988 ± 0.006 | 0.677 ± 0.004 | **1.000 ± 0.000** | 0.673 ± 0.000 |
| student-por | 0.965 ± 0.012 | 0.923 ± 0.009 | 0.969 ± 0.019 | 0.916 ± 0.013 | 0.966 ± 0.020 | 0.904 ± 0.023 | **0.981 ± 0.006** | 0.916 ± 0.016 | 0.950 ± 0.045 | 0.900 ± 0.027 | 0.976 ± 0.021 | **0.928 ± 0.010** |

Table 32: Test parity score in equal opportunity (EOpp) and accuracy (Acc) for the sparse decision trees with depth limit 4.

| Model | CART | | POST_CART | | POST_GBT | | DPF | | RSET_eopp | | RSET_opt | |
|---|---|---|---|---|---|---|---|---|---|---|---|---|
| Metric | EOpp | Acc | EOpp | Acc | EOpp | Acc | EOpp | Acc | EOpp | Acc | EOpp | Acc |
| adult | **0.967 ± 0.022** | 0.832 ± 0.009 | 0.961 ± 0.010 | 0.825 ± 0.006 | 0.953 ± 0.021 | **0.840 ± 0.003** | 0.963 ± 0.010 | 0.809 ± 0.003 | 0.959 ± 0.039 | 0.833 ± 0.007 | 0.954 ± 0.029 | 0.837 ± 0.005 |
| bank | 0.973 ± 0.027 | 0.896 ± 0.002 | 0.974 ± 0.025 | 0.895 ± 0.003 | 0.979 ± 0.016 | **0.901 ± 0.002** | 0.976 ± 0.010 | 0.899 ± 0.001 | 0.967 ± 0.019 | 0.894 ± 0.004 | **0.994 ± 0.013** | 0.886 ± 0.005 |
| compas | 0.925 ± 0.035 | **0.663 ± 0.008** | 0.946 ± 0.034 | 0.648 ± 0.010 | 0.943 ± 0.031 | 0.654 ± 0.007 | **0.960 ± 0.013** | 0.618 ± 0.007 | 0.916 ± 0.025 | 0.660 ± 0.013 | 0.925 ± 0.033 | **0.663 ± 0.006** |
| german-credit | 0.926 ± 0.066 | 0.682 ± 0.017 | 0.910 ± 0.047 | 0.681 ± 0.022 | **0.942 ± 0.042** | **0.750 ± 0.029** | 0.940 ± 0.037 | 0.709 ± 0.037 | 0.938 ± 0.037 | 0.691 ± 0.022 | 0.931 ± 0.061 | 0.705 ± 0.031 |
| oulad | 0.985 ± 0.011 | 0.682 ± 0.001 | 0.988 ± 0.007 | 0.682 ± 0.003 | 0.985 ± 0.011 | **0.685 ± 0.007** | 0.982 ± 0.009 | 0.683 ± 0.003 | 0.986 ± 0.011 | 0.683 ± 0.008 | **1.000 ± 0.000** | 0.673 ± 0.000 |
| student-por | 0.974 ± 0.015 | 0.908 ± 0.016 | 0.972 ± 0.017 | 0.901 ± 0.010 | 0.966 ± 0.025 | 0.895 ± 0.021 | 0.955 ± 0.026 | 0.889 ± 0.015 | 0.951 ± 0.027 | 0.909 ± 0.017 | 0.976 ± 0.021 | **0.928 ± 0.010** |

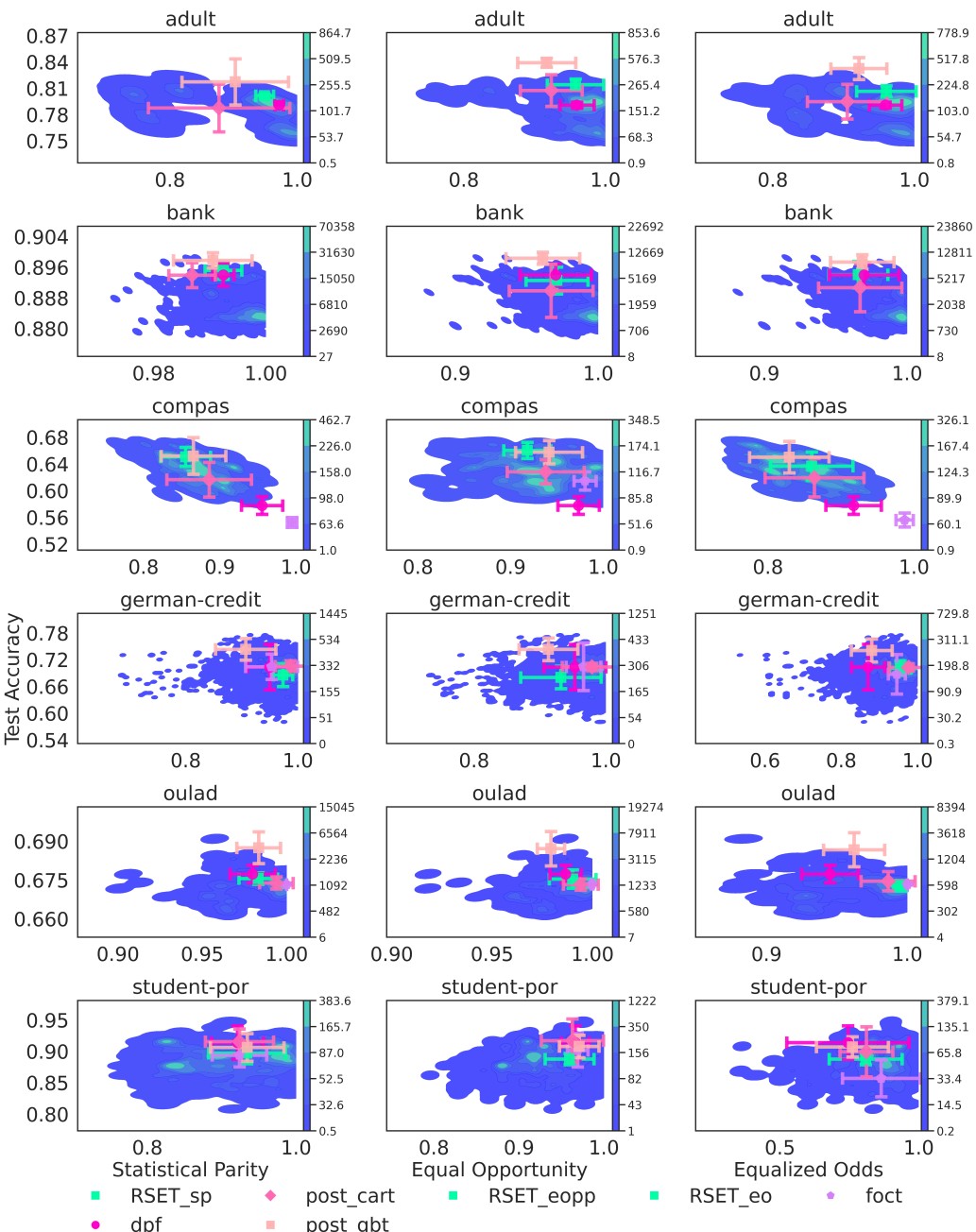

Figure 10: Model-to-model comparison of test accuracy and fairness between baselines and selected tree in the Rashomon set at depth 2.

Our earlier plots already demonstrate that the Rashomon set covers the hypothesis-space of fairness models within the parameters we sweep. Here we show clearly that we can select a model that is comparable with other fairness baselines. In Figure 10-12, we display the performance of the Rashomon set as blue contours, and we plot the table results of fairness baselines and the fair tree selected in the Rashomon set as dots with error bars.

In these results, we fixed the $\alpha$ parameter to control tradeoff. When $\alpha = 1$, the fairness is valued as much as accuracy. During our experiment, we chose $\alpha = 0.3$ to be consistent with baseline setups to demonstrate that a representative from the Rashomon set can be comparable with baselines that optimize fairness, but we don't claim $\alpha = 0.3$ to be the optimal value.

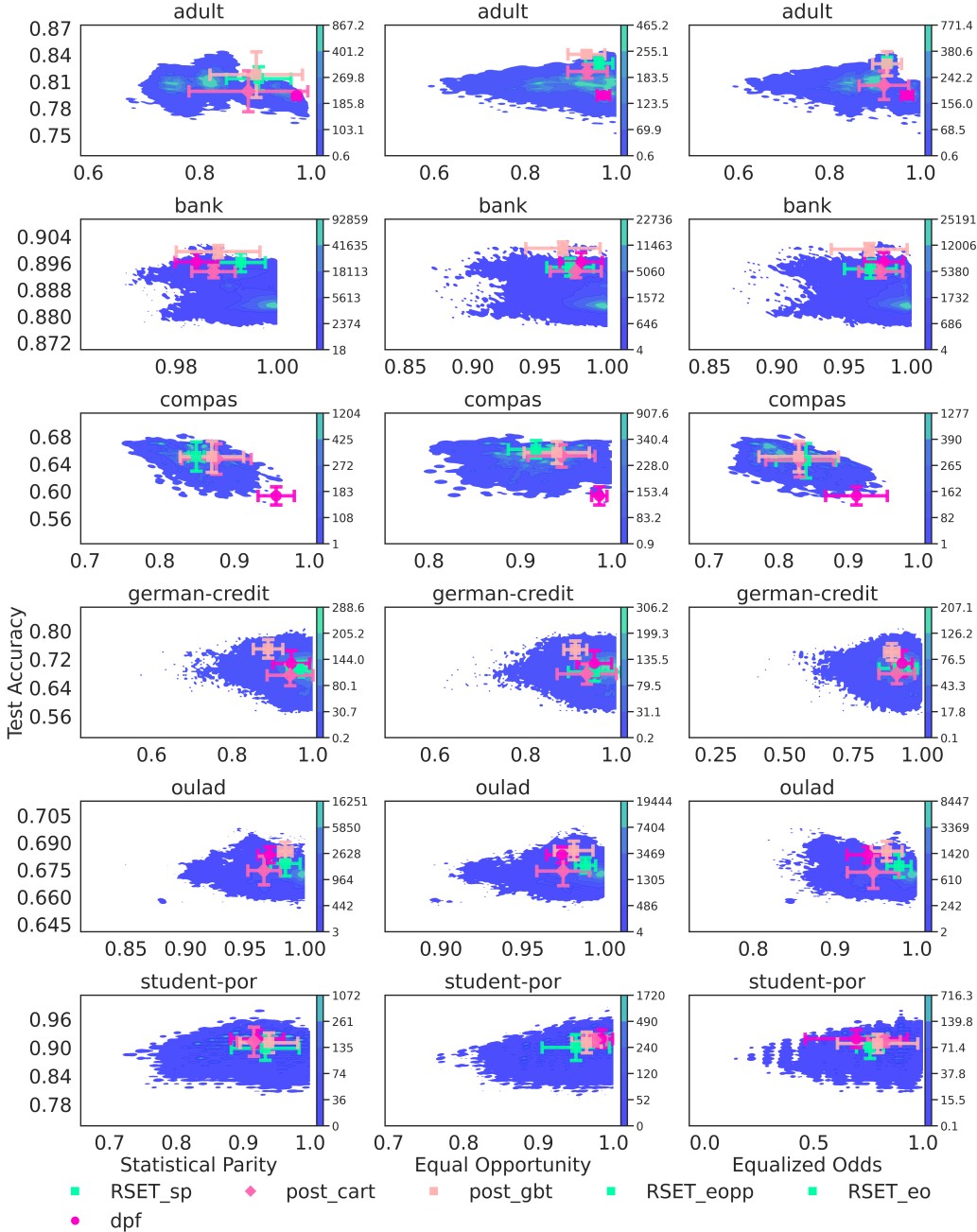

Figure 11: Model-to-model comparison of test accuracy and fairness between baselines and selected tree in the Rashomon set at depth 3.

Table 33: Results for Train Time across depths (Avg±Std).

| Dataset | Depth 2 | | | Depth 3 | | Depth 4 | |
|---|---|---|---|---|---|---|---|
| | DPF | FOCT | RSET | DPF | RSET | DPF | RSET |
| adult | 0.042±0.001 | 3999.251±1.778 | 0.354±0.006 | 0.169±0.011 | 1.613±0.023 | 931.531±319.693 | 12.711±0.489 |
| bank | 0.123±0.006 | 4729.554±5.793 | 2.055±0.123 | 1.435±0.095 | 35.772±2.646 | 23654.268±9915.746 | 1286.891±31.113 |
| compas | 0.011±0.001 | 3615.251±1.286 | 0.055±0.005 | 0.018±0.000 | 0.116±0.005 | 4.237±1.468 | 0.269±0.016 |
| german | 0.015±0.001 | 3603.118±0.425 | 0.304±0.004 | 1.312±0.127 | 35.523±2.978 | 336.964±98.572 | 11334.616±1919.513 |
| oulad | 0.031±0.001 | 3661.748±4.469 | 0.574±0.040 | 0.971±0.013 | 10.236±0.335 | 23595.305±4914.847 | 261.087±16.785 |
| stud-por | 0.010±0.001 | 1055.451±326.725 | 0.140±0.021 | 0.249±0.020 | 14.208±1.552 | 16.007±1.268 | 3134.127±668.377 |

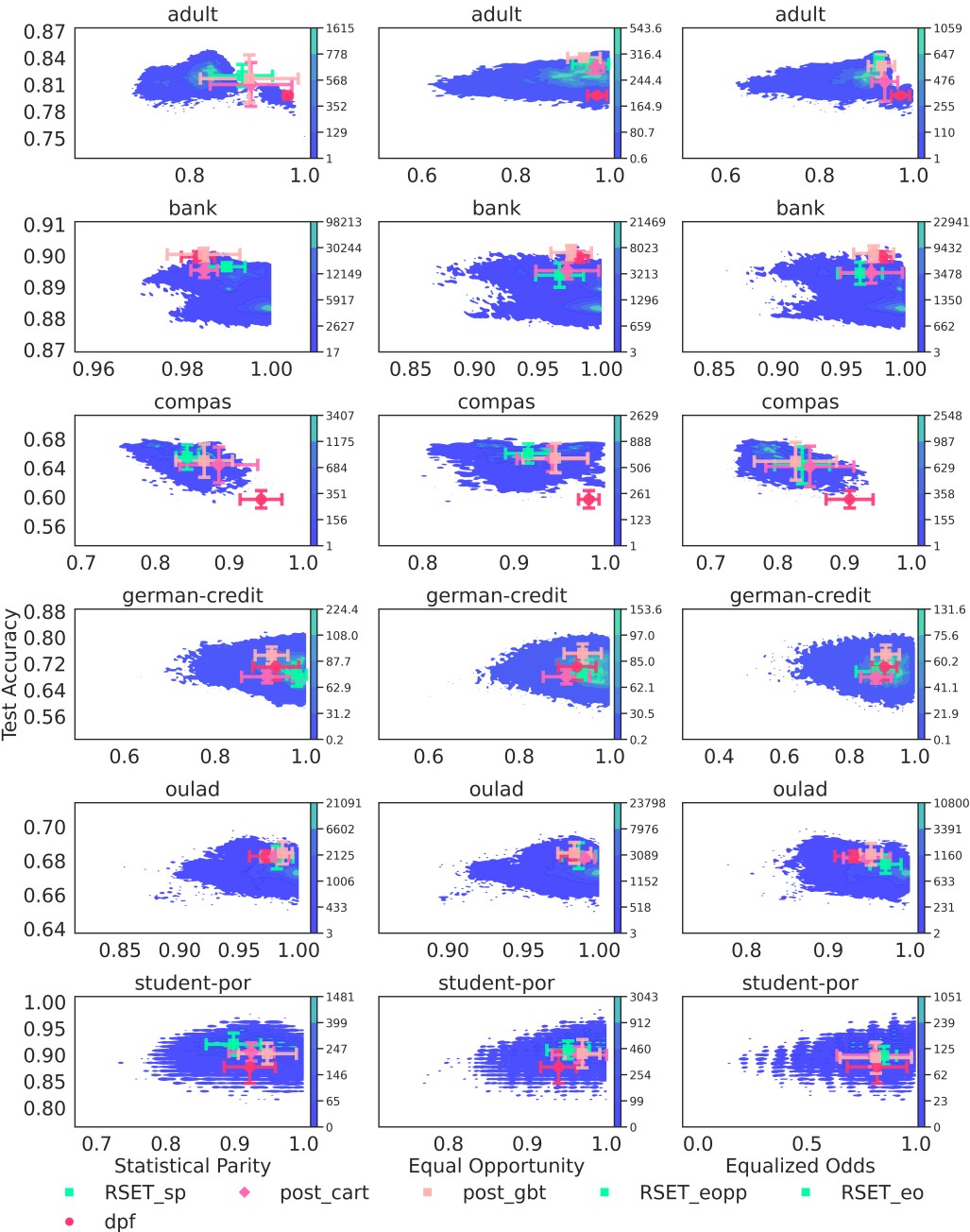

Figure 12: Model-to-model comparison of test accuracy and fairness between baselines and selected tree in the Rashomon set at depth 4.

**Training Time**    Table 33 shows the average training time of TreeFARMS (RSET), DPF, and FOCT. We discover that TreeFARMS training time is significantly shorter than FOCT. At deeper depths, training time varies across datasets – sometimes DPF takes absurdly long time to train, while some times TreeFARMS takes longer to train. Overall, we do see TreeFARMS scales better than DPF as the training time increase smaller at deeper depth. FOCT consistently shows extremely long training times across all datasets. For example, on the bank dataset, FOCT takes over 4500 seconds, whereas both DPF and TreeFARMS require less than a few seconds. This huge gap suggests that, even at shallow depths, FOCT cannot find fair trees efficiently.

**Sparsity Plots**    Lastly, as an additional study in how sparsity affects fairness, we provide an interesting visualization of this at Figure 13.

In this figure, we plot a scatter of trees and highlight the selected trees with distinct markers. While we do not observe consistent trends across datasets, we do see recurring patterns that may spark further research questions.

## D    Additional Hypothesis Space: Random Forests and FasterRisk

To support our findings with other hypothesis spaces, we conducted additional experiments on random forests for both fairness and robustness. To approximate the Rashomon set in this setting, we trained 100 random forests with different random seeds. As baselines, we used PostRF [93] for fairness (a post-hoc editing method) and GROOT-RF [83] for robustness (also a post-hoc method). We observed that this approximate Rashomon set tends to contain models that outperform the baselines in terms of both fairness and robustness.

Tables 34 and 35 show fairness and robustness comparisons for the random forest model class, respectively. "X Win Rate" refers to the proportion of models in the Rashomon set that outperform the baseline in metric X (e.g., accuracy, fairness, or robustness). "Joint Win Rate" denotes the proportion of models that outperform the baseline in both accuracy and the trustworthiness metric. Results are reported over 5 folds, with mean and standard deviation.

Table 34: Fairness comparison between the Random Forest Rashomon set constructed with different random seeds and PostRF. (Avg $\pm$ Std)

| Dataset | Joint Win Rate | Accuracy Win Rate | SP Win Rate |
|---|---|---|---|
| adult | $8.0 \pm 11.0\%$ | $60.0 \pm 21.0\%$ | $36.0 \pm 33.0\%$ |
| california-houses | $0.0 \pm 0.0\%$ | $99.0 \pm 2.0\%$ | $0.0 \pm 0.0\%$ |
| default-credit | $19.0 \pm 15.0\%$ | $43.0 \pm 9.0\%$ | $46.0 \pm 28.0\%$ |
| diabetes-130US | $44.0 \pm 24.0\%$ | $68.0 \pm 18.0\%$ | $59.0 \pm 29.0\%$ |

Table 35: Robustness comparison between the Random Forest Rashomon set and baseline. (Avg $\pm$ Std)

| Dataset | Joint Win Rate | Accuracy Win Rate | Adv. Acc. Win Rate |
|---|---|---|---|
| adult | $16.0 \pm 10.0\%$ | $100.0 \pm 0.0\%$ | $16.0 \pm 10.0\%$ |
| california-houses | $63.0 \pm 24.0\%$ | $100.0 \pm 0.0\%$ | $63.0 \pm 24.0\%$ |
| default-credit | $49.0 \pm 24.0\%$ | $75.0 \pm 33.0\%$ | $51.0 \pm 22.0\%$ |
| diabetes-130US | $1.0 \pm 2.0\%$ | $100.0 \pm 0.0\%$ | $1.0 \pm 2.0\%$ |

We note that in the california-houses dataset, we were unable to find a fair model in the approximated Rashomon set. At this point, we cannot conclude whether this is due to limitations in our approximation method or because the baseline (PostRF) is a post-hoc method that may not correspond to any near-optimal model in the Rashomon set. If this experiment had been conducted in the sparse decision tree setting, we would have had a definitive answer due to the ability to enumerate the entire Rashomon set and the availability of fairness-optimal algorithms.

We can also extend our framework to sparse linear models. However, to the best of our knowledge, there is no method that can exactly enumerate the Rashomon set for linear classifiers; only approxi-

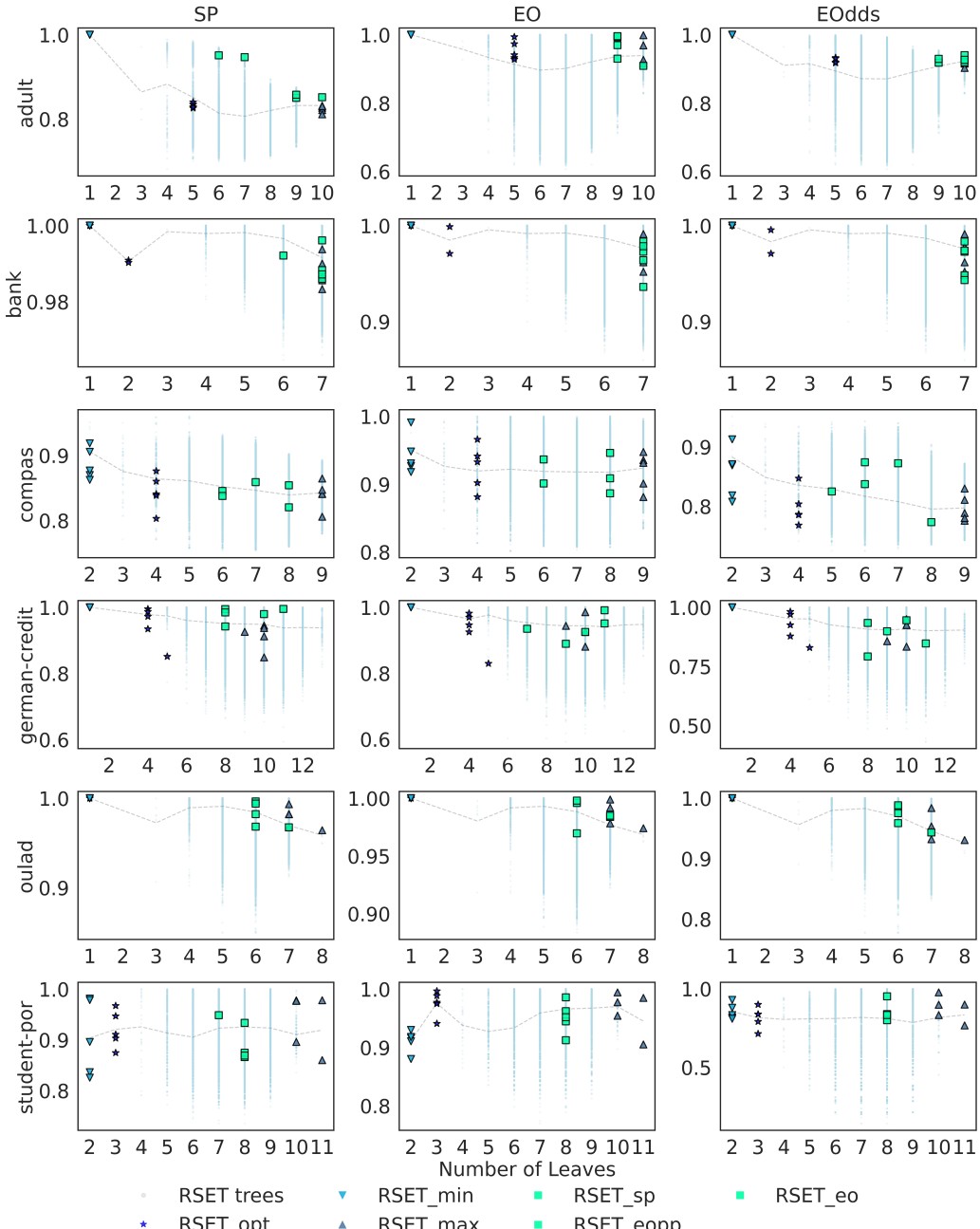

Figure 13: Sparsity plot for TreeFARMS at depth 4. Trees evaluated over five folds are concatenated. Specific trees like optimal tree, best selection min/max leaf tree, and best selection fairness tree are plotted with different markers. The y-axis is the fairness parity score and the x-axis is the number of leaves.

mators are available. For example, we use the FasterRisk algorithm [60] for sparse linear models and report the fairness results in Table 36.

Table 36: Fairness results using the FasterRisk Algorithm. (Avg $\pm$ Std)

| Dataset | Joint Win Rate | Accuracy Win Rate | SP Win Rate |
|---|---|---|---|
| california-houses | $0.0 \pm 0.0\%$ | $75.0 \pm 0.0\%$ | $0.0 \pm 0.0\%$ |
| default-credit | $7.0 \pm 12.0\%$ | $12.0 \pm 18.0\%$ | $75.0 \pm 26.0\%$ |
| diabetes-130US | $8.0 \pm 16.0\%$ | $13.0 \pm 15.0\%$ | $48.0 \pm 27.0\%$ |
| german-credit | $1.0 \pm 2.0\%$ | $4.0 \pm 6.0\%$ | $51.0 \pm 38.0\%$ |

We also observe that for california-houses we could not outperform the baseline. The limitation of the approximated Rashomon set makes empirical analysis less conclusive than in our current setting with decision trees, where the Rashomon set can be exactly enumerated.

# E Evaluation of Selected Trees Across Multiple Criteria

**Setup:** As mentioned in Section 4.5, we are interested in evaluating the performance of decision trees across various metrics. To achieve this, we consider eight datasets: adult, bank, california-houses, compas, credit-fusion, default-credit, diabetes-130US, and german-credit. These datasets are binarized using the threshold guessing procedure described in Section A.2. We train TreeFARMS after removing the sensitive feature, as required for the evaluation of fairness metrics. For robustness and privacy considerations, these omitted features do not interact with the framework. For adversarial attacks, trees are converted into continuous domain using the binarized thresholds.

**Result:** Figure 14 presents a comprehensive comparison of six trees within the Rashomon set across eight datasets, evaluated using seven different metrics discussed in this paper. For privacy metrics, we compute $1 -$ score to provide an intuitive visualization, where taller bars indicate better metric performance. Since all trees belong to the Rashomon set, they achieve comparable performance based on the test accuracy, as visible using the y-axis scale. We see that none of the trees are outside the $\pm 3\%$ range. However, we observe a sparsity-accuracy tradeoff within the trees with minimum and maximum number of leaves on many datasets such as adult, bank, and compas. In terms of adversarial accuracy, we still see that our approach in choosing the best adversarial tree (gray bar) on the validation set is always the best performer except on the german-credit dataset.

For statistical (demographic) parity, we find that the selected fair tree performs well on multiple datasets (for example, see adult and california-houses). Interestingly, these fair trees selected using the validation set often transfer across the fairness metrics (for example, see bank and default-credit). We also observe that the tree with minimum leaves (RSET_min) can be fair, but this observation is not consistent across all datasets. Finally, regarding privacy metrics, we observe that most trees perform similarly around $50\%$. As discussed in Section 4.1, these attacks are generally ineffective against tree-based models.

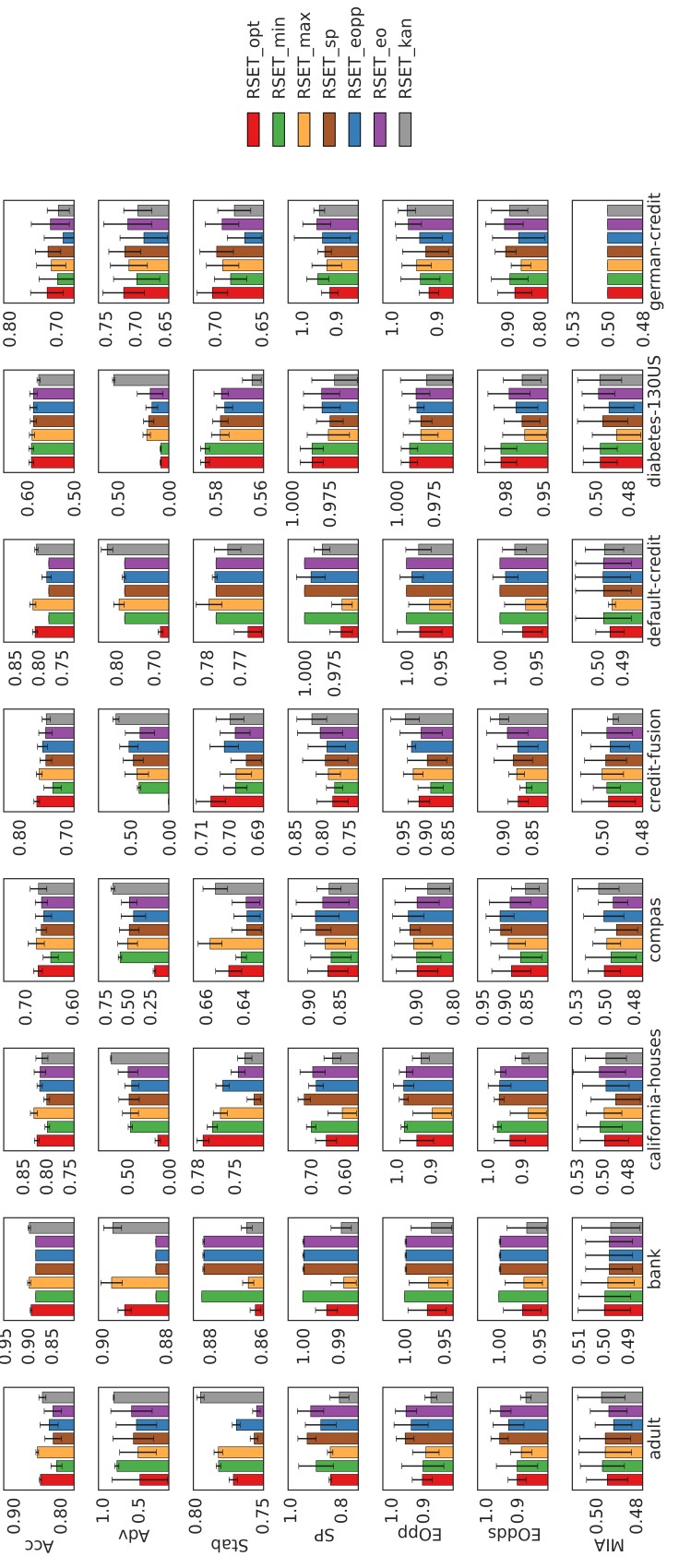

Figure 14: Bar plots of metric accuracy on multiple datasets. Metrics from top to bottom are Test Accuracy, Test Adversarial Accuracy, Stability, Statistical Parity, Equal Opportunity, Equalized Odds, and 1 - shadow model MI attack

