# OpenReview forum: "The Rashomon Set Has It All: Analyzing Trustworthiness of Trees under Multiplicity"
_NeurIPS.cc/2025/Datasets_and_Benchmarks_Track — NeurIPS 2025 Datasets and Benchmarks Track poster_

### Official Review · Reviewer_N9bH · 2025-06-18

**Rating:** 6
**Confidence:** 3

**Summary:**

This paper presents a systematic analysis of multiple trustworthiness criteria—including robustness, privacy, and fairness—in the context of sparse decision trees within the Rashomon set. Through extensive experiments on a variety of datasets, the authors demonstrate that the Rashomon set of sparse decision trees contains models that naturally exhibit desirable trustworthy properties. Moreover, some of these models achieve comparable levels of trustworthiness to those explicitly optimized for such properties.

**Dataset Code Accessibility:**

Yes

**Dataset Code Comments:**

The authors have provided both the source code and the corresponding datasets, which facilitates reproducibility and further research based on their work.

**Ethical Considerations:**

No, there are no or only very minor ethics concerns

**Final Justification:**

I recommend a score of 6 for this paper.

The authors have addressed my concern regarding the separation of model-intrinsic properties and the added value introduced by the Rashomon effect. They included experiments that isolate the impact of the Rashomon effect, which strengthens the paper's findings.

I acknowledge the academic contribution of the authors' work, particularly how the diversity of the Rashomon set contributes to trustworthy properties.

From an application perspective, the authors clarified the method's value in tabular datasets. However, sparse decision trees may be less effective with more complex data, and current methods for Rashomon set construction in other hypothesis spaces are still limited, restricting the approach's broader applicability.

**Limitations Weaknesses:**

1. One limitation of the paper is that it does not clearly separate the effects of model-intrinsic properties—such as the inherent robustness of sparse decision trees—from the added value introduced by the Rashomon effect in improving trustworthiness. It would strengthen the paper to include a more explicit discussion on the isolated impact of the Rashomon effect. Specifically, in experiments on robustness, MIA, and fairness, including standard sparse decision trees as a baseline would help empirically demonstrate the added benefit of the Rashomon effect beyond what the sparse decision tree alone provides.
2. A minor writing suggestion: in line 292, it would be helpful to clarify that “MIA” stands for “membership inference attack,” to aid readers who may not be familiar with the acronym.
3. In Section 3.3.2, a brief introduction to the task of machine unlearning would improve readability and help contextualize the experimental setup for readers less familiar with this area.

**Strengths Contributions:**

1. This paper significantly broadens the application scope of the Rashomon set in the domain of trustworthy machine learning. While prior work has primarily focused on fairness or sparsity individually, this study systematically investigates a wider range of trustworthiness criteria, including robustness, privacy, and fairness.
2. A key contribution of this work lies in its empirical finding that the Rashomon set of sparse decision trees naturally contains models exhibiting multiple trustworthy properties. This insight suggests a promising alternative to explicitly optimizing for trustworthiness: selecting models with desirable properties directly from the Rashomon set.
3. The observations presented in Section 4.5 are particularly noteworthy. The authors propose an efficient strategy for identifying models that satisfy multiple trustworthiness requirements. Compared to designing algorithms that simultaneously optimize for several criteria, the approach of selecting suitable models from the Rashomon set appears simpler and more computationally efficient.
4. Overall, the paper is well-written and logically structured, which makes it easy to follow. The figures and tables are informative and effectively support the paper’s key messages.

---

> ### Author Rebuttal · Authors · 2025-07-30
>
> Thank you very much for your review and feedback.
>
> > Q: Separation of model-intrinsic properties and added value of multiplicity.
>
> Thank you for pointing this out. It is true that for some trustworthy metrics, such as privacy, our results benefit from the hypothesis space of sparse decision trees. Since the optimal model is quite small, it tends to be more private compared to complex models, such as gradient boosted trees, for example. However, for fairness and robustness, sparsity of the optimal model—or, in fact, of any model—will not be sufficient. In these cases, there are greater benefits from the existence of a diverse Rashomon set, as we can choose models that better align with other objectives. Figures 1, 3, 4, 5, Tables 3, 6-8 contain results for the optimal trees where this point can be observed.
>
> We will make these points more explicit in the manuscript for clarity. Thank you for bringing this up to our attention.
>
> > Q: Clarifying MIA in line 292
>
> Thank you for pointing this out. We will revise the text at line 292 to clarify that 'MIA' stands for 'membership inference attack' to ensure clarity for all readers.
>
> > Q: Brief intro to the machine unlearning
>
> Sure, we will add a brief description about machine unlearning at the beginning of Section 3.3.2.

---

> > ### Comment · Reviewer_N9bH · 2025-08-02
> >
> > Thanks for the authors’ response. The concerns I raised have been addressed.
> >
> > Additionally, I have read the feedback from other reviewers. They expressed concerns regarding the model type limitations. The authors have provided a clear explanation for why the analysis was conducted using sparse decision trees—due to the tabular dataset format, the selection of comparison algorithms is easier, and the Rashomon Set can be constructed  via full enumeration.
> >
> > I acknowledge the academic contribution of the authors’ research. Their work demonstrates that the diversity of the Rashomon set can directly contribute to various trustworthy properties.
> >
> > From an application perspective, the authors’ response clarifies the value of this method in tabular datasets. However, when dealing with more complex data, sparse decision trees may have limited effectiveness. In such cases, more complex models may be required (e.g., deep learning models). However, it seems that the existing methods for Rashomon set construction in other hypothesis spaces remain relatively limited, which somewhat restricts the applicability of the approach presented in the paper.

---

> > > ### Author Response · Authors · 2025-08-02
> > >
> > > Thank you very much for your thoughtful response. We appreciate your recognition of our work’s contribution and constructive comments on its applicability.

---

### Official Review · Reviewer_va16 · 2025-07-01

**Rating:** 5
**Confidence:** 4

**Summary:**

This article focuses on the Rashomon effect in machine learning and suggest that leveraging the Rashomon set for model selection can be effective and reliable for building ML systems.

**Dataset Code Accessibility:**

Yes

**Ethical Considerations:**

No, there are no or only very minor ethics concerns

**Final Justification:**

My concerns are generally addressed by the third point and Reviewer N9bH's comments, and as a result I raised my rating.

**Limitations Weaknesses:**

My primary concern relates to the generalizability of the findings beyond decision trees. I do wonder how much of these findings generalize beyond decision trees. The whole framework is built around sparse trees, and while that’s fine for this paper, it’s a pretty limited class of models. It’s not clear if you’d see similar results for more powerful models like neural networks or ensembles—which are common in practice. The paper briefly mentions this at the end, but I think it deserves more discussion.

Also, since TreeFARMS is designed specifically for decision trees, it’s not obvious how the approach could be adapted to other model types. If the whole thing only works for trees, that’s fine, but then the broader message about Rashomon sets being a good source of trustworthy models might be a bit overstated.

**Strengths Contributions:**

- The idea is interesting and the presentation is good, I enjoy reading it.
- The authors adequate anecdote experiments.
- The results are significant, by my judgement.

---

> ### Author Rebuttal · Authors · 2025-07-30
>
> Thank you very much for your review and feedback.
>
> There are a few important reasons for our choice of the hypothesis space of sparse decision trees.
>
> First, for the tabular dataset, sparse decision trees perform quite well, as good as black box models (please see McTavish et al., 2022 and Grinsztajn et al., 2022 for more details).
>
> Second, because of the logical structure of trees and the popularity of this hypothesis space, the research community has developed many algorithms for trees that directly optimize trustworthy objectives while providing guarantees of global optimality. For other hypothesis spaces, trustworthy algorithms are often greedy or post-hoc.
>
> Third, and most crucially, sparse decision trees currently represent the only hypothesis space for which the Rashomon set can be constructed exactly via full enumeration. For other model classes such as random forests, neural networks, and generalized additive models, only approximate methods are available to explore the Rashomon set. These approximations do not guarantee completeness, which can be problematic: if a trustworthy model is not found in the approximated Rashomon set, it is unclear whether this is due to the absence of such a model or due to approximation error. Therefore, we focused on sparse decision trees and TreeFARMS to allow conclusive and rigorous statements about the Rashomon set.
>
> However, to support our findings with additional hypothesis space, we conducted additional experiments on random forests for both fairness and robustness. To approximate the Rashomon set in this setting, we trained 100 random forests with different random seeds. As baselines, we used PostRF [1] for fairness (a post-hoc editing method) and GROOT-RF [2] for robustness (also a post-hoc method). We observed that this approximate Rashomon set tends to contain models that outperform the baselines in terms of both fairness and robustness.
>
> Please see the additional results in the tables below. "X Win Rate" refers to the proportion of models in the Rashomon set that outperform the baseline in metric X (e.g., accuracy, fairness, or robustness). "Joint Win Rate" denotes the proportion of models that outperform the baseline in both accuracy and the trustworthiness metric. Results are reported over 5 folds, with mean and std.
>
> Table 1: Fairness comparison between the Random forest Rashmon set constructed with different random seeds and PostRF.
> | Dataset            | Joint Win Rate       | Accuracy Win Rate (Avg ± Std) | Fairness Win Rate (Avg ± Std) |
> |--------------------|----------------------|-------------------------------|-----------------------------|
> | adult              | 0.08 ± 0.11          | 0.60 ± 0.21                    | 0.36 ± 0.33                   |
> | california-houses  | 0.00 ± 0.00          | 0.99 ± 0.02                   | 0.00 ± 0.00                   |
> default-credit     | 0.19 ± 0.15          | 0.43 ± 0.09                   | 0.46 ± 0.28                   |
> | diabetes-130US     | 0.44 ± 0.24          | 0.68 ± 0.18                   | 0.59 ± 0.29                   |
>
> Table 2: Robustness comparison between the Random Forest Rashomon set and baseline
> | Dataset                | Joint Win Rate       | Accuracy Win Rate (Avg ± Std) | Adversarial Accuracy Win Rate (Avg ± Std) |
> |------------------------|----------------------|-------------------------------|-------------------------------|
> | adult                  | 0.16 ± 0.10          | 1.00 ± 0.00                   | 0.16 ± 0.10                   |
> | california-houses      | 0.63 ± 0.24          | 1.00 ± 0.00                   | 0.63 ± 0.24                   |
> | default-credit         | 0.49 ± 0.24          | 0.75 ± 0.33                   | 0.51 ± 0.22                   |
> | diabetes-130US         | 0.01 ± 0.02          | 1.00 ± 0.00                   | 0.01 ± 0.02                   |
>
> We note that in the california-houses dataset, we were unable to find a fair model in the approximated Rashomon set. At this point, we cannot conclude whether this is due to limitations in our approximation method or because the baseline (PostRF) is a post-hoc method that may not correspond to any near-optimal model in the Rashomon set. If this experiment had been conducted in the sparse decision tree setting, we would have had a definitive answer due to the ability to enumerate the entire Rashomon set and the availability of fairness-optimal algorithms.
>
> It is true that TreeFARMS is specifically designed for decision trees. However, our goal is to use TreeFARMS as a case study to investigate a hypothesis that the Rashomon set often contains models with favorable trustworthiness properties. While TreeFARMS itself is not directly applicable to other model classes, the framework we use for empirical analysis can be extended to other model classes for which the Rashomon set can be found.
>
> [1] Ruicheng Xian and Han Zhao. A Unified Post-Processing Framework for Group Fairness, 2024.
>
> [2] Daniël Vos and Sicco Vewer. Efficient training of robust decision trees against adversarial examples. 2021.

---

> > ### Comment · Reviewer_va16 · 2025-08-07
> > **Reply**
> >
> > Thanks for the rebuttal. My concerns are generally addressed by the third point and Reviewer N9bH's comments, and as a result I raised my rating.

---

> > > ### Author Response · Authors · 2025-08-07
> > >
> > > Thank you for the discussion and comments. We sincerely appreciate your consideration in raising the score.

---

### Official Review · Reviewer_kPVh · 2025-07-06

**Rating:** 5
**Confidence:** 3

**Summary:**

In this work, the authors study the Rashomon set of decision trees, and compare the performance of trees in the set along different axes such as accuracy and metrics of trustworthiness (robustness, fairness, privacy, etc.). They find the trees in the Rashomon set often outperform or perform on par with baselines that are explicitly optimized for trustworthiness. They show that this framework can be utilized for benchmarking trustworthiness of interpretable models.

**Dataset Code Accessibility:**

Yes

**Dataset Code Comments:**

This work uses public datasets, and code is available

**Ethical Considerations:**

No, there are no or only very minor ethics concerns

**Final Justification:**

The authors have addressed my main concerns (generalizability to other models) in the rebuttal. The results on the california-houses dataset are interesting and worth further exploration, but does not outweigh the benefits of accepting the paper in my opinion.

**Limitations Weaknesses:**

- It is unclear if the results are generalizable to models apart from decision trees. For example, consider linear classifiers where sparsity corresponds to number of features. Is it possible to identify Rashomon sets in this manner?
- It is unclear what the loss (train loss) for the models explicitly optimized for trustworthiness are -- can authors shed some light on this?
- The results in Table 2 are confusing: can authors clarify how predictions for trees in Rashomon set remain unaffected?

**Strengths Contributions:**

- Paper is well-written, easy to follow, and has several intuitive results (some models in the Rashomon set being fairer, following results from prior work on model multiplicity etc)
- Some results are surprising: such as sparser models being more trustworthy, some models in the Rashomon slightly outperforming models optimized explicitly for fairness (Figure 2)
- The benchmarking approach makes sense, and the authors have studied different axes of trustworthiness

---

> ### Author Rebuttal · Authors · 2025-07-30
>
> Thank you very much for your review and feedback.
>
> > Q: It is unclear if the results are generalizable to models apart from decision trees.
>
> Yes, we can use other hypothesis spaces, however then we will have to approximate the Rashomon set as there is no other algorithm known to us besides TreeFarns, that computes the Rashomon set completely.
>
> To support our findings with additional hypothesis space, we conducted additional experiments on random forests for both fairness and robustness. To approximate the Rashomon set in this setting, we trained 100 random forests with different random seeds. As baselines, we used PostRF [1] for fairness (a post-hoc editing method) and GROOT-RF [2] for robustness (also a post-hoc method). We observed that this approximate Rashomon set tends to contain models that outperform the baselines in terms of both fairness and robustness.
>
> Please see the additional results in the tables below. "X Win Rate" refers to the proportion of models in the Rashomon set that outperform the baseline in metric X (e.g., accuracy, fairness, or robustness). "Joint Win Rate" denotes the proportion of models that outperform the baseline in both accuracy and the trustworthiness metric. Results are reported over 5 folds, with mean and std.
>
> Table 1: Fairness comparison between the Random forest Rashmon set constructed with different random seeds and PostRF.
> | Dataset            | Joint Win Rate       | Accuracy Win Rate (Avg ± Std) | Fairness Win Rate (Avg ± Std) |
> |--------------------|----------------------|-------------------------------|-----------------------------|
> | adult              | 0.08 ± 0.11          | 0.60 ± 0.21                    | 0.36 ± 0.33                   |
> | california-houses  | 0.00 ± 0.00          | 0.99 ± 0.02                   | 0.00 ± 0.00                   |
> default-credit     | 0.19 ± 0.15          | 0.43 ± 0.09                   | 0.46 ± 0.28                   |
> | diabetes-130US     | 0.44 ± 0.24          | 0.68 ± 0.18                   | 0.59 ± 0.29                   |
>
> Table 2: Robustness comparison between the Random Forest Rashomon set and baseline
> | Dataset                | Joint Win Rate       | Accuracy Win Rate (Avg ± Std) | Adversarial Accuracy Win Rate (Avg ± Std) |
> |------------------------|----------------------|-------------------------------|-------------------------------|
> | adult                  | 0.16 ± 0.10          | 1.00 ± 0.00                   | 0.16 ± 0.10                   |
> | california-houses      | 0.63 ± 0.24          | 1.00 ± 0.00                   | 0.63 ± 0.24                   |
> | default-credit         | 0.49 ± 0.24          | 0.75 ± 0.33                   | 0.51 ± 0.22                   |
> | diabetes-130US         | 0.01 ± 0.02          | 1.00 ± 0.00                   | 0.01 ± 0.02                   |
>
> We note that in the california-houses dataset, we were unable to find a fair model in the approximated Rashomon set. At this point, we cannot conclude whether this is due to limitations in our approximation method or because the baseline (PostRF) is a post-hoc method that may not correspond to any near-optimal model in the Rashomon set. If this experiment had been conducted in the sparse decision tree setting, we would have had a definitive answer due to the ability to enumerate the entire Rashomon set and the availability of fairness-optimal algorithms.
>
> > Q: Is it possible to identify Rashomon sets for linear classifiers with l0 sparsity.
>
> We can also extend our framework to sparse linear models. However, to the best of our knowledge, there is no method that can exactly enumerate the Rashomon set for linear classifiers; only approximators are available. For example, we use the FasterRisk algorithm (Liu et al., 2022) for sparse linear models and report the fairness results in the Table below.
>
> | Dataset              | Joint Win Rate       | Accuracy Win Rate (Avg ± Std) | Fairness Win Rate (Avg ± Std) |
> |----------------------|----------------------|-------------------------------|-------------------------------|
> | german-credit        | 0.01 ± 0.02          | 0.04 ± 0.06                   | 0.51 ± 0.38                   |
> |diabetes-130US       | 0.08 ± 0.16          | 0.13 ± 0.15                   | 0.48 ± 0.27                   |
> | default-credit       | 0.07 ± 0.12          | 0.12 ± 0.18                   | 0.75 ± 0.26                   |
> | california-houses    | 0.00 ± 0.00          | 0.75 ± 0.00                   | 0.00 ± 0.00                   |
>
> We also observe that for california-houses we couldn’t outperform the baseline. The limitation of the approximated Rashomon set makes empirical analysis less conclusive than in our current setting with decision trees, where the Rashomon set can be exactly enumerated.
>
> > Q: It is unclear what the loss (train loss) for the models explicitly optimized for trustworthiness are -- can authors shed some light on this?
>
> Different trustworthy properties require us to solve different specialized optimization problems. For example, fairness-aware models usually minimize 0-1 loss subject to constraints such as statistical parity [3], e.g.,
> $$\min_{f}1[f(X)\neq y]\ \  s.t.\ \  |P(f(X)=1|y=1, A=a) - P(f(X)=1|y=1, A=b)| \leq \delta,$$ where $A$ denotes sensitive features. This optimization problem is known to be NP-hard.
>
> For robustness, the optimal robust tree algorithm [4] solves
> $$\min_f \sum_{(x, y)\sim D} \max_{\delta \in S} 1[f(x+\delta)\neq y],$$ where $S$ is the perturbation domain and a MIP solver is used to solve this problem.
> We will provide a detailed explanation of these training losses in the final version of the paper.
>
> > Q: The results in Table 2 are confusing: can authors clarify how predictions for trees in Rashomon set remain unaffected?
>
> The results in Table 2 follow directly from a theoretical guarantee in TreeFARMS (Theorem 5.3 in [5]), which shows that if the Rashomon set is constructed with a tolerance $\epsilon \geq 2K/n$, where $K$ is the number of removed points and $n$ is the original data size, then the optimal tree trained after unlearning remains within the original Rashomon set. With a properly chosen $\epsilon$, there is no need to retrain the model from scratch. Instead, we can simply select the tree from the original Rashomon set that minimizes loss on the reduced dataset. Since this selected tree is exactly the same as the one that would have been trained after removal, their predictions on the test set are identical. This explains the 0% disagreement rate between the unlearned and retrained models shown in Table 2.
>
>
> [1] Ruicheng Xian and Han Zhao. A Unified Post-Processing Framework for Group Fairness, 2024.
>
> [2] Daniël Vos and Sicco Verwer. Efficient training of robust decision trees against adversarial examples. 2021.
>
> [3] Jacobus van der Linden, Mathijs de Weerdt, and Emir Demirovic´. Fair and optimal decision trees: A dynamic programming approach. 2022.
>
> [4] Daniël Vos and Sicco Verwer. Robust optimal classification trees against adversarial examples. 2022.
>
> [5] Rui Xin, Chudi Zhong, Zhi Chen, Takuya Takagi, Margo Seltzer, and Cynthia Rudin. Exploring the whole rashomon set of sparse decision trees. 2022.

---

> > ### Comment · Reviewer_kPVh · 2025-08-06
> >
> > Thanks to the authors for the additional experiments! I will keep my current score.
> >
> > The results on the california-houses dataset are interesting and worth further exploration.
> >
> > Regarding my comment on loss (train loss) -- I understand that loss forms during optimization might vary, but was wondering about the train performance in general (for example, measured via a standard metric like train accuracy). Essentially, how well the chosen model in each case underfits/overfits the train dataset.

---

> > > ### Author Response · Authors · 2025-08-07
> > >
> > > Thank you!
> > >
> > > We will explore the california-houses dataset more. Overall (if this is not an approximation issue), different data distributions/datasets can lead to different landscapes of the Rashomon set, which is an interesting research direction by itself. We will include discussion on this in the paper and also add open questions for future, more fundamental research.
> > >
> > > Regarding train loss, for both Rashomon set models and trustworthy-specialized optimized models, we saw a small gap between train and test error. Basically, the sparse trees generalize quite well on the tabular datasets that we used. We will add the training loss results to the appendix to provide complete information.

---

### Official Review · Reviewer_My5Y · 2025-07-13

**Rating:** 4
**Confidence:** 2

**Summary:**

A systematic analysis of different properties across models in the Rashomon set (the set of near-optimal models) has not been previously conducted. To address this gap, this paper introduces a framework that evaluates and compares the performance of models (sparse  decision trees) from the Rashomon set against those explicitly optimized for a trustworthiness criterion.
The results demonstrate that these near-optimal decision trees from the Rashomon set can perform comparably to trees explicitly optimized for trustworthiness criteria.

**Dataset Code Accessibility:**

Yes

**Ethical Considerations:**

No, there are no or only very minor ethics concerns

**Final Justification:**

The core innovation lies in the new empirical findings that the Rashomon set of models is a practical resource for trustworthiness. It would be valuable to explore whether the findings generalize to other model classes.

**Limitations Weaknesses:**

Weaknesses:

W1: The innovation is limited. The paper doesn't introduce new dataset or new theoretical developments or algorithmic improvements. The framework primarily repackages existing methods into an analysis pipeline. This lowers the novelty of the work.

W2: As shown in Section 4.4, selecting a single trustworthy model from millions in the Rashomon set remains computationally expensive: “...the selection of a single model might be costly.” This undermines the practical utility of the framework for real-world deployment.

W3: The framework is designed for sparse decision trees. While interpretable, these models are limited in representational power compared to more widely deployed architectures such as ensembles or deep neural networks. This constrains the generalizability and relevance of the findings.

**Strengths Contributions:**

Strengths:

S1: The paper is well-organized and clearly written.

S2: It proposes a comprehensive evaluation framework to evaluate sparse decision trees within Rashomon set across robustness (adversarial and stability), privacy (membership inference and unlearning), and fairness (statistical parity, equalized odds, and equal opportunity).

S3: This work conducted a comprehensive empirical analysis across 21 datasets. The results demonstrate that these near-optimal decision trees from the Rashomon set can perform comparably to trees explicitly optimized for trustworthiness criteria.

---

> ### Author Rebuttal · Authors · 2025-07-30
>
> Thank you very much for your review and feedback.
>
> > Q: The innovation is limited.
>
> The core innovation lies in the new empirical findings and insights that model diversity is a practical resource for trustworthiness. Our key contribution is to show, for the first time, that the Rashomon set of sparse decision trees often already contains models that are comparably robust, fair, and private to those produced by state-of-the-art methods specifically optimized for those trustworthiness criteria. This is not an obvious result and directly challenges the common belief that specialized optimization is necessary to achieve each trust goal. The framework we use is not merely a wrapper around existing methods. It enables systematic, multi-metric benchmarking of trustworthiness. This framework is modular and can be extended to other model classes or even more complex models in future research once appropriate methods exist to construct the corresponding Rashomon sets. We believe that our findings fit in the scope of the track and provide valuable insights as well as the framework for future research.
>
> > Q: Selecting a model from the Rashomon set is computationally expensive.
>
> Thank you for pointing this out. We apologize for the ambiguity in our original wording. Our intent was not to suggest that the framework is computationally prohibitive, but to be transparent that evaluating all models in a large Rashomon set can take time proportional to the number of models. However, it is important to note that:
> 1. Enumerating the Rashomon set is typically faster than training a single optimal robust or fair tree, especially for modern exact optimization methods (e.g., MIP solvers). This is demonstrated in Section 4.4.
> 2. Evaluating each tree is fast, due to the simplicity of decision trees and the low computational cost of most trustworthiness metrics. While looping through a very large Rashomon set can take time proportional to its size, this cost is linear and can be reduced through subsampling or parallelization. It remains far more efficient than retraining new models from scratch.
> 3. With the Rashomon set, no retraining is required. Once the set is constructed, selecting a model that aligns with a trustworthiness criterion becomes a post hoc process. In contrast, optimization-based methods (e.g., optimal robust or fair trees) require a full retraining cycle if the initial model does not satisfy user needs, which can be very time-consuming.
>
> In summary, while selecting from an enormous Rashomon set can take time, the overall approach is significantly more practical and efficient than repeatedly retraining specialized models. We will revise the text in Section 4.4 to clarify this practical advantage.
>
> > Q: The framework is designed for sparse decision trees... This constrains the generalizability and relevance of the findings.
>
> There are a few important reasons for our choice of the hypothesis space.
>
> First, for the tabular dataset, sparse decision trees perform quite well, as good as black box models (please see McTavish et al., 2022 and Grinsztajn et al., 2022 for more details).
>
> Second, because of the logical structure of trees and the popularity of this hypothesis space, the research community has developed many algorithms for trees that directly optimize trustworthy objectives while providing guarantees of global optimality. For other hypothesis spaces, trustworthy algorithms are often greedy or post-hoc.
>
> Third, and most crucially, sparse decision trees currently represent the only hypothesis space for which the Rashomon set can be constructed exactly via full enumeration. For other model classes such as random forests, neural networks, and generalized additive models, only approximate methods are available to explore the Rashomon set. These approximations do not guarantee completeness, which can be problematic: if a trustworthy model is not found in the approximated Rashomon set, it is unclear whether this is due to the absence of such a model or due to approximation error. Therefore, we focused on sparse decision trees and TreeFARMS to allow conclusive and rigorous statements about the Rashomon set.
>
> However, to support our findings with additional hypothesis space, we conducted additional experiments on random forests for both fairness and robustness. To approximate the Rashomon set in this setting, we trained 100 random forests with different random seeds. As baselines, we used PostRF [1] for fairness (a post-hoc editing method) and GROOT-RF [2] for robustness (also a post-hoc method). We observed that this approximate Rashomon set tends to contain models that outperform the baselines in terms of both fairness and robustness.
>
> Please see the additional results in the tables below. "X Win Rate" refers to the proportion of models in the Rashomon set that outperform the baseline in metric X (e.g., accuracy, fairness, or robustness). "Joint Win Rate" denotes the proportion of models that outperform the baseline in both accuracy and the trustworthiness metric. Results are reported over 5 folds, with mean and std.
>
> Table 1: Fairness comparison between the Random forest Rashmon set constructed with different random seeds and PostRF.
> | Dataset            | Joint Win Rate       | Accuracy Win Rate (Avg ± Std) | Fairness Win Rate (Avg ± Std) |
> |--------------------|----------------------|-------------------------------|-----------------------------|
> | adult              | 0.08 ± 0.11          | 0.60 ± 0.21                    | 0.36 ± 0.33                   |
> | california-houses  | 0.00 ± 0.00          | 0.99 ± 0.02                   | 0.00 ± 0.00                   |
> default-credit     | 0.19 ± 0.15          | 0.43 ± 0.09                   | 0.46 ± 0.28                   |
> | diabetes-130US     | 0.44 ± 0.24          | 0.68 ± 0.18                   | 0.59 ± 0.29                   |
>
> Table 2: Robustness comparison between the Random Forest Rashomon set and baseline
> | Dataset                | Joint Win Rate       | Accuracy Win Rate (Avg ± Std) | Adversarial Accuracy Win Rate (Avg ± Std) |
> |------------------------|----------------------|-------------------------------|-------------------------------|
> | adult                  | 0.16 ± 0.10          | 1.00 ± 0.00                   | 0.16 ± 0.10                   |
> | california-houses      | 0.63 ± 0.24          | 1.00 ± 0.00                   | 0.63 ± 0.24                   |
> | default-credit         | 0.49 ± 0.24          | 0.75 ± 0.33                   | 0.51 ± 0.22                   |
> | diabetes-130US         | 0.01 ± 0.02          | 1.00 ± 0.00                   | 0.01 ± 0.02                   |
>
> We note that in the california-houses dataset, we were unable to find a fair model in the approximated Rashomon set. At this point, we cannot conclude whether this is due to limitations in our approximation method or because the baseline (PostRF) is a post-hoc method that may not correspond to any near-optimal model in the Rashomon set. If this experiment had been conducted in the sparse decision tree setting, we would have had a definitive answer due to the ability to enumerate the entire Rashomon set and the availability of fairness-optimal algorithms.
>
> Also please note that while our current experiments focus on sparse decision trees, components in this framework (Kanch attack, stability, membership inference attack, fairness evaluation) can be reused or modified and extended to other model classes for future study. So, we will also refactor the code into a modular structure to facilitate future extensions and adaptations.
>
> [1] Ruicheng Xian and Han Zhao. A Unified Post-Processing Framework for Group Fairness, 2024.
>
> [2] Daniël Vos and  Sicco Verwer. Efficient training of robust decision trees against adversarial examples. In: International conference on machine learning. 2021.

---

> ### Comment · Reviewer_My5Y · 2025-08-04
>
> Thank you for your detailed response. I appreciate the core innovation lies in the **new empirical findings** that Rashomon set of models is a practical resource for trustworthiness.
>
> However, I still have a key concern: it’s unclear whether the empirical findings generalize to other model classes like neural networks. Because Rashomon set can only be fully enumerated for decision trees, other types of methods cannot adopt similar approaches.
>
> Also, while selecting from a precomputed Rashomon set is more efficient than retraining, constructing the set itself can be costly.

---

> > ### Author Response · Authors · 2025-08-04
> >
> > Thank you for engaging in discussion with us! We appreciate your time and response.
> >
> > > It’s unclear whether the empirical findings generalize to other model classes like neural networks
> >
> > We would like to make three comments regarding the space of NN:
> >
> > - We expect the empirical **findings in our paper to hold for the Rashomon sets or their approximations that contain diverse models**. By definition, the Rashomon set consists of many models that perform similarly well on the training objective. We expect that for Rashomon sets that can differ significantly in their structure and behavior, this diversity leads to the existence of well-performing models with desirable trustworthiness properties. So as long as the resulting Rashomon set contains models with different classification patterns or classification logic, we expect empirical findings to hold for neural networks as well.
> > However, if the neural networks are often overparameterized, one can end up with the Rashomon set of many models that perform well using the same logic. So even if the Rashomon set is large, it might not be very diverse in how the models behave. That makes it harder to find models with different, potentially more trustworthy properties just by searching the set. In these cases, it may be more effective to train models with trustworthiness in mind from the start.
> > To summarize, the empirical findings of our paper will be applicable depending on the way that the Rashomon set is approximated. *We'd like to use our results to encourage more research into the diversity of the Rashomon set, especially for complex models classes and we add this to the paper discussion.*
> > -  **We will make our framework even more modular and flexible** so that it would be easier to use and extend as new methods to measure/approximate the Rashomon set occur. We already showed that our framework is adaptable to other model classes such as random forests, and we have included preliminary results in our previous comments to demonstrate this.
> > - We also want to emphasize that **inherently interpretable models have distinct advantages for real-world applications, especially in high-stakes domains where transparency is critical**. Sparse decision trees, in particular, are widely used. They can handle most tabular datasets and have been successfully applied in diverse real-world settings, including stroke prediction [1], metamaterials analysis [2], HIV research [3], and lab tests [4]. Focusing on sparse decision trees allows us to study this phenomenon rigorously while maintaining high practical relevance.
> >
> >
> > > While selecting from a precomputed Rashomon set is more efficient than retraining, constructing the set itself can be costly.
> >
> > As we showed in Table 4 **constructing the Rashomon set is at least 7 times faster than optimizing directly for robustness criteria**. As we explained above, this is due to the efficiency of TreeFARMS. It runs in seconds and was improved after publication to be even faster. We honestly don’t think that at this moment there exists a faster algorithm for constructing the Rashoon set. It’s much faster than some optimal decision tree algorithms that use MIP, for example. Also, real-world model development is inherently iterative, and the Rashomon set supports this process by enabling users to efficiently explore diverse, high-performing models without repeated retraining.
> >
> > [1] Interpretable Classifiers Using Rules and Bayesian Analysis: Building a Better Stroke Prediction Model, AOAS 2015
> >
> > [2] How to See Hidden Patterns in Metamaterials with Interpretable Machine Learning, Extreme Mechanics Letters 2022.
> >
> > [3] Impact of cannabis use on immune cell populations and the viral reservoir in people with HIV on suppressive antiretroviral therapy, JID 2023
> >
> > [4] Urgency Prediction for Medical Laboratory Tests Through Optimal Sparse Decision Tree: Case Study With Echocardiograms. JMIR AI, 2025.

---

> > > ### Comment · Reviewer_My5Y · 2025-08-07
> > >
> > > Thank you very much for your response. I understand constructing the Rashomon set is faster than optimizing directly for robustness criteria. I still have concerns regarding the limitations of the model types.
> > >
> > > I will raise my score for the authors' careful response, and I look forward to seeing the framework become even more modular and flexible.

---

> > > > ### Author Response · Authors · 2025-08-07
> > > >
> > > > Thank you for your thoughtful discussion and for raising the score. We sincerely appreciate it.

---

### Official Review · Reviewer_gToP · 2025-07-14

**Rating:** 4
**Confidence:** 4

**Summary:**

The paper considers the Rashomon set of (sparse) decision trees -- that is, the set of all (sparse) decision trees that perform equally well (w.r.t. accuracy) on a given data set. The paper empirically studies several different properties, linked to the trustworthiness of the trees, within the Rashomon set -- namely robustness, privacy, unlearning, and fairness. The paper finds that many instances within the Rashomon set already perform quite well, or even better, than trees specifically tuned on different robustness properties.

**Additional Feedback:**

While the paper has a clear contribution, I am a bit surprised to see it in the benchmark track.
The paper does a good job of investigating the properties of the Rashomon set for the particular case of sparse trees. However, it remains somewhat unclear to me what the consequences for other researchers or practitioners are. In my opinion, the paper does not introduce a benchmark that can be used by other researchers. It is too much focusing on decision trees, and the provided Python code does not allow for easily switching the model class.

Furthermore, it would be good if the paper elaborates more on the consequences for practitioners. The paper briefly touches on model selection but does not elaborate further on this. In my opinion, this is important for a paper that focuses on empirically investigating the properties of smth. existing, but does not introduce any new algorithms/methods.

I am happy to discuss those points.

**Dataset Code Accessibility:**

Partly

**Dataset Code Comments:**

A few comments on the Python implementation:
- The README contains some placeholders at the end.
- Provide more details about the implementation in the README—for instance, clearly state which implementations can be found in which files.
- While most of the Python files are well documented, they still contain some ToDos and uncommented code.
- requirements.txt is a bit crowded. Please only specify the packages that must be actively installed by the user. Do not include dependencies that get installed automatically.
- Remove __pycache__ folders
- Remove .DS_store files

**Ethical Considerations:**

No, there are no or only very minor ethics concerns

**Final Justification:**

The authors were able to clarify all open questions and major concerns. Nevertheless, I still think that the paper should either include other model types or state the focus on decision trees more explicitly. I therefore only raised my score a bit.

**Limitations Weaknesses:**

- Practical implications are not well described
- Findings are limited to (sparse) decision trees
- The benefit for the research community is a bit unclear

Please see the section on "Additional Feedback" for details.

**Strengths Contributions:**

The paper is well-written and easy to follow. Furthermore, it appears to be scientifically sound as state-of-the-art methods are considered in the empirical evaluation.

---

> ### Author Rebuttal · Authors · 2025-07-30
>
> Thank you very much for your review and feedback.
>
> > Q: Practical implications are not well described.
>
> The practical implication in our paper is that practitioners do not necessarily need to solve constrained optimization problems to obtain models that satisfy trustworthiness goals such as robustness, privacy, and fairness. Instead, they can simply utilize the Rashomon set and it will very likely (according to the results) contain models that meet these secondary objectives. This is valuable in practice because solving the constrained optimization, often involving a specialized non-convex formulation or can be time-consuming (see Table 4). In contrast, once the Rashomon set is constructed, users can select trustworthy models post hoc at significantly lower cost. In this sense, our findings suggest a potential “free lunch” on trustworthiness empirically as the Rashomon set already contains models with desirable properties, without the need for custom optimization.
>
> Moreover, real-world model development is inherently iterative. Users or domain experts rarely accept the first model they see, instead, they often raise concerns and request alternatives after model evaluation and visualization. This often requires several rounds of problem reformulation, retraining, and reevaluation, which could again be really time-consuming. The Rashomon set approach aligns naturally with this process: instead of retraining from scratch, users can explore the precomputed set to find models that satisfy evolving needs, accelerating the modeling cycle. We believe that it will be useful for practitioners to see exactly how based on our framework.
>
> > Q: Findings are limited to (sparse) decision trees
>
> We understand your concern. However, there are a few important reasons for our choice of the hypothesis space.
>
> First, for the tabular dataset, sparse decision trees perform quite well, as good as black box models (please see McTavish et al., 2022 and Grinsztajn et al., 2022 for more details).
>
> Second, because of the logical structure of trees and the popularity of this hypothesis space, the research community has developed many algorithms for trees that directly optimize trustworthy objectives while providing guarantees of global optimality. For other hypothesis spaces, trustworthy algorithms are often greedy or post-hoc.
>
> Third, and most crucially, sparse decision trees currently represent the only hypothesis space for which the Rashomon set can be constructed exactly via full enumeration. For other model classes such as random forests, neural networks, and generalized additive models, only approximate methods are available to explore the Rashomon set. These approximations do not guarantee completeness, which can be problematic: if a trustworthy model is not found in the approximated Rashomon set, it is unclear whether this is due to the absence of such a model or due to approximation error. Therefore, we focused on sparse decision trees and TreeFARMS to allow conclusive and rigorous statements about the Rashomon set.
>
> However, to support our findings with additional hypothesis space, we conducted additional experiments on random forests for both fairness and robustness. To approximate the Rashomon set in this setting, we trained 100 random forests with different random seeds. As baselines, we used PostRF [1] for fairness (a post-hoc editing method) and GROOT-RF [2] for robustness (also a post-hoc method). We observed that this approximate Rashomon set tends to contain models that outperform the baselines in terms of both fairness and robustness.
>
> Please see the additional results in the tables below. "X Win Rate" refers to the proportion of models in the Rashomon set that outperform the baseline in metric X (e.g., accuracy, fairness, or robustness). "Joint Win Rate" denotes the proportion of models that outperform the baseline in both accuracy and the trustworthiness metric. Results are reported over 5 folds, with mean and std.
>
> Table 1: Fairness comparison between the Random forest Rashmon set constructed with different random seeds and PostRF.
> | Dataset            | Joint Win Rate       | Accuracy Win Rate (Avg ± Std) | Fairness Win Rate (Avg ± Std) |
> |--------------------|----------------------|-------------------------------|-----------------------------|
> | adult              | 0.08 ± 0.11          | 0.60 ± 0.21                    | 0.36 ± 0.33                   |
> | california-houses  | 0.00 ± 0.00          | 0.99 ± 0.02                   | 0.00 ± 0.00                   |
> default-credit     | 0.19 ± 0.15          | 0.43 ± 0.09                   | 0.46 ± 0.28                   |
> | diabetes-130US     | 0.44 ± 0.24          | 0.68 ± 0.18                   | 0.59 ± 0.29                   |
>
> Table 2: Robustness comparison between the Random Forest Rashomon set and baseline
> | Dataset                | Joint Win Rate       | Accuracy Win Rate (Avg ± Std) | Adversarial Accuracy Win Rate (Avg ± Std) |
> |------------------------|----------------------|-------------------------------|-------------------------------|
> | adult                  | 0.16 ± 0.10          | 1.00 ± 0.00                   | 0.16 ± 0.10                   |
> | california-houses      | 0.63 ± 0.24          | 1.00 ± 0.00                   | 0.63 ± 0.24                   |
> | default-credit         | 0.49 ± 0.24          | 0.75 ± 0.33                   | 0.51 ± 0.22                   |
> | diabetes-130US         | 0.01 ± 0.02          | 1.00 ± 0.00                   | 0.01 ± 0.02                   |
>
> We note that in the california-houses dataset, we were unable to find a fair model in the approximated Rashomon set. At this point, we cannot conclude whether this is due to limitations in our approximation method or because the baseline (PostRF) is a post-hoc method that may not correspond to any near-optimal model in the Rashomon set. If this experiment had been conducted in the sparse decision tree setting, we would have had a definitive answer due to the ability to enumerate the entire Rashomon set and the availability of fairness-optimal algorithms.
>
> [1] Ruicheng Xian and Han Zhao. A Unified Post-Processing Framework for Group Fairness, 2024.
>
> [2] Daniël Vos and Sicco Verwer. Efficient training of robust decision trees against adversarial examples. In: International conference on machine learning. 2021.
>
> > Q: The benefit for the research community is a bit unclear
>
> Our work can benefit the research community in the following aspects:
>
> (1) New insights that inspire future research directions: We show that the diversity of the Rashomon set can directly contribute to model trustworthiness. This frames model multiplicity as a resource and opens up new research directions, such as “When and why does model diversity lead to trustworthiness?”, “Under what conditions do multiple trustworthiness properties co-occur?”, “How can we quantify the relationship between diversity and trustworthiness?”, “Can similar effects be observed in more complex model classes?” Furthermore, trade-offs between trustworthiness metrics are often discussed but not well understood. The existence of the Rashomon set and our framework provides a concrete setting to study these trade-offs systematically.
>
> (2) A framework for trustworthy ML evaluation: We contribute a framework that integrates multiple trustworthiness metrics and attacks, including two robustness attacks and five privacy attacks. While our current experiments focus on sparse decision trees, components in this framework (Kanch attack, stability, membership inference attack, fairness evaluation) can be reused or modified and extended to other model classes for future study.
>
> > Q: A few comments on the Python implementation
>
> Thank you for your detailed suggestions regarding the code. We will update the README with clearer implementation details, clean up the code (including addressing TODOs), revise the requirements.txt to include only essential packages, and remove all pycache folders and .DS_Store files. We will also refactor the code into a modular structure to facilitate future extensions and adaptations.  In addition, we will provide example execution commands, example notebooks, and detailed instructions for reproducing the paper’s results and using the framework to evaluate new models or datasets.
>
> > Q: Additional feedback on contribution and track fit
>
> Thank you for your thoughtful comments. While our main contribution lies in the new insights gained from empirical analysis, namely, that when the Rashomon set is present, specialized optimization may not be necessary, we also provide a framework that integrates multiple model-agnostic trustworthiness metrics and attacks, which we believe can support future research beyond our specific focus on sparse decision trees.
>
> We acknowledge that our submission does not introduce a new dataset. However, we believe it aligns with the scope of the Datasets and Benchmarks track, particularly the call for “studies in data-centric AI that bring important new insight” and “systematic analyses of existing systems on novel datasets yielding important new insight.” Our work systematically evaluates multiple trustworthiness criteria across a diverse set of existing datasets and highlights an underexplored connection between model diversity and trustworthiness, which could inform future benchmark design and evaluation standards.
>
> > Q: Additional feedback on consequences for practitioners
>
> We will add more specific and expanded details on the framework's use, future work, and practical implications to the paper introduction and discussion as we gain more space if the paper is accepted. We will also restructure the code to make it more modular and extensible to support future methods across different model classes and we will provide further examples and documentation to support this.

---

> > ### Comment · Reviewer_gToP · 2025-08-01
> >
> > Thanks for the clarification.
> >
> > > We acknowledge that our submission does not introduce a new dataset. However, we believe it aligns with the scope of the Datasets and Benchmarks track, particularly the call for “studies in data-centric AI that bring important new insight” and “systematic analyses of existing systems on novel datasets yielding important new insight.” Our work systematically evaluates multiple trustworthiness criteria across a diverse set of existing datasets and highlights an underexplored connection between model diversity and trustworthiness, which could inform future benchmark design and evaluation standards.
> >
> > Got your point. However, the limited number of model types (sparse decision trees only) and somewhat limited number of data sets weakens the contribution a bit. While I do understand the rationale behind investigating decision trees (and random forests), I am still a bit unsure if this is enough to make the paper impactful. Let's see what the other reviewers have to say about that....

---

> > > ### Author Response · Authors · 2025-08-01
> > >
> > > Thank you for your timely reply. Sparse decision trees are a practical and widely used model class, particularly well-suited for high-stakes domains where interpretability is essential. They can handle most tabular datasets and have been successfully applied in diverse real-world settings, including stroke prediction [1], metamaterials analysis [2], HIV research [3], and lab tests [4]. Focusing on sparse decision trees allows us to study the phenomenon while maintaining strong practical relevance rigorously. As we noted in our previous response, our framework is not inherently limited to trees, and we also show that it can generalize to other model classes and provide preliminary results to support this extension.
> > >
> > > [1] Interpretable Classifiers Using Rules and Bayesian Analysis: Building a Better Stroke Prediction Model, AOAS 2015
> > >
> > > [2] How to See Hidden Patterns in Metamaterials with Interpretable Machine Learning, Extreme Mechanics Letters 2022.
> > >
> > > [3] Impact of cannabis use on immune cell populations and the viral reservoir in people with HIV on suppressive antiretroviral therapy, JID 2023
> > >
> > > [4] Urgency Prediction for Medical Laboratory Tests Through Optimal Sparse Decision Tree: Case Study With Echocardiograms. JMIR AI, 2025.

---

> > > > ### Comment · Reviewer_gToP · 2025-08-04
> > > >
> > > > I see. I fully understand why you selected decision trees, and I agree that they are highly relevant in practice. I will raise my score.
> > > >
> > > > Thanks for the response and further clarifications.

---

> > > > > ### Author Response · Authors · 2025-08-04
> > > > >
> > > > > Thank you for your thoughtful discussion and for raising the score. We sincerely appreciate it.

---

### Note · Authors · 2025-08-12

We thank all reviewers for their constructive feedback and thoughtful discussion. Our main contribution is showing that the Rashomon set (set of near-optimal models) of sparse decision trees often already contains models that match state-of-the-art methods explicitly optimized for trustworthiness (across robustness, fairness, and privacy), highlighting model diversity in the Rashomon set as a practical resource for trustworthy ML and model selection.

Besides minor clarifications, we will update the manuscript and codebase to:

* Clarify practical implications of sparse trees and potential for applying findings to other model classes given future Rashomon set algorithms (R gToP, va16, My5Y, kPVh).

* Add already computed training loss results and extra baseline results on the Rashomon set of random forests (R gToP, va16, My5Y, kPVh).

* Explicitly communicate the difference between model-intrinsic properties from Rashomon effect contributions (R N9bH).

* Improve code documentation, modularity, and usability (R gToP).

We appreciate the feedback and will incorporate these improvements in the next version of the manuscript. Thank you for helping us improve our work

---

### Decision · Program_Chairs · 2025-09-18

**Decision:**

Accept (poster)

**Comment:**

The Rashomon set is a set of near-optimal models (i.e., a set of models with similar performance to the optimal model). This paper proposes an evaluation framework for sparse decision trees in the Rashomon set which are enumerated by the TreeFARMS algorithm. Models in the Rashomon set are evaluated by the fairness, robustness, and privacy. This paper systematically compares sparse decision trees in the Rashomon set to trees optimized for specific criteria, such as stability to data perturbations, adversarial robustness, protection of privacy against membership attacks, unlearning a small portion of data, and fairness (statistical parity, equalized odds and equal opportunity). Experimental results on many data sets show that the Rashomon set of sparse decision trees contains models that are not only interpretable but also exhibit other desirable trustworthy properties.

I think that one strength of this paper is its usefulness since there is no exiting framework for evaluating models in the Rashomon set with respect to various evaluation criteria. Another strength is that interesting observations are obtained using the proposed framework such as "for real-world data sets, the Rashomon set often includes models that are as robust, stable, fair, and private as multiple base lines specifically optimized for one of these criteria" and "sparser trees tend to be more private and adversarially robust (the minimum complexity tree performs on average better than the maximum complexity tree for the majority of the data sets)".

Most reviewers pointed out one weakness of this paper: This paper evaluates only sparse decision trees in the Rashomon set. As a result, it is not clear whether some interesting observations for sparse decision trees can be generalized to other models (e.g., neural networks). Other reviewers asked questions about the computation time for enumerating models in the Rashomon set and the computation time for choosing a single model from the Rashomon set. It seems that the authors handled these issues successfully in the author-reviewer discussion phase. For example, the authors clearly explained why sparse decision trees are used in this paper. The authors also clearly explained that the computation time for enumerating all sparse decision trees in the Rashomon set is less time consuming if compared with the training time of a single model for a specific criterion using experimental results. As a result, the average score has increased to 4.80 after the author-reviewer discussion period.


I would like to recommend the acceptance of this paper for the following reason in addition to the high average score 4.80. Some of the obtained observations about the Rashomon set is interesting, and the use of the Rashomon set for the design of trustworthy ML models is an interesting and promising research direction.